# Human SARS-CoV-2 challenge uncovers local and systemic response dynamics

Rik G. H. Lindeboom[1,2,14 ✉], Kaylee B. Worlock[3,14], Lisa M. Dratva[1,13], Masahiro Yoshida[3], David Scobie[4], Helen R. Wagstaffe[5], Laura Richardson[1], Anna Wilbrey-Clark[1], Josephine L. Barnes[3], Lorenz Kretschmer[1], Krzysztof Polanski[1], Jessica Allen-Hyttinen[3], Puja Mehta[3], Dinithi Sumanaweera[1], Jacqueline M. Boccacino[1], Waradon Sungnak[1,6], Rasa Elmentaite[1,7], Ni Huang[1], Lira Mamanova[1], Rakesh Kapuge[1], Liam Bolt[1], Elena Prigmore[1], Ben Killingley[8], Mariya Kalinova[9], Maria Mayer[9], Alison Boyers[9], Alex Mann[9], Leo Swadling[10], Maximillian N. J. Woodall[11], Samuel Ellis[11], Claire M. Smith[11], Vitor H. Teixeira[3], Sam M. Janes[3], Rachel C. Chambers[3], Muzlifah Haniffa[1], Andrew Catchpole[9], Robert Heyderman[4], Mahdad Noursadeghi[4], Benny Chain[4], Andreas Mayer[4], Kerstin B. Meyer[1], Christopher Chiu[5], Marko Z. Nikolić[3,15 ✉] & Sarah A. Teichmann[1,12,13,15 ✉]

The COVID-19 pandemic is an ongoing global health threat, yet our understanding of the dynamics of early cellular responses to this disease remains limited[1]. Here in our SARS-CoV-2 human challenge study, we used single-cell multi-omics profiling of nasopharyngeal swabs and blood to temporally resolve abortive, transient and sustained infections in seronegative individuals challenged with pre-Alpha SARS-CoV-2. Our analyses revealed rapid changes in cell-type proportions and dozens of highly dynamic cellular response states in epithelial and immune cells associated with specific time points and infection status. We observed that the interferon response in blood preceded the nasopharyngeal response. Moreover, nasopharyngeal immune infiltration occurred early in samples from individuals with only transient infection and later in samples from individuals with sustained infection. High expression of *HLA-DQA2* before inoculation was associated with preventing sustained infection. Ciliated cells showed multiple immune responses and were most permissive for viral replication, whereas nasopharyngeal T cells and macrophages were infected non-productively. We resolved 54 T cell states, including acutely activated T cells that clonally expanded while carrying convergent SARS-CoV-2 motifs. Our new computational pipeline Cell2TCR identifies activated antigen-responding T cells based on a gene expression signature and clusters these into clonotype groups and motifs. Overall, our detailed time series data can serve as a Rosetta stone for epithelial and immune cell responses and reveals early dynamic responses associated with protection against infection.

COVID-19 is a potentially fatal disease caused by severe acute respiratory syndrome coronavirus 2 (SARS-CoV-2), which gave rise to one of the most severe global public health emergencies in recent history. Studies have uncovered that perturbed antiviral and immune responses to SARS-CoV-2 infection underlie severe and fatal outcomes. For example, impaired type I interferon responses[2,3], decreased circulating T cell and monocyte subsets[4–6] and increased clonal expansion of T cells and B cells[5] were associated with a more severe outcome. However,

accurate detection and interpretation of the immune response during COVD-19 has been hampered by heterogeneous responses caused by numerous factors that affect immune and clinical outcomes that are frequently unmeasurable and uncontrolled. These factors include infection characteristics such as viral dose, strain and time since exposure, together with clinical features including comorbidities, standard of care and pre-existing immunity. In particular, the observed immune response may represent different phases—from early viral detection

[1]Wellcome Sanger Institute, Wellcome Genome Campus, Cambridge, UK. [2]The Netherlands Cancer Institute, Amsterdam, The Netherlands. [3]UCL Respiratory, Division of Medicine, University College London, London, UK. [4]Research Department of Infection, Division of Infection and Immunity, University College London, London, UK. [5]Department of Infectious Disease, Imperial College London, London, UK. [6]Department of Microbiology, Faculty of Science, and Integrative Computational BioScience Center, Mahidol University, Bangkok, Thailand. [7]Ensocell Therapeutics, BioData Innovation Centre, Wellcome Genome Campus, Hinxton, UK. [8]Department of Infectious Diseases, University College London Hospital, London, UK. [9]hVIVO, London, UK. [10]Division of Infection and Immunity, Institute of Immunity and Transplantation, University College London, London, UK. [11]UCL Great Ormond Street Institute of Child Health, London, UK. [12]Theory of Condensed Matter, Cavendish Laboratory, Department of Physics, University of Cambridge, Cambridge, UK. [13]Present address: Wellcome MRC Cambridge Stem Cell Institute, University of Cambridge, Cambridge, UK. [14]These authors contributed equally: Rik G. H. Lindeboom, Kaylee B. Worlock. [15]These authors jointly supervised this work: Marko Z. Nikolić, Sarah A. Teichmann. ✉e-mail: r.lindeboom@nki.nl; m.nikolic@ucl.ac.uk; sat1003@cam.ac.uk

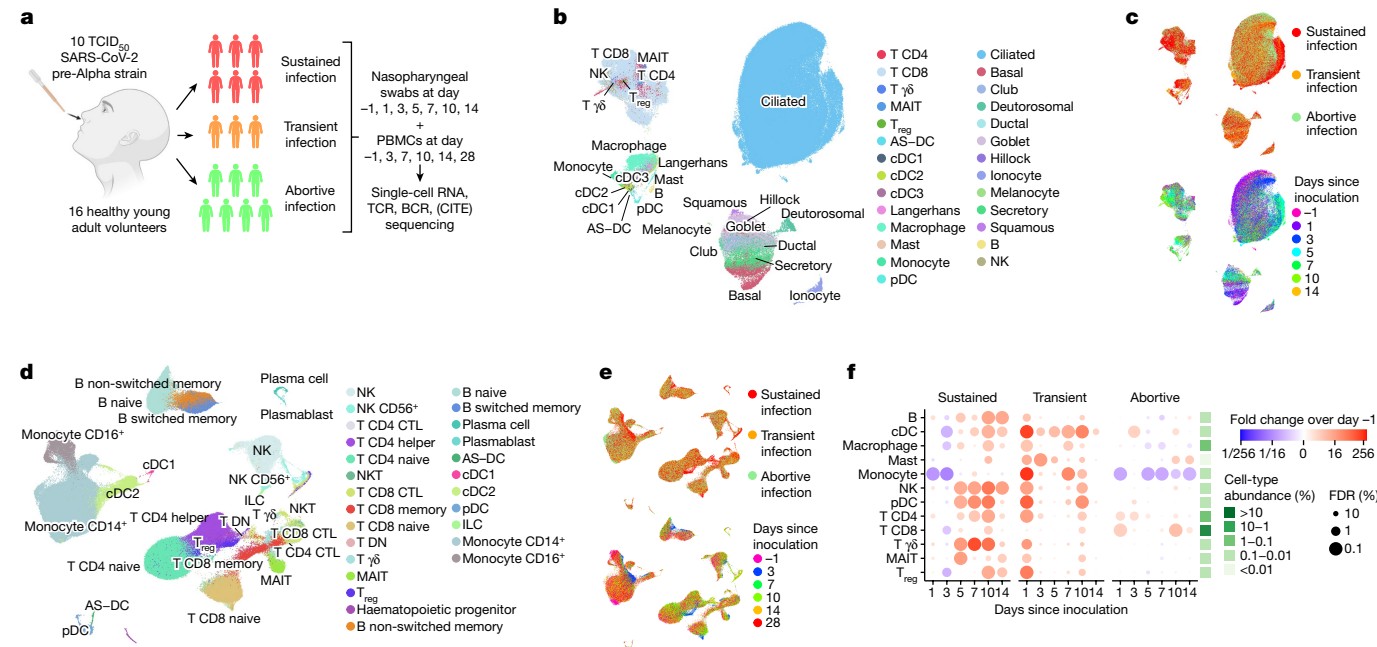

**Fig. 1 | Extensive temporal cell-state dynamics after SARS-CoV-2 inoculation. a**, Illustration of the study design and cohort composition. **b**,**c**, Uniform manifold approximation and projection (UMAP) plots of all nasopharyngeal cells (*n* = 234,182), colour coded by their broad cell-type annotation (**b**), by the infection group (**c**, top) and by days since inoculation (**c**, bottom). Only cells from sustained infection cases are shown in **c**, bottom. T<sub>reg</sub>, regulatory T cell; AS–DC, AXL<sup>+</sup>SIGLEC6<sup>+</sup> dendritic cell. **d**,**e**, UMAP plots as in **b** and **c**, but showing all PBMCs (*n* = 371,892). CTL, cytotoxic T lymphocyte; DN, double negative. **f**, Fold changes in abundance of nasopharynx-resident broad immune cell-type categories. Immune cell abundance was scaled to the total amount of detected epithelial cells in every sample before calculating the fold changes over days since inoculation compared with pre-infection (day −1) by fitting a GLMM on scaled abundance. Fitted fold changes are colour coded and we used the local true sign rate and Benjamini–Hochberg procedure to calculate false-discovery rates (FDRs), which are shown as the size of each dot. The mean cell-type proportions over all cells and samples are shown in the green heatmap to the right of the dot plot to aid the interpretation of changes in cell-type abundance. Illustration in **a** was created using BioRender (https://www.biorender.com).

to later adaptive responses—depending on the time between infection and sampling.

As the exact time at which patients were exposed to SARS-CoV-2 is nearly always unknown, it can be challenging to accurately delineate temporally restricted responses such as early interferon signalling and late adaptive immune responses[2–7]. Determining the dynamics of SARS-CoV-2 infection is therefore crucial for understanding how the immune response is orchestrated and how risk factors can affect this response. In addition, although many studies have investigated responses during the course of COVID-19 disease[8–10], it has thus far not been possible to study the early phases of exposure and the infection event itself in humans. In particular, studies of natural infection are unable to capture events in those who are exposed to the virus but do not develop sustained viral infection, which might be crucial in preventing dissemination and disease. Furthermore, the activation and expansion of antigen-responding T cells (versus bystanders) has been difficult to pinpoint in previous snapshot datasets[5,6]. Here we used a human SARS-CoV-2 challenge model and single-cell multi-omics multi-organ profiling to overcome limitations that complicate patient-based studies to decipher the antiviral responses against SARS-CoV-2 in a time-resolved manner.

## Human SARS-CoV-2 challenge model

To resolve epithelial and immune cell responses over time from SARS-CoV-2 exposure, we conducted a human SARS-CoV-2 challenge study[7]. In this model, young adults seronegative for SARS-CoV-2 spike protein were intranasally inoculated with a wild-type pre-Alpha SARS-CoV-2 virus strain (SARS-CoV-2/human/GBR/484861/2020) in a controlled environment. Before challenge, volunteers underwent

extensive screening to exclude risk factors for severe disease and to eliminate confounding effects of comorbidities. As risk mitigation and to maximize physiological relevance, participants were inoculated with the lowest culture-quantifiable inoculum dose of 10 tissue culture infectious dose 50 (TCID<sub>50</sub>). There were no serious adverse events and all symptoms were resolved in the participants selected for this single-cell data cohort.

We studied local and systemic immune responses at single-cell resolution in 16 participants. The highly controlled nature of this experimental model enabled baseline measurements on the day before inoculation. This was followed by detailed time series analyses (https://covid19cellatlas.org) of cellular responses after inoculation and subsequent infection, both systemically in blood and locally in the nasopharynx, to decipher antiviral responses against SARS-CoV-2 in a precise time-resolved manner.

Following inoculation, six participants from the cohort developed a sustained infection as defined by at least two consecutive quantifiable viral load detections by nasal and/or throat PCR along with symptoms (Fig. 1a and Extended Data Fig. 1). Three individuals produced multiple sporadic and borderline-positive PCR tests between day 1.5 and day 7 after inoculation. As these participants did not meet the earlier established criteria to be classified as 'sustained infection', we assigned them to a separate 'transient infection' group to investigate factors associated with this distinct phenotype (see Methods for considerations for infection group nomenclature).

Seven participants remained PCR-negative throughout the quarantine period, which indicated that these individuals successfully prevented the onset of a sustained or transient infection. Because these participants all remained seronegative but were observed to display early innate immune responses (see below), we termed these

abortive infections (as opposed to uninfected owing to, for example, antibody-mediated sterilizing immunity). The achieved infection rate of our model was similar to the infection rate observed in a closed household of unvaccinated individuals[11], which indicated that our administered viral dose was physiologically relevant.

## Cellular trends over time and infection

To comprehensively identify and temporally chart responses to SARS-CoV-2 exposure in these phenotypically divergent groups, we performed single-cell RNA sequencing (scRNA-seq) and single-cell T cell receptor (TCR) and B cell receptor (BCR) sequencing at up to seven time points (Fig. 1a and Extended Data Fig. 1a). In addition, in peripheral blood mononuclear cells (PBMCs), cellular indexing of transcriptomes and epitopes by sequencing (CITE-seq) measurements were used to quantify 123 surface proteins to aid cell-type annotation. At each time point, we collected PBMCs and nasopharyngeal swabs to study both the systemic immune response (PBMCs) and the epithelial and local immune response at the site of inoculation (swabs). Although most PBMC and nasopharyngeal time points were matched, we included more early nasopharyngeal and later PBMC time points as we anticipated more immediate local responses. In total, we generated more than 600,000 single-cell transcriptomes across 181 samples, which included 371,892 PBMCs and 234,182 nasopharyngeal cells. We used predictive models and marker gene expression to annotate a total of 202 cell states (Methods and Extended Data Figs. 2–4), including multiple newly identified cell states that will be discussed throughout this article. Notably, both datasets contained almost all expected cell types (Fig. 1b,d and Extended Data Fig. 2a,b), which enabled us to study both the local and systemic cellular response. Even when visualizing all cells at once (Fig. 1b,d), the 'infection group' and 'days since inoculation' marked specific groups of cells (Fig. 1c,e). This result indicated that there are large transcriptional and cellular changes over time and infection groups across the different cell-type compartments.

## Local immune infiltration

We first investigated how the immune landscape is affected by viral inoculation and subsequent infection. We used generalized linear mixed models (GLMMs) to quantify the changes in cell-type abundance over time since inoculation compared with the day before inoculation (day −1). This analysis enabled us to perform paired longitudinal modelling of donor-specific effects while accounting for technical and biological variation using random effect terms. In sustained and transient infections, we observed that all immune cell types significantly infiltrated the site of inoculation after exposure to SARS-CoV-2 (Fig. 1f). During sustained infections, immune infiltration started only at day 5 after inoculation and continued to increase until day 10. By contrast, transient infections led to immediate and substantial immune infiltration at day 1, followed by a decrease and smaller secondary infiltration event at day 10. Last, in the abortive infection group, we saw only a few changes, but this included the infiltration of CD4[+] and CD8[+] T cells on day 1.

Notably, both sustained and transient infections led to infiltration of innate and adaptive immune cells. In sustained infections, the increase in innate immune cells such as plasmacytoid dendritic cells (pDCs), natural killer (NK) cells, γδ T cells and mucosal-associated invariant T (MAIT) cells was quicker and of greater magnitude than infiltration by adaptive immune cells. Likewise, in transient infections, the increase in immune cells at day 1 was also greatest in the innate immune compartment. The observed difference in timing of immune infiltration between transient and sustained infections suggests that immediate immune recruitment and responses are associated with containing SARS-CoV-2 infection and preventing the progression to sustained replicative infection and COVID-19.

## Interferon response in blood before nose

We next attempted to detect antiviral gene expression programs in any of the tissue-resident and circulating cells during infection. Gene expression analysis revealed that interferon response genes made up the dominant infection-induced gene expression module in participants with sustained infection (Fig. 2a and Supplementary Table 1n–o). Interferon signalling was strongly activated in every cell type of both the blood and the nasopharynx (including epithelial cells), with some cell types such as circulating innate lymphoid cells (ILCs) and nasopharyngeal γδ T cells completely taking on a distinct interferon-stimulated cell state for the entire population at day 3 and day 5 after inoculation, respectively (Extended Data Fig. 5a,b and annotated in Extended Data Figs. 2–4 as IFN stim), which underscored the widespread and dominant effect of the interferon response pathway. Activation of interferon signalling was absent in abortive and transient infections (Extended Data Fig. 5a,c). Notably, at the site of inoculation, we only detected widespread interferon activation from day 5 after inoculation, whereas the interferon response in the blood peaked at day 3 after inoculation and seemed to be stronger. Using bulk RNA-seq data from an additional 20 individuals challenged with the virus[12], we were able to validate this observation ($P = 0.008669$ by Mann–Whitney $U$-test comparing the earliest time point when $z$ score-normalized interferon signalling exceeds quartile 3 in nose versus blood, median difference = 2 days; Extended Data Fig. 7h,i). The additional data also enabled further refinement of the timeline of the interferon response in the blood revealed that this rapid systemic response in circulating cells is initiated as early as day 2 after inoculation. This result is unexpected, as we assumed that the cells that reside in the inoculated tissue should be the first to respond through direct exposure to the virus and infected cells. It is possible that this observation is due to the lack of nasopharynx-associated lymphoid tissue access in this experimental clinical challenge study, and potentially a limitation of nasopharyngeal swab sampling.

## Rapid decrease in inflammatory monocytes

Investigating the potential role of professional antigen-presenting cells in the early immune response to SARS-CoV-2 revealed a decrease in circulating (cDC3 cells) and nasopharyngeal myeloid cells (multiple DCs, macrophages and monocytes) at day 3 in sustained infections. This was followed by an increase in myeloid cells at the site of inoculation only, which suggests that there is redistribution of myeloid cells between circulation and tissue compartments during early infection (Extended Data Fig. 7e,f and Supplementary Note 1). A significant decrease in circulating inflammatory monocytes (marked by *IL1B*, *IL6* and *CXCL3* high) was observed across all groups (Fig. 2b and Extended Data Fig. 7e), which suggested the presence of an immediate monocyte response, even if the infection was rapidly controlled. This marked effect implies that exposure alone in the absence of a sustained viral infection can result in detectable (but restricted) immune responses.

## MAIT cell activation

We next asked whether such a detectable immune response across all infection groups could also be observed in other cell types. When annotating unconventional T cells, we noted that MAIT cells in blood could be further divided into two subgroups: classical MAIT cells and activated MAIT cells with increased expression of cytotoxicity and activation markers such as *PRF1* and *CD27* (Extended Data Fig. 5d). These markers have previously been shown to be indicative of TCR-independent activation[13]. At day 3 after inoculation, we observed near complete activation of the entire MAIT cell population in the blood in sustained infections (Fig. 2c). Notably, the activation of MAIT cells was also present in abortive and transient infections, which suggests that MAIT cells may rapidly sense, either directly or indirectly, exposure

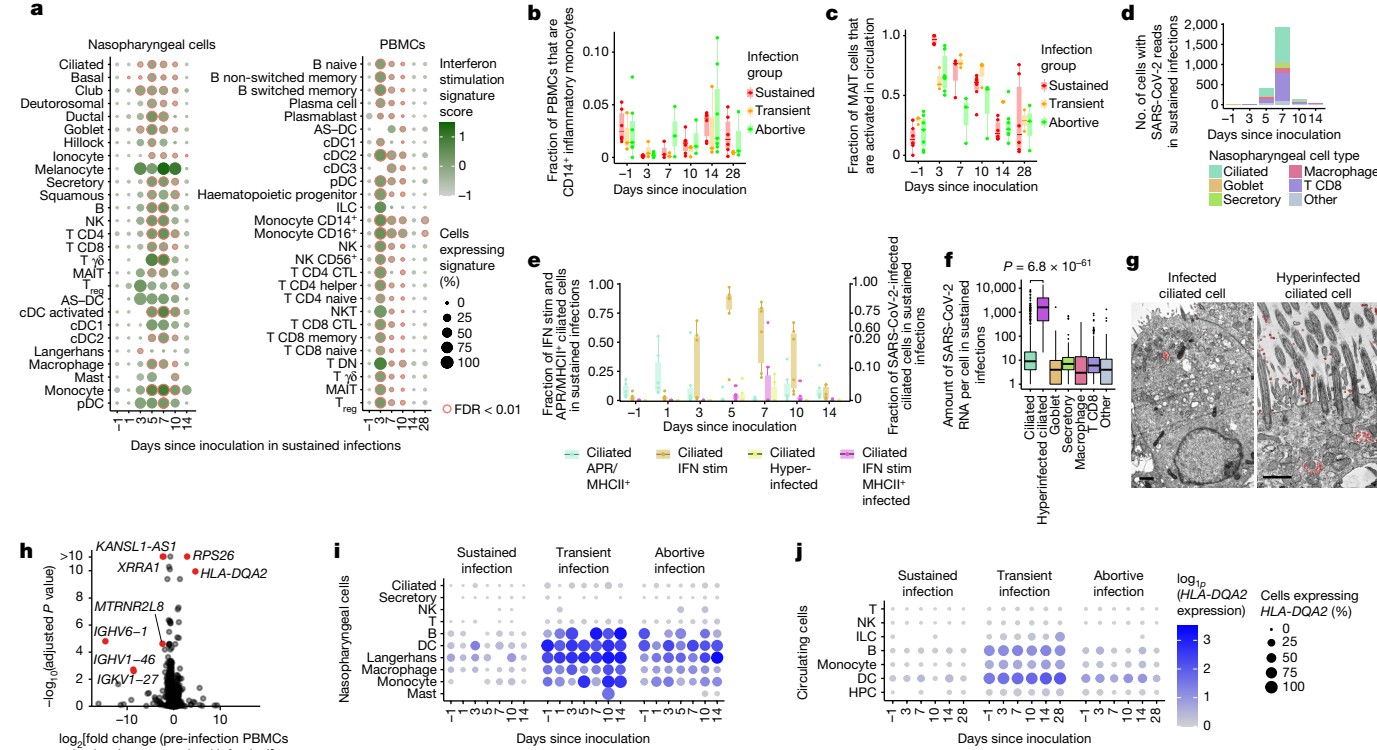

**Fig. 2 | Cell-state-specific antiviral responses and infection. a**, Mean expression of interferon-stimulated genes in participants with sustained infections for nasopharyngeal cells (left) and PBMCs (right). Red circles indicate significant change from day −1 by Bonferroni-corrected Mann−Whitney *U*-test. **b**, Relative fraction of circulating inflammatory monocytes over time since inoculation. *n* = 8,479 inflammatory monocytes examined over 73 unique samples. **c**, Fraction of circulating MAIT cells that are activated. *n* = 6,370 MAIT cells examined over 77 unique samples. **d**, Number of SARS-CoV-2⁺ nasopharyngeal cells (after background subtraction) in sustained infection cases. **e**, Fraction of ciliated cells that are annotated into response or infection cell states. Sustained infection cases are shown. Interferon-stimulated (IFN stim) and APR⁺ ciliated cells are shown on the left *y* axis, whereas infected ciliated cells are shown on the right *y* axis. *n* = 61,087 ciliated cells examined over 42 unique samples. MHCII, MHC class II. **f**, Number of viral reads per SARS-CoV-2⁺ cell. *P* value by

two-sided Mann−Whitney *U*-test, *n* = 2,505 infected cells examined over 12 unique samples. **g**, Representative transmission electron micrographs of infected (left) and hyperinfected ciliated cells (right) from SARS-CoV-2-infected in vitro human nasal epithelial cultures. Viral particles are false coloured red to aid visualization. Scale bar, 1 μm. **h**, Differential expression analysis of pre-inoculation PBMCs (day −1), comparing abortive with sustained infection cases. Red points highlight significantly changing genes with a FDR < 0.01 and a log₂ fold change > 2 or < −2. Nasopharyngeal analysis is shown in Extended Data Fig. 6e. Adjusted *P* values by Wald test accounting for sex, cell type and sequencing library identifier. **i,j**, *HLA-DQA2* expression in nasopharyngeal cells (**i**) and in PBMCs (**j**). HPC, haematopoietic progenitor cell. Significance for **b**, **c** and **e** is shown in Extended Data Fig. 7e−g. In all box plots, the centre line is the median, the box shows the interquartile range (IQR) and the whiskers are extreme values after removing outliers.

to a virus. Analyses of published fluorescence activated cell sorting (FACS) data from at-risk healthcare workers[14] validated the presence of an activated subpopulation of MAIT cells (Extended Data Fig. 5e,f). Thus, both MAIT cells and inflammatory monocytes might play a key part in the immediate response to SARS-CoV-2 infection. This finding further supports the notion that viral exposure that does not lead to a sustained infection and subsequent COVID-19 can still induce a detectable, yet restricted immune response.

## Detection of viral RNA peaks at day 7

To study how the observed immune responses relate to viral infection dynamics, we included the SARS-CoV-2 ssRNA genome and its transcripts in our analyses. This enabled us to quantify virions and viral gene expression alongside transcriptome dynamics of infected host cells. As expected, infected cells were almost exclusively found in the nasopharynx of participants with sustained infections (2,505 out of 2,512 cells with viral RNA). We detected infection of multiple cell types at day 5 after inoculation, which peaked at day 7 (Fig. 2d), followed by a rapid decrease at days 10–14 after inoculation, which highlighted the narrow time window over which SARS-CoV-2 virion production occurred. These changes over time were in line with quantitative PCR (qPCR) results (Extended Data Fig. 1b,c and Supplementary Table 1a,b), albeit with

the latter being more sensitive. We also observed viral reads in both immune and epithelial cells in the nasopharynx (Fig. 2d). We detected large numbers of SARS-CoV-2-containing CD8⁺ T cells. Although this result is unexpected because of the lack of ACE2 and TMPRSS2 expression (Extended Data Fig. 6a), infection of T cells by SARS-CoV-2 has been observed in vitro and in human lung tissue[15–17]. However, although we found evidence of productive viral infection of goblet and ciliated cells, SARS-CoV-2 RNA within T cells and macrophages seemed to be non-productive (Supplementary Note 2). Our results therefore show that although epithelial cells can support viral replication, mucosal CD8⁺ T cells are either infected non-productively or might capture viral fragments from surrounding cells.

## Hyperinfection of ciliated cells

Based on the detection of productive viral infections in ciliated and goblet cells, we sought to identify the cells that contributed the most to viral spread. We noted a small but distinct cluster of ciliated cells with an extremely high viral load (Fig. 2f and Extended Data Fig. 2b), in which we detected >1,000 viral RNAs per cell on average. Other infected cells typically contained <10 detectable viral RNAs. Although this hyperinfected subcluster of ciliated cells represents only 4% of all infected cells, they contained 67% of all detectable viral RNA. This

result uncovers a possible role for this subset of ciliated cells as major virion producers. In line with this finding, the hyperinfected ciliated cell state was the only epithelial or infected cell state for which the relative abundance significantly correlated with viral load as measured by qPCR with reverse transcription (RT–qPCR) (Extended Data Fig. 7j). Notably, gene expression analysis revealed that hyperinfected ciliated cells upregulated anti-inflammatory molecules while dampening the interferon response, which may contribute to viral spread and survival (Supplementary Note 3). We used transmission electron microscopy to validate that the viral load in SARS-CoV-2-infected ciliated cells in vitro can vary extensively, and that both infected and hyperinfected ciliated cells are distinguishable (Fig. 2g and Extended Data Fig. 9b).

## Ciliated cell acute-phase response

To further investigate the role of ciliated cells in the local response to SARS-CoV-2 infection, we delineated the ciliated cell compartment into a conventional ciliated state and four distinct dynamic cell states. In addition to the abovementioned interferon-stimulated, infected and hyperinfected clusters, we detected a relatively abundant subset of ciliated cells with high expression of genes involved in the acute-phase response (APR), antigen presentation and innate immunity, but lacking active interferon signalling (Extended Data Fig. 5g). Before infection, only a few ciliated cells showed this APR response, but in participants with sustained infections, up to 50% of ciliated cells become APR+ on day 1 after infection. (Fig. 2e and validated in Extended Data Fig. 5h). At day 3 after inoculation, interferon-stimulated ciliated cells emerged and peaked at day 5, at which point APR+ ciliated cells disappeared completely. At day 5, infected and hyperinfected ciliated cells started appearing, which peaked at day 7 after inoculation. At days 10–14, the number of interferon-stimulated cells decreased but remained higher than baseline, whereas APR+ ciliated cells re-emerged. Of note, APR+ ciliated cells were also immediately upregulated in abortive but not in transient infections, whereas all other ciliated cell states were present in sustained infections only (Extended Data Fig. 7g).

Together, this result underscores the highly dynamic nature of the ciliated cell compartment and uncovers a potential early-response role for APR+ ciliated cells. Notably, infected but not hyperinfected ciliated cells also activated APR genes, and APR+ ciliated cells with or without SARS-CoV-2 infection expressed major histocompatibility complex (MHC) class II molecules (Extended Data Fig. 6c). Although epithelial cells normally only express MHC class I molecules to present antigens to CD8+ T cells, there is evidence to indicate that inflammation and viral infection can also induce MHC class II expression in epithelial cells[18,19]. The colocalization of MHC class II+ ciliated cells with CD4+ T helper cells has previously been reported[19], and epithelial antigen presentation is a regulator of tissue-resident CD4+ T cell function[20]. This result therefore raises the possibility that MHC class II expression in infected ciliated cells could have additional antigen-presentation capabilities.

## HLA-DQA2 predicts infection outcome

Inspired by the identified and potentially protective responses in the non-sustained infection cases immediately after inoculation, we next set out to identify genes for which pre-infection expression levels could predict disease outcome. At the day before viral inoculation, *HLA-DQA2* expression in blood immune and nasopharyngeal cells was higher in participants in whom the virus did not succeed in establishing a sustained infection (Fig. 2h–j and Extended Data Fig. 6e,q). HLA-DQA2 is a poorly characterized non-polymorphic MHC class II molecule[21,22], whose increased expression in blood has been associated with milder COVID-19 progression[23,24]. Our data suggest that *HLA-DQA2* expression is indicative of protection against productive SARS-CoV-2 infections, which we confirmed using cross-validation and in our independent validation cohort (Extended Data Fig. 6f–h,q). This is, to our knowledge, the

first gene expression-derived predictor that is not based on acquired immunological memory.

## Identification of activated T cells

To investigate the anatomical and temporal distribution of CD4+ and CD8+ T cells following infection, we annotated the T cell compartment in blood (Fig. 3a) and nasopharynx (Fig. 3e) at high resolution into 54 distinct T cell states. These states included subtypes of CD4+, CD8+ and regulatory T cell states that expressed T cell activation markers such as *CD38*, *CD28*, *CD27* and *ICOS* at high levels (Fig. 3b), but did not upregulate classical activation-induced markers such as *CD40LG*, *CD69*, *LAMP1*, *TNFRSF9*, *TNFRSF4*, *IL2RA* and *CD274*. Although T cells that become activated during SARS-CoV-2 infection have so far been difficult to detect without enrichment experiments, we detected these activated T cells as distinct clusters in both the circulating and nasopharyngeal T cell compartments (Fig. 3a,e).

Reassuringly, many nasopharyngeal and circulating activated T cells expressed the same TCR sequences (Extended Data Fig. 6i), which showed that they originated from the same clones found both in circulation and nasopharynx as a response to infection. In addition, the immune repertoires of activated T cells were significantly more restricted and clonal than other mature T cell types (Extended Data Fig. 6p), which suggested that they were activated and expanded in a TCR-specific and antigen-specific manner. As expected from activation through TCR signalling, we also detected high frequencies of cycling T cells within the activated T cell compartment. Of note, many activated T cells were not cycling, and many cycling T cells did not seem to be activated, which implied that our activation signature was at least partially independent of the cell cycle gene signature.

To test whether these newly identified activated T cells are antigen-specific and can recognize SARS-CoV-2 peptides, we performed peptide–MHC staining on PBMCs using DNA-barcoded Dextramers loaded with SARS-CoV-2 antigens to detect peptide–MHC binding in parallel with scRNA-seq and single-cell TCR sequencing. These experiments revealed that activated T cells are significantly enriched and indeed specifically bind SARS-CoV-2 peptides compared with unmatched peptide–MHC molecules (Extended Data Fig. 8a–c,e). Together, the identification of activated T cells and their transcriptome signature in unsorted PBMC and tissue samples presents an opportunity to study the T cell response to SARS-CoV-2 in detail.

## Activated T cell dynamics

To better understand the characteristics of these activated T cells, we quantified their abundance over time and across infection groups (Fig. 3c,d,f). This analysis revealed significant expansion of activated CD4+ and CD8+ T cells, peaking in both blood and nasopharynx at day 10 after inoculation. This expansion was strongly time-restricted, only appearing in the circulation after day 7 and contracting rapidly thereafter, which we were able to confirm in our bulk-sequenced validation cohort (Extended Data Fig. 8g,h). Although this decrease meant that activated T cells were barely detectable at day 28 after inoculation, the associated TCR clonotypes in circulation could still be identified, having transitioned into memory and effector T cells (Extended Data Fig. 6j).

We integrated our single-cell resolved T cell data with highly sensitive bulk TCR sequencing from the blood to validate that activated T cell-associated TCR sequences indeed clonally expand after day 7 after inoculation in sustained infections (Fig. 3g) but not in abortive infections (Extended Data Fig. 6k). The emergence of these cells at day 10 after inoculation closely resembled the temporal dynamics of a typical antigen-specific adaptive immune response to vaccination and infection. At this time point, we also observed clearance of detectable virus and a reduction in interferon stimulation in the nasopharynx, which suggested that the onset of an adaptive T cell response is

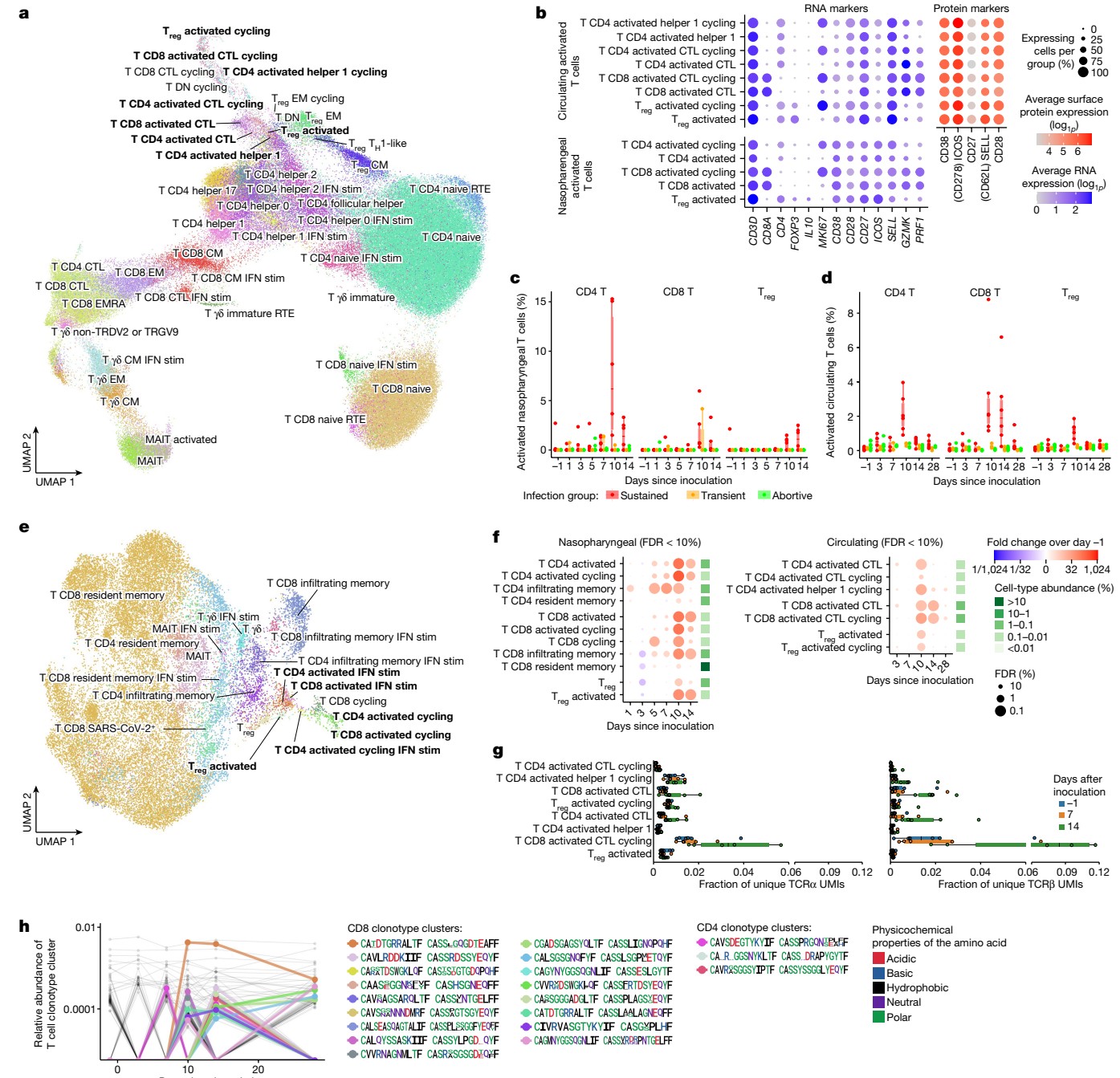

**Fig. 3 | Adaptive immune responses emerge at day 10 after inoculation.**
**a**, Circulating T cells (from PBMCs) across all infection groups, with distinct clusters of activated T cells highlighted in bold. CM, central memory; EM, effector memory; EMRA, CD45RA⁺ effector memory; RTE, recent thymic emigrant; T$_H$1, T helper 1. **b**, Marker gene and protein expression of activated T cell subsets. **c**, Percentages of nasopharyngeal T cells across all infection groups that were activated. $n$ = 28,426 T cells examined over 104 unique samples. **d**, Boxplot as in **c**, but showing circulating activated T cells. $n$ = 155,058 T cells examined over 77 unique samples. **e**, UMAP as in **a**, but showing nasopharyngeal T cells. **f**, Fold changes in conventional T cell state abundance compared with pre-inoculation in sustained infections. Only cell states that significantly change at a FDR < 10% at least once are shown. Nasopharyngeal T cell abundance was scaled to the total amount of detected epithelial cells.

Fold changes and significance were calculated by fitting a GLMM as shown in Fig. 1. The mean cell-type proportions over all cells and samples is shown in the green heatmap to the right of the dot plots. **g**, TCR clonality and expansion of activated T cells at day 14 in sustained infection cases was validated using bulk PBMC TCR sequencing. For TCRs that matched the single-cell gene expression, normalized clonality TCRα (left) and TCRβ (right) data are separated by type and expressed as the average fraction of total clones in sample contributed by a cell of that type. $n$ = 1,988 activated T cells examined over 30 unique samples. UMI, unique molecular identifier. **h**, Abundance of TCR clusters relative to all TCRs, with activated TCR clusters colour coded and their TCR motifs shown. In all box plots, the centre line is the median, the box shows the IQR and the whiskers are extreme values after removing outliers.

associated with clearance of the infection. Notably, activated T cells emerged in all participants with sustained infections, but in none of the individuals with abortive infections, a result that underscores their

specificity to infection. We did, however, detect a small increase in the number of activated T cells in the nasopharynx of two out of the three individuals with transient infections (Fig. 3c). This result suggests that

a smaller T cell response can be established without going through a sustained infection.

In contrast to activated CD4[+] and CD8[+] T cells for which infiltration peaked at day 10, the number of activated regulatory T cells was highest at day 14 at the site of infection (Fig. 3c), where they strongly upregulated expression of the anti-inflammatory cytokine *IL10* (Fig. 3b). This peak of activated regulatory T cells coincided with resolution of the observed global immune infiltrate (Fig. 1f) and downregulation of the interferon-stimulated response (Fig. 2a). This result suggests that these regulatory T cells have a role in suppressing further local inflammation after the infection has been cleared.

Notably, the time window during which activated CD8[+] T cells were increased was broader in blood (Fig. 3d), whereas activated CD4[+] T cells were detected for longer periods in the nasopharynx (Fig. 3c). In addition, activated CD4[+] T cells were significantly more abundant at the site of infection, where they represented up to 15% of all nasopharyngeal-resident T cells at day 10 after inoculation. The predominance of activated CD4[+] T cells in the respiratory mucosa was unexpected, as CD8[+] T cells are classically understood to be the major effectors in the local cytotoxic response. Activated CD4[+] T cells expressed high amounts of cytotoxicity genes (for example, *PRF1*; Fig. 3b and Extended Data Figs. 3a and 4a) that are normally expressed in NK and CD8[+] T cells. However, several studies have reported their emergence during the adaptive immune response against SARS-CoV-2 infection[25,26], and they have been reported to have a specific and antiviral effector function in influenza challenge models[27]. Taken together, these results suggest that CD4[+] T cells may play an unexpected and important part as local effectors.

In addition to conventional T cell responses, we observed a γδ T cell response that was dominated by γδ T cells that do not express TRDV2 and TRGV9. Concurrently, a marked B cell response was observed 10–14 days after inoculation. These observations are further discussed in Supplementary Notes 4 and 5.

## Cell2TCR identifies virus-specific TCRs

We next used distinct B cell and T cell states to identify BCR and TCR clonotypes, respectively, that specifically recognize SARS-CoV-2 (see Methods for details). We designed a cell-state-driven approach that enabled us to detect activated TCR and BCR clonotype groups by adaptive sequence divergence thresholding. We selected activated clonotype groups that seemed to expand in an antigen-specific manner. That is, they express multiple independent but highly similar TCR or BCR sequences in activated T cells or antibody-secreting B cells, respectively. Reassuringly, this selection method exclusively produced activated clonotypes in participants with sustained infections (Extended Data Fig. 6l,m). In total, we detected 20 activated TCR clonotype groups and 15 activated BCR clonotype groups in the 6 participants with sustained infections (Fig. 3h and Extended Data Fig. 7c). These clonotype groups first emerged after 1 week, with most appearing at day 10 and some remaining detectable at day 28 after inoculation. When we applied our Cell2TCR single-cell paired chain motif inference analysis pipeline on all activated CD8[+] T cells and on all HLA-matched CD8[+] T cell data from the Dextramer assay, we found 14 clonotype groups that contained cells from both datasets. This highly significant congruence validates our predictions of the SARS-CoV2 antigen recognition specificity of these clonotype groups (Supplementary Table 1c).

Notably, even at the peak of expansion at day 10 after inoculation, all but one of the activated clonotype groups had only very low abundance (<0.001% of all T cells), which is at the detection limit of single-cell genomics approaches. Such low prevalence makes activated clonotypes difficult to detect and distinguish from bystander cells when simply performing enrichment analysis of the entire immune repertoire between samples from healthy individuals and individuals with infection. This result highlights the importance of considering single-cell phenotypes in V(D)J analyses and the utility of our newly identified activated T cell-state expression signature.

Importantly, in contrast to activation or enrichment assays that require in vitro incubation with antigens[28,29], our Cell2TCR approach for detecting clonotypes that are activated in a disease of interest is not restricted or biased towards known antigens. Hence, it can be applied to any infection, inflammatory disease or cancer scRNA-seq and V(D)J sequencing dataset to extract paired chains that recognize antigens.

## Public SARS-CoV-2-specific TCR motifs

We proposed that our detailed characterization of the adaptive immune response in PBMCs could be leveraged when analysing data from patients with COVID-19, in particular to study activated T cell states and associated SARS-CoV-2-specific TCR repertoires. To this end, we integrated our data with scRNA-seq data from five large-scale PBMC studies using a deep generative model (scVI variational autoencoder, Methods). We obtained just under 1 million T cells from several hundred individuals, including more than 240 patients with acute COVID-19 (Supplementary Table 1d,e,m). We next projected our highly detailed cell-type annotation, including the activated T cell states, onto the patient data (Fig. 4a,c). This analysis revealed that activated T cells are also present in patients with COVID-19 who were sampled outside a viral challenge setting, and that these activated subsets formed distinct clusters of cells within the T cell compartment. Notably, the fraction of activated T cells was significantly higher in samples from patients with COVID-19 and individuals who had recovered from COVID-19 compared with healthy individuals, a result that underscores the involvement of these cells in the immune response to COVID-19 (Fig. 4b)

We then used our cell-state-aware clonotype group selection approach (Cell2TCR) to identify activated clonotypes. This analysis resulted in 29,486 COVID-19-associated clonotype groups, of which 326 comprised 2 or more distinct TCR clones (Supplementary Table 1f). Notably, 266 of these activated clonotype groups were shared among patients (largest groups shown in Fig. 4d), which highlighted the antigen-specificity of this approach, with the most common motif being shared by 18 individuals. This result also implies that a relatively small set of highly immunogenic SARS-CoV-2 peptides results in most of the T cell responses in COVID-19. Finally, we wanted to validate the antigen specificity of the COVID-19-associated clonotype groups that we found in the public datasets and our challenge study data. Thus, we intersected the CDR3 amino acid sequences with databases containing experimentally validated SARS-CoV-2-specific TCRs (Methods). Notably, this revealed that activated clonotype groups, including groups that contain TCRs from this study, are fivefold enriched for SARS-CoV-2-specific paired-chain TCRs compared with all other T cell states (*P* = 0.00044, Methods).

This result provides strong validation that activated T cells indeed represent the antigen-specific T cell response against SARS-CoV-2 (Extended Data Fig. 6n). Most of the activated T cell clonotype groups recognize viral proteins encoded by ORF1ab, but we also identified membrane-specific and spike-specific TCR clonotype groups. Because our cell-state-aware clonotype selection method identifies SARS-CoV-2-specific TCRs without any previous antigen information, our results may also include TCRs that recognize SARS-CoV-2 antigens that have not yet been tested. Together, these results validate the specificity of the adaptive immune response that we observed at day 10 and highlight the power of defining activated T cells for detecting disease-specific antigens in an unbiased manner.

## Molecular responses precede symptoms

Last, we investigated how the single-cell resolved timeline of immune responses related to clinical manifestations that are typically observed

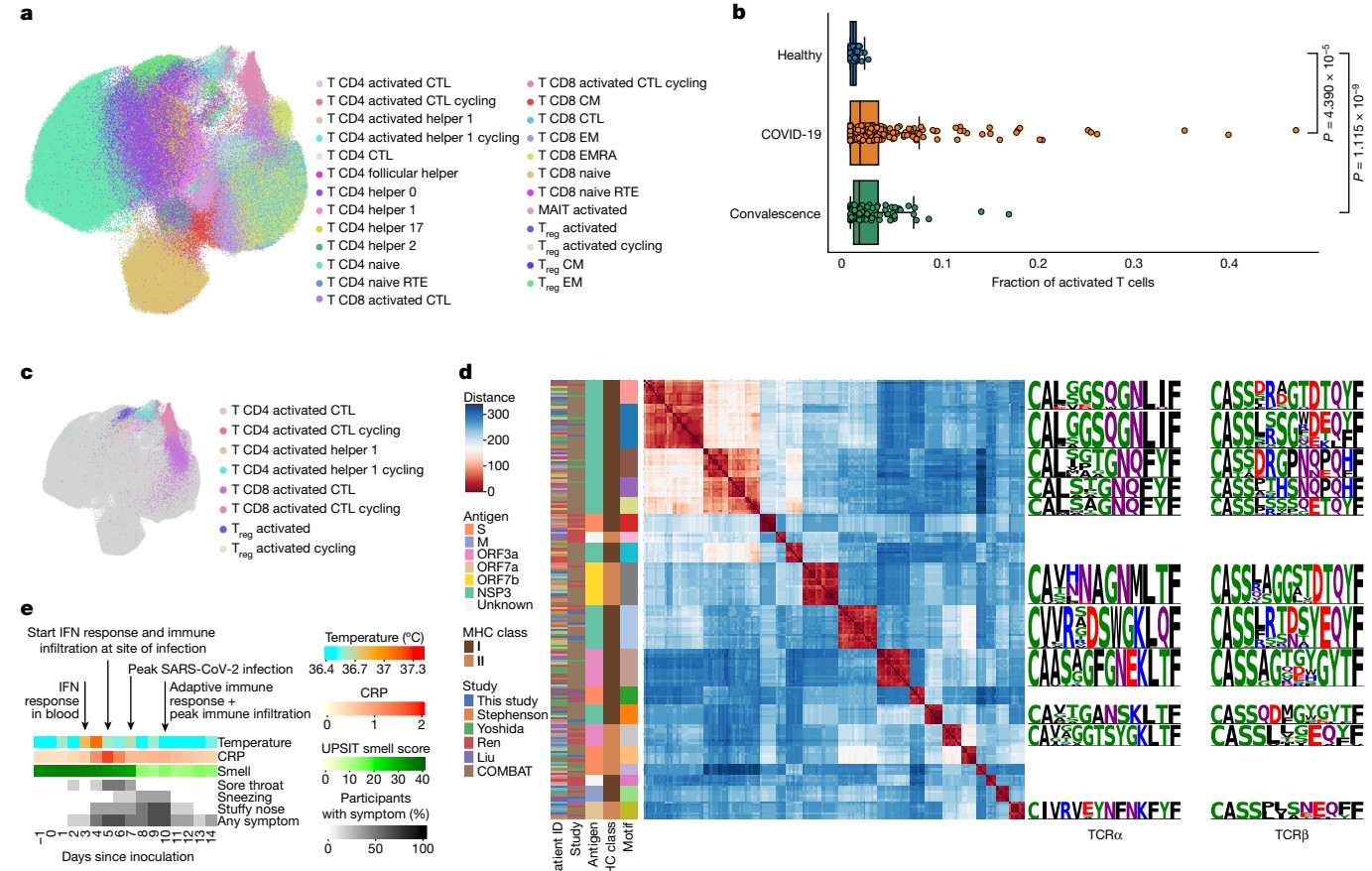

**Fig. 4 | Integrating COVID-19 patient data reveals public SARS-CoV-2 TCR motifs. a**, UMAP of the integration of five COVID-19 patient datasets ($n = 946,584$ T cells) with paired RNA and V(D)J sequencing data. Cell-type labels were inferred using CellTypist, trained on manual annotations of PBMCs from the current study. **b**, Fraction of activated T cells across all T cells in COVID-19 ($n = 240$ samples, 517,485 PBMCs), convalescent ($n = 82$ samples, 149,653 PBMCs) and healthy ($n = 88$ samples, 206,860 PBMCs) samples of five COVID-19 patient datasets. Significance levels after two-sided and uncorrected Mann–Whitney–Wilcoxon testing are shown. In all box plots, the centre line is the median, the box shows the IQR and the whiskers are extreme values after removing outliers. **c**, Activated T cell types highlighted on the UMAP representation from **a**. **d**, Clustermap of pairwise TCR distances with the sequence logos for 11 shared

motifs on the right-hand side. Each column and row corresponds to a unique TCR, and the distance to each TCR in the set is indicated by colour. Only activated T cells with public motifs shared by five or more individuals are shown. Low distances indicate similar TCRs, with distances of 40 and less potentially producing TCRs that recognize the same epitopes. For sequence logos, letter height indicates frequency of amino acid at that position across T cells pertaining to the motif. Amino acids are coloured by side chain chemistry: acidic (red), basic (blue), hydrophobic (black), neutral (purple), polar (green). Data were obtained from refs. 5,6,31–33. **e**, Clinical measurements and symptoms averaged over participants challenged with SARS-CoV-2 and developing sustained infection for day −1 to day 14 after inoculation. Major molecular events in the immune response are highlighted with arrows.

and measured in patients with COVID-19. The experimental setting of our human challenge model enabled us to collect highly detailed and time-resolved clinical data for all participants. The timing of the most relevant and dynamic COVID-19 features showed that even the earliest symptoms appeared mostly at day 4 after inoculation in sustained infections (Fig. 4e), which was later than some of the molecular responses that we described. Specifically, the upregulation of APR in ciliated cells, the activation of MAIT cells, depletion of inflammatory monocytes and the global activation of interferon signalling in blood were all observed before or at day 3 after challenge (Fig. 2).

By contrast, a slight increase in temperature was only significantly detectable at day 4 after inoculation ($P = 5 \times 10^{-6}$), at which early upper-airway-related symptoms such as nasal congestion and a sore throat also appeared. This was then followed by global immune infiltration and activation of interferon signalling at the site of infection at day 5, which was also the first time point that we detected infected cells. This coincided with a threefold increase in C-reactive protein (CRP) in blood ($P = 0.04$). At day 7 after inoculation we observed that the number of detectable infected cells peaked. Notably, from day 8 onwards, we also observed that all but one of the participants with a sustained

infection significantly lost their sense of smell ($P = 0.004$), together with worsening sneezing and nasal congestion. This was followed by a strong reduction in the number of infected cells at day 10 and a peak in the amount of nasopharyngeal immune infiltration, which coincided with the onset and expansion of an adaptive immune response and clearance of most symptoms. In summary, we observed that clinical manifestations and different waves of immune responses dynamically change over time, which can aid the molecular interpretation of COVID-19 based on clinical observations and improves our understanding of the therapeutic time windows in this disease.

## A dynamic human COVID-19 reference atlas

Finally, to optimize the utility of our time-resolved COVID-19 data, we used Gaussian process regression and latent variable models to predict the stage of immune response in 361 COVID-19 samples, which revealed that severe COVID-19 cases exhibit delayed immune responses (Supplementary Note 6). In addition, we provide annotation models for 202 cell states on CellTypist.org (https://www.celltypist.org) for simplified cell-type identification. We also make our single-cell expression data

accessible on COVID19CellAtlas.org (https://www.covid19cellatlas.org) for comprehensive online analysis.

## Discussion

Our single-cell human SARS-CoV-2 challenge study revealed several new insights (Extended Data Fig. 10). We detected multiple response states that precede the onset of clinical manifestations, including the activation of MAIT cells and decrease in inflammatory monocytes. These results represent newly discovered immune responses that emerge when exposure to SARS-CoV-2 does not lead to COVID-19. These monocyte and MAIT responses during very early and abortive infections can be used as biomarkers of an immediate immune response following viral exposure. During sustained infections that lead to COVID-19, we observed an immediate and new APR in ciliated cells at the site of infection. In addition, we discovered a distinct cell state for activated conventional T cells that harbour SARS-CoV-2-specific TCRs, and we showed that this signature can be projected onto patient cohort data to identify disease-specific T cell responses.

In sustained infections, we observed global activation of interferon signalling that affected all circulating immune cells. Unexpectedly, the activation of interferon signalling in blood precedes widespread activation at the site of inoculation, which might indicate that a highly efficient relay to the systemic immune system exists, possibly through the lymphatic system, which we are missing in this study set-up. The activation of interferon signalling at days 5–7 after inoculation coincides with global immune infiltration and a peak of detectable virally infected cells. This relatively slow immune infiltration at the site of inoculation is in contrast to the immediate immune infiltration that we observed in infections that were only transiently detectable. Our data suggest that individuals with high *HLA-DQA2* expression are better at preventing the onset of a sustained viral infection.

In sustained infections, we also detected large numbers of cells containing viral RNA, including immune cells, but we provided evidence that only epithelial cells support successful viral replication. Here we found that a small subset of hyperinfected ciliated cells becomes anti-inflammatory and a major source of viral production. We provide electron microscopy evidence for large heterogeneity in infection levels across ciliated cells in vitro.

The timing of our challenge experiments in the early stages of a pandemic with a new virus—before most of the population acquired immune memory through natural infections and vaccine rollout—enabled us to recruit and study immune responses in adult participants who were completely naive to this pathogen. The resulting data will be essentially impossible to replicate in future efforts as the population builds memory to many SARS-CoV-2 strains. In addition to the responses during sustained infections and COVID-19, we were able to study abortive and transient infections that would be difficult to detect outside a controlled challenge setting, and we revealed previously unknown immune response signatures associated with successfully preventing sustained infections.

Although our results included matched pre-infection samples and almost all expected cell types from a total of 181 samples from 16 participants, we cannot exclude the possibility that our infection group sizes remained underpowered to detect subtle or time-restricted responses. We also note that neutrophils, which play an important part in COVID-19, are frequently under-represented in microfluidics based scRNA-seq[30]. This limitation is probably further exacerbated by cryopreservation of samples used within this study. In addition, the participants enrolled in this study cleared the infection with mild symptoms, which means that caution should be taken when extrapolating our findings to patients critically ill with COVID-19.

At day 10 after inoculation, we detected the onset and expansion of the adaptive immune response. In addition to antibody-secreting B cells, this response includes activated conventional T cells. This is the first time, to our knowledge, that these cells have been described in single-cell transcriptomics assays, which may be because of the limited early time window in which these activated T cells are detectable. Two weeks after inoculation, the amount of activated regulatory T cells at the site of inoculation peaks, whereas the abundance of other immune cells normalizes again, which coincides with a near absence of any remaining infected cells. These activation states have key marker genes, and we can identify these activated CD4+, CD8+ and regulatory T cell states using machine-learning models. We integrated their prediction into a computational pipeline (Cell2TCR), which includes paired chain TCR motif inference. This is a tool applicable to any scRNA-seq and V(D)J dataset, including those from infection, inflammation, tumour immune response and healthy samples.

Together, this study provides a comprehensive and detailed time-resolved description of the course of mild SARS-CoV-2 infection, or any other infectious disease, and gives new insights into responses that are associated with resisting a sustained infection and disease.

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

## Methods

### Study participants and design

Sixteen healthy adults aged 18–30 years, with no evidence of a previous SARS-CoV-2 infections or vaccinations (seronegative), were included for scRNA-seq sample processing and analysis from the wider cohort (36 participants) enrolled as part of the human SARS-CoV-2 challenge study, pioneered by the government task force, Imperial College London, Royal Free London NHS Foundation Trust, University College London and hVIVO[7]. These participants were enrolled as part of cohorts 5 and 6, from June to August 2021. Additionally, 20 healthy adults were included as part of the same study (earlier cohorts)[7], and blood and nasal (mid-turbinate) samples were processed for bulk RNA-seq as previously described[12] (see Supplementary Table 1q for an overview of the bulk RNA-seq validation cohort and samples included). Of these participants, ten individuals received pre-emptive remdesivir as previously described[7]. Volunteers were tested for the presence of anti-SARS-CoV-2 protein antibodies using a MosaiQ COVID-19 antibody microarray (Quotient) before enrolment and excluded based on a positive test, as well as on risk factors assessed by clinical history, physical examinations and screening assessments. See ref. 7 for the full list of inclusion and exclusion criteria and for further details regarding the challenge set-up and ethics. In brief, written informed consent was obtained from all volunteers before screening and study enrolment. The clinical study was registered with ClinicalTrials.gov (identifier NCT04865237). This study was conducted in accordance with the protocol, the Consensus ethical principles derived from international guidelines, including the Declaration of Helsinki and Council for International Organizations of Medical Sciences International Ethical Guidelines, applicable ICH Good Clinical Practice guidelines, and applicable laws and regulations. The screening protocol and main study were approved by the UK Health Research Authority—Ad Hoc Specialist Ethics Committee (reference: 20/UK/2001 and 20/UK/0002).

Participant 11, who fulfilled enrolment criteria, was later found to have low pre-inoculation levels of neutralizing and spike-binding antibodies (see serum antibody titre methods below). This individual was classified as an abortive infection based on virus kinetics (see virology method below). When tested, the exclusion of this individual was found not to alter any of our conclusions (data not shown).

The participants were followed for 1 year after inoculation, with continued samples and metadata collected for the use in future studies and to benefit the research community. No participants enrolled in the study were observed to present with any long-COVID symptoms at this final time point (1 year), which included an interview by a study clinician to assess for symptoms and a complete physical examination. The UPSIT scores for all participants had returned to baseline and no other symptoms were reported, with physiological observations and physical examination of vital signs were all seen to be normal (including temperature, heart rate, blood pressure, respiratory rate, saturation of peripheral oxygen level [SpO$_2$], spirometry and electrocardiogram). Of note, although most symptoms were seen to spontaneously resolve themselves, one participant (participant 2) out of the six total who reported anosmia or dysosmia as part of the single-cell cohort received additional smell training and a short course of steroids (28 days after inoculation)[7]. This study, however, focused primarily on the first 28 days after inoculation (with the exception of 46 days for one participant as noted below, see sample collection below).

Of note, after the participants were discharged from quarantine and before their day 28 follow-up (when additional blood samples were collected), two participants reported either to have had their first SARS-CoV-2 vaccine (participant 9) or a community infection (participant 7). In brief, participant 9 had their first vaccine on day 14 after inoculation (2 weeks before the day 28 sample was taken). Participant 7 tested positive before their day 28 visit was due. The follow-up was therefore delayed by 2 weeks, resulting in the day 28 sample for

this participant instead being taken at day 46 after inoculation. ELISpot performed on this participant revealed a response in the day 28 and day 90 samples (data not shown). Moreover, participant 8 tested positive on day 29 after inoculation, a day after their day 28 sample was taken. However, for this participant, the ELISpot showed no response at day 28 and a small response at day 90. See Extended Data Fig. 1a for overview of the samples and time points included from each participant. These individuals and time points were found not to alter any of our conclusions.

### Challenge virus

Participants were intranasally inoculated with a wild-type pre-Alpha SARS-CoV-2 challenge virus (SARS-CoV-2/human/GBR/484861/2020) at dose 10 TCID$_{50}$ at day 0. A volume of 100 μl per naris was pipetted between both nostrils and the participant was asked to remain supine (face and torso facing up) for 10 min, followed by 20 min in a sitting position wearing a nose clip after inoculation to ensure maximum contact time with the nasal and pharyngeal mucosa. Mid-turbinate nose and throat samples were collected twice daily using flocked swabs and placed in 3 ml of viral transport medium (BSV-VTM-001, Bio-Serv) that was aliquoted and stored at −80 °C to evaluate viral kinetics (infection status) as described in the section 'Virology' below. Participants remained in quarantine for a minimum of 14 days after inoculation until the following discharge criteria were met: two consecutive daily nose and/or throat swabs with no viral detection or a qPCR Ct value > 33.5 and no viable virus by overnight incubation viral culture with detection by immunofluorescence. For details of the protocol and ethics used within the human SARS-CoV-2 challenge study, see the 'Challenge virus' section of the methods in ref. 7.

### Sample collection for scRNA-seq cohort

**Nasopharyngeal swabs.** Samples were collected at the Royal Free Hospital by trained healthcare providers at 7 time points: day −1 (pre-inoculation) and days 1, 3, 5, 7, 10 and 14 after inoculation. The participants were asked to clear any mucus from their nasal cavities, and nasopharyngeal samples were collected using FLOQSwabs (Copan flocked swabs, ref. 501CS01) inserted along the nasal septum, above the floor of the nasal passage to the nasopharynx until a slight resistance was felt. The swab was then rotated in this position in both directions for 10 s and slowly removed while still rotating and immediately stored in a pre-cooled cryovial on wet ice containing freeze medium (90% heat-inactivated FBS and 10% dimethyl sulfoxide (DMSO)). On wet ice, the cryovials were transferred to the hospital chutes where they were sent down to the laboratory (<2 min at room temperature), placed in a slow-cooling device (Mr. Frosty Freezing Container, Thermo Fisher Scientific) and stored at −20 °C until all samples were collected, at which point they were moved to −80 °C freezers for at least 48 h for optimum freezing. Samples were moved and stored in liquid nitrogen for later processing.

**PBMC isolation from peripheral blood.** Peripheral whole blood was collected at the Royal Free Hospital in EDTA tubes at 5 time points: day −1 (pre-inoculation) and days 3, 5, 10, 14 and 28 after inoculation. Each day, the blood was transferred at room temperature to Imperial College London for fresh isolation and collection of PBMCs by means of Histopaque Ficoll separation (Merck, H8889-500ML). The peripheral whole blood was first diluted 1:1 with 1× PBS (Merck, D8662-500ML) before being gently overlaid onto a maximum of 15 ml of Histopaque, at a ratio of 2:1 (blood to Histopaque). The samples were then centrifuged at 400g (with no breaks) for 30 min at room temperature and the PBMC white buffer layer was collected, washed (with PBS about 50 ml) and spun down (400g for 10 min at room temperature), before the supernatant was carefully discarded and the cell pellet was resuspended in 10 ml PBS. The cells were filtered using a 40 or 70 μm cell strainer and then both the cell number and viability were assessed using

Trypan Blue. The cells were further centrifuged (400*g* for 10 min) and resuspended in the required volume of cell freezing medium (90% FBS (Sigma, F9665-500ML) and 10% DMSO (Sigma, D2650-100ML)), before being cryopreserved at −80 °C using a slow-cooling device. The blood and nasopharyngeal samples were collected within 2 h of each other.

## Clinical assessments

Participants were carefully monitored and assessed daily using an array of blood tests, spirometry, electrocardiograms and clinical assessments (vital signs, symptom diaries and clinical examination). Full details of all the safety and clinical data collected with the human SARS-CoV-2 challenge study can be obtained in the methods in ref. 7, with an overview of metadata and demographics for the 16 participants enrolled for the scRNA-seq part of this study (up to 28 day after inoculation) in Supplementary Table 1g.

## Virology

From 24 h after inoculation, twice daily samples (swabs) were taken at 12-h intervals from both the nose (mid-turbinate) and throat (pharyngeal) to assess and quantify the viral kinetics of each participant before and after inoculation (morning and afternoon) for their quarantine period (minimum 14 days, which was extended with the continued detection of virus). These were measured using two independent assays: (1) RT−qPCR with N gene primers/probes adapted from the Centers for Disease Control and Prevention protocol[34] (updated 29 May 2020); and (2) quantitative culture by focus forming assay (FFA). For full details of each assay and statistical analysis, refer to the methods in ref. 7.

The lower limit of quantification (LLOQ) for RT−qPCR was 3 $\log_{10}$ copies per ml, with positive detections less than the LLOQ assigned a value of 1.5 $\log_{10}$ copies per ml and undetectable samples assigned a value of 0 $\log_{10}$ copies per ml. Only samples in which participants presented with consecutive positive RT−qPCR results were further tested using the FFA assay. In the FFA, the LLOQ was 1.27 FFU ml$^{-1}$. Viral detection less than the LLOQ was assigned 1 $\log_{10}$ FFU ml$^{-1}$, and undetectable samples were assigned 0 $\log_{10}$ FFU ml$^{-1}$.

Infection intervals for each participant were calculated based on the time of the first and last RT−qPCR test with detectable virus (across the nose and/or throat), time points in which tests below the LLOQ (1.5) were also counted if they occurred <2 days of a quantifiable (>LLOQ) test result.

An overview of the virology in each of the 16 participants included in the single-cell cohort (<28 days after inoculation) is provided in Extended Data Fig. 1b,c, with CT and FFA (virus titre) values provided in Supplementary Table 1a,b,h,i.

## Infection group nomenclature

A sustained laboratory-confirmed infection was defined as quantifiable RT−qPCR detection greater than the LLOQ from mid-turbinate and/or throat (pharyngeal) swabs on 2 or more consecutive 12-h time points, starting from 24 h after inoculation and up to discharge from quarantine. Participants for whom only a single or two non-consecutive RT−qPCR tests returned quantifiable results (>LLOQ) were classified as transient infections. Participants for whom no RT−qPCR tests returned quantifiable results (>LLOQ) were classified as abortive infections (Extended Data Fig. 1b and Supplementary Table 1a,b,h,i). The nomenclature of sustained, transient and abortive infection groups was carefully chosen based on the hypotheses that viral exposure through inoculation leads to sustained, transient and aborted viral replication, respectively, in these participants. Here sustained infection events resemble typical COVID-19 cases, whereby after viral infection, the virus spreads through the upper airway tissues and replicated to highly detectable levels. Transient infections represent a new group of cases whereby we propose that successful but limited replicative infection has taken place, leading to viral loads that were borderline detectable.

Finally, we propose that non-replicative viral infection (that is, abortive viral infections) has taken place in the participants who belong to the abortive infection group.

## Nasopharyngeal swab dissociation and processing for scRNA-seq

Following freezing, nasopharyngeal swabs were transferred to a category level 3 facility at University College London, stored and processed in batches of 7–8 samples at a time to a single-cell suspension. All work was carried out in a MSC class I hood in compliance with standard category level 3 safety practices. The dissociation and collection of cells from nasopharyngeal swabs was carried out in accordance with the previously described protocol[35,36], with minor modifications. This approach involves multiple parallel washes and digestion steps using both the nasopharyngeal swab and collected freezing and wash medium to help ensure maximum cell recovery. First, samples are exposed to DTT for 15 min, followed by an Accutase digestion step for 30 min, before cells from the same sample (collected directly from the swab or the freezing medium and washes from that swab) are quenched, pooled and filtered before checking cell number and viability.

In brief, samples were rapidly thawed (tube A) and the liquid collected in an empty 15 ml Falcon tube (tube B). The cryovial, lid and swab was then carefully rinsed three times with 1 ml warm RPMI 1640 medium, which was added dropwise to the 15 ml tube while gently swirling the tube to slowly dilute the DMSO from the freezing medium to help prevent the cells bursting. After waiting 1 min, the tube (tube B) was then topped up with an extra 2 ml of warm RPMI 1640 medium and centrifuged at 400*g* for 5 min at 4 °C. The cell pellet was then resuspended in RPMI 1640 and 10 mM DTT (Thermo Fisher, R0861), and incubated for 15 min on a thermomixer (37 °C, 700 r.p.m.), centrifuged as above and the supernatant was aspirated and the cell pellet was resuspended in 1 ml Accutase (Merck, A6964-500ML). This was then incubated for a further 30 min on the thermomixer (37 °C, 700 r.p.m.).

In parallel to the processing of the cell freezing medium and washes above, the swab was moved to a new 1.5 ml Eppendorf tube (tube C) containing 1 ml RPMI 1640 and 10 mM DTT and placed on the thermomixer (37 °C, 700 r.p.m.) for 15 min. In accordance with the steps above, the swab was next transferred to a new 1.5 ml Eppendorf (tube D) containing 1 ml Accutase and incubated with agitation (700 r.p.m.) at 37 °C. The 1 ml RPMI 1640 and 10 mM DTT from the nasopharyngeal swab incubation (in tube C) was centrifuged at 400*g* for 5 min at 4 °C to pellet cells, the supernatant was discarded, and the cell pellet was resuspended in 1 ml Accutase and incubated for 30 min at 37 °C with agitation (700 r.p.m.).

Following the Accutase digestion step, all cells were combined (tubes B, C and D) and filtered using a 70 μm nylon strainer (pre-wetted with 3 ml quenching medium: RPMI 1640, 10% FBS and 1 mM EDTA (Invitrogen, 1555785-038)) in a 50 ml conical tube (tube E). The filter, tubes and swab were then further thoroughly rinsed with quenching medium to collect all cells, and the washes were combined. The dissociated, filtered cells (tube E) were then centrifuged at 400*g* for 5 min at 4 °C, and supernatant discarded. The cell pellet was resuspended in residual volume (about 500 μl) and transferred to a new 1.5 ml Eppendorf tube (tube F). Tube E was then washed with a further 500 μl of RPMI 1640 with 10% FBS and combined with tube F, centrifuged as above, the supernatant removed and the cells resuspended in 20 μl RPMI 1640 and 10% FBS. Using Trypan Blue, total cell counts and viability were assessed. The cell concentration was adjusted for 7,000 targeted cell recovery according to the 10x Chromium manual before loading onto a 10x chip (between 700 and 1,000 cells per μl) and processing immediately for 10x 5′ single-cell capture using a Chromium Next GEM Single Cell V(D)J Reagent kit v.1.1 (Rev E Guide). For samples in which fewer than 13,200 total cells were recovered, all cells were loaded.

Note that owing to the sample type, necessary freezing process and no access to a class 3 flow facility to sort viable cells, the majority of the

samples processed were seen to have low viability (ranging from 5.4% to 57.85%, with the average viability of samples processed being 26.89%).

## PBMC CITE-seq staining for single-cell proteogenomics

Frozen PBMC samples were thawed and processed in batches of 16 to enable a carefully designed pooling strategy. Here each sample was pooled twice into two distinct pools containing up to four PBMC samples per pool from mixed time points. Note that only one sample from each donor was ever pooled together at a time to assist with subsequent demultiplexing. This pooling strategy was used to help remove and correct for any protocol-based batch effects.

In brief, PBMC samples were rapidly thawed at 37 °C in a water bath. Warm RPMI 1640 medium (20–30 ml) containing 10% FBS (RPMI 1640 and FBS) was added slowly to the cells before centrifuging at 300$g$ for 5 min. This was followed by a wash in 5 ml RPMI 1640 and FBS. The PBMC pellet was collected, and the cell number and viability were determined using Trypan Blue.

PBMCs from 4 different donors were then pooled together ($1.25 \times 10^5$ PBMCs from each donor) to make up $5.0 \times 10^5$ cells in total. The remaining cells were used for DNA extraction (Qiagen, 69504). The pooled PBMCs were resuspended in 22.5 µl cell staining buffer (BioLegend, 420201) and blocked by incubation for 10 min on ice with 2.5 µl Human TruStain FcX block (BioLegend, 422301). The PBMC pool was then stained with TotalSeq-C Human Cocktail, V1.0 antibodies (BioLegend, 399905) according to the manufacturer's instructions (1 vial per pool). For a full list of TotalSeq-C antibodies (130 antibodies and 7 isotype controls) refer to Supplementary Table 1j. Following a 30-min incubation period with the TotalSeq-C Human Cocktail V1.0 antibodies (at 4 °C in the dark), the PBMCs were topped up using cell staining buffer and centrifuged down to a pellet (500$g$ for 5 min at 4 °C), discarding the supernatant. The pellet was then resuspended and washed in the same manner 2 more times using the resuspension buffer (0.05% BSA in HBSS), before finally being resuspended in 20–30 µl resuspension buffer and counted again. The PBMC pools were then processed immediately for 10x 5′ single-cell capture (Chromium Next GEM Single Cell V(D)J Reagent kit v.1.1 with Feature Barcoding technology for cell Surface Protein-Rev D protocol). A total of 25,000 cells were loaded from each pool onto a 10x chip.

## PBMC Dextramer staining for SARS-CoV-2 antigen-specific T cell enrichment and single-cell sequencing

To further validate and investigate the SARS-CoV-2 antigen-specific T cell populations in our single-cell dataset, day 10, 14 and 28 post-inoculation PBMCs samples from all 16 participants were further enriched and processed for single-cell sequencing using a multi-allele panel of 44 SARS-CoV-2 antigen-specific dCODE Dextramers (10x compatible) (Immudex, see Supplementary Table 1k for full panel). This panel includes five antigen-specific T-cell populations, spanning four MHC class I and one MHC class II alleles (covering a total of 15 participants; see Supplementary Table 1l) and several negative controls. Samples were then stained with several FACS antibodies (for monocyte and T cells) and sorted using a MACSQuant Tyto cell sorter (Miltenyi Biotec), after which PE-dCODE Dextramer-positive cells were collected and processed for 10x 5′ single-cell capture. This enabled the quantification of paired clonal TCR sequence and TCR specificity by overlaying single-cell V(D)J expression onto dCODE Dextramer-positive cell clusters.

The Dextramer staining protocol was taken from Immudex and optimized and adapted to suit our samples and pooling and staining strategy. In brief, the PBMC samples were thawed in batches of 7–8 samples and the cell number and viability for each sample calculated using Trypan Blue as described above. All cells from each sample were then pooled together in a fresh 1.5 ml Eppendorf tube. Note that the pooling strategy here was such that only one sample per participant or donor was used per pool to enable subsequent demultiplexing by genotype, with each pool containing a mixture of time points to help

reduce batch effect. To ensure the collection of as many cells as possible, each of the original sample tubes was then washed with 200 µl staining buffer (1× PBS pH 7.4 containing 5% heat-inactivated FBS (Thermo Fisher Scientific, 10500064) and 0.1 g l⁻¹ herring sperm DNA (Thermo Fisher Scientific, 15634017)) and added to the pool. The tube was then topped up to 1.4 ml with staining buffer and centrifuged down to a pellet (400$g$ for 5 min at 4 °C). The supernatant was carefully removed and the cell pellet gently resuspended in a total of 30–40 µl staining buffer, depending on pellet size, ready for staining.

In parallel, the dCODE Dextramer master mix was prepared (in the dark) as per the manufacturer's protocol. To help avoid aggregates, each individual Dextramer reagent was first microcentrifuged at full speed for 5 min before adding 2 µl from each dCODE Dextramer specificity to a low-bind nucleus-free 1.5 ml Eppendorf tube (Eppendorf, 30108051) containing 8.8 µl 100 µM D-Biotin (Avidity Science, BIO200) (0.2 µl D-Biotin per number of dCODE Dextramer specificity i.e., 44). The dCODE Dextramer master mix was mixed by gently pipetting before the total volume (96.8 µl) was added to the resuspended cells. The sample was then thoroughly mixed and incubated at room temperature for 30 min in the dark. Following the addition of anti-human CD14-FITC (BioLegend, 325603) and CD3-APC (BioLegend, 300458) (at 1:50) the cells were incubated for a further 20 min (at room temperature in the dark) before being topped up to 1.4 ml with wash buffer (1× PBS pH 7.4 containing 5% heat-inactivated FBS). The cells were centrifuged down to a pellet (400$g$ for 5 min at 4 °C) and the supernatant discarded. The wash step was then repeated 2 times, with the latter using the addition of 1.4 ml wash buffer and 1:5,000 DAPI (Sigma) as live/dead stain. The supernatant was removed and the cell pellet resuspended in 4 ml FACS buffer (1× PBS, 1% FBS, 25 mM HEPES (Thermo Fisher Scientific, 15630-056) and 1 mM EDTA). The samples were then filtered (35 µm nylon mesh cell strainer) and PE dCODE Dextramer-positive cells were sorted using a MACSQuant Tyto cell sorter per the manufacturer's guidelines (settings: mix speed = 800 r.p.m., chamber temperature = 4 °C, pressure = 150 hPA, noise threshold = 14.40, trigger threshold = off). Note, in order to collect as many cells as possible during sorting, the entire sample was run on the MACSQuant Tyto, with the negative run through collected and re-run a second time to ensure that no true positives were lost. See Extended Data Fig. 8d for the gating strategy for sorting. The PE dCODE Dextramer-positive cells were then collected, centrifuged (400$g$ for 5 min at 4 °C) and resuspended in resuspension medium before counting the cells. The entire sample was then processed for 10x 5′ single cell capture (Chromium Next GEM Single Cell V(D)J Reagent kit v.1.1 with Feature Barcoding technology for cell Surface Protein-Rev D protocol). For cases when more than 25,000 cells were collected, the sample was split equally and loaded over two lanes.

To provide additional controls, participants with non-compatible HLA types, including one volunteer (participant_4) matching none of the HLA types for the multi-allele dCODE Dextramer panel, were also processed and used to determine background noise.

## Library generation and sequencing

A Chromium Next GEM Single Cell 5′ V(D)J Reagent kit (v.1.1 chemistry) was used for scRNA-seq library construction for all nasopharyngeal swab samples, and a Chromium Next GEM Single Cell V(D)J Reagent kit v.1.1 with Feature Barcoding technology for cell surface proteins was used for PBMCs, both to process the PBMCs stained with the CITE-seq antibody panel and the dCODE Dextramer (10x compatible) panel. GEX and V(D)J libraries were prepared according to the manufacturer's protocol (10x Genomics) using individual Chromium i7 sample indices. Additional TCR γ/δ enriched libraries were generated based on an in-house protocol as previously described[37]. The cell surface protein libraries were created according to the manufacturer's protocol with slight modifications used for the creation of libraries generated from the CITE-seq antibody panel. These included doubling the SI primer amount per reaction and reducing the number of amplification cycles

to 7 during the index PCR to avoid the daisy chain effect. GEX, V(D)J and the CITE-seq-derived cell surface protein indexed libraries were pooled at a ratio of 1:0.1:0.4 and sequenced on a NovaSeq 6000 S4 Flowcell (paired-end, 150 bp reads), aiming for a minimum of 50,000 paired-end reads per cell for GEX libraries and 5,000 paired-end reads per cell for V(D)J and cell surface protein libraries. The Dextramer-derived cell surface protein indexed libraries were submitted at a ratio of 0.1.

### Single-cell genomics data alignment

scRNA-seq and CITE-seq data from PBMCs were jointly aligned against the GRCh38 reference that 10x Genomics provided with CellRanger (v.3.0.0), and alignment was performed using CellRanger (v.4.0.0). CITE-seq antibody-derived tag (ADT) barcodes were aligned against a barcode reference provided by the supplier, which we annotated to add informative protein names and made available in our GitHub repository (https://github.com/Teichlab/COVID-19_Challenge_Study). scRNA-seq data from nasopharyngeal swab samples were aligned against the same reference using STARSolo (v.2.7.3a) and post-processed with an implementation of emptydrops extracted from CellRanger (v.3.0.2). To detect viral RNA in infected cells, we added 21 viral genomes including pre-Alpha SARS-CoV-2 (NC_045512.2) to the abovementioned reference genomes for RNA-seq alignment, as previously described[6]. Single-cell αβ TCR and BCR data were aligned using CellRanger (v.4.0.0) with the accompanying GRCh38 V(D)J reference that 10x Genomics provided. Single-cell γδ TCR data were aligned against the GRCh38 reference that 10x Genomics provided with CellRanger (v.5.0.0), using CellRanger (v.6.1.2).

### Single-cell genomics data processing

Both scRNA-seq and ADT-seq data were corrected using SoupX[38] to remove free-floating and background RNAs and ADTs. To correct ADT counts, SoupX 1.5.2 parameters soupQuantile and tfidfMin parameters were set to 0.25 and 0.2, respectively, and lowered by decrements of 0.05 until the contamination fraction was calculated using the autoEstCont function. SoupX on RNA data was performed using default settings. To confidently annotate SARS-CoV-2-infected cells, we used SoupX-corrected viral RNA counts to remove false positives due to freely floating SARS-CoV-2 virions. However, when quantifying the amount of reads per cell in Fig. 2h and their distribution over the viral genome in Fig. 2f, we used the raw counts and sequencing data. To profile the distribution of viral reads, we removed PCR duplicates from the aligned BAM files that STARSolo produced with MarkDuplicates in picard (https://broadinstitute.github.io/picard/) and tallied the location within the SARS-CoV-2 genome using the start of each sequencing read. Aligned scRNA-seq data were imported from the filtered_feature_bc_matrix folder into Seurat (v.4.1.0) for processing, keeping only cells with at least 200 RNA features detected. Nasopharyngeal cells and PBMCs with more than 50% and 10% of the counts coming from mitochondrial genes, respectively, were excluded. SoupX-corrected gene expression and ADT counts were normalized by dividing it by the total counts per cell and multiplying by 10 000, followed by adding one and a natural-log transformation ($\log(1p)$).

### Demultiplexing and patient identity assignment

Each PBMC sample was pooled twice into two distinct pools containing up to four PBMC samples per pool, followed by CITE-seq and single-cell V(D)J sequencing as described above. Souporcell (v.2.0)[39] was used to demultiplex each pool based on the genotype differences between the mixed samples. Souporcell analyses were performed with the skip_remap parameter enabled and using the common SNP database that was provided by the software. We used two complementary approaches to confidently assign participant identity to each Souporcell cluster. First we compared the cluster genotypes with SNP array derived genotyping data, generated for all participants and performed using the Affymetrix UK Biobank Axiom Array kit by Cambridge Genomic Services.

Second, the combinations of samples within each pool was unique, which enabled assignment of participant identity based on the presence of unique participant-specific combinations of identical genotypes in two separate pools. This multiplexing and replication strategy furthermore enabled us to distinguish library specific batch effects from participant specific effects in downstream analyses.

### Doublet detection

We used the output from Souporcell to identify ground-truth doublets in PBMCs by selecting droplets that contained two genotypes from different participants. We then included these ground-truth doublets into the iterative rounds of subclustering and cell-state annotation to look for doublet specific clusters that emerged, which we then subsequently removed. Doublets in the nasopharyngeal data were removed during iterative rounds of subclustering and cell-state annotation by identifying cell clusters that expressed marker genes from multiple distinct cell types.

### Clustering and cell-type annotation

Principal component analysis was run on corrected gene expression counts from selected hypervariable genes, and the first 30 principal components were selected to construct a nearest neighbour graph and UMAP embedding. We used harmony[40] to perform batch correction on the PBMC data on the sequencing library identity to remove technical batch effects. Leiden clustering[41] performed at resolutions of 0.5, 1, 4 and 32 on nearest neighbour graphs and embeddings created with 500, 1,000, 2,000, 4,000, 6,000 and 8,000 selected hypervariable genes (excluding TCR and BCR genes) were used to perform iterative rounds of cell-type annotation based on marker gene expression and subsetting of clusters to obtain a highly granular cell state annotation. We used previously described cell-type marker genes[5,6] to define cell types. Our cell-type annotation was furthermore guided by predicted cell-type labels using models provided in CellTypist[42] and custom-trained models based on previously described annotations[5,6].

### Single-cell TCR and BCR data processing

Aligned single-cell BCR and αβ TCR sequencing data were imported in scirpy[43] to obtain a cell by TCR or BCR formatted table, which was then added to Seurat objects containing gene expression data. Aligned single-cell γδ TCR data were reannotated using Dandelion (v.0.2.4)[44].

### Differential gene expression and gene ontology analysis

We used DESeq2 (ref. 45) to identify significantly changing genes and gene sets. Samples were pseudobulked on cell state and sample, and we used a Wald test to compute adjusted $P$ values. To identify genes associated with infection outcome at day −1, we fitted gene expression from pre-infection samples on cell type, sex and infection outcome. We also included sequencing library identity as a covariate in the differential expression analyses on PBMCs. To quantify interferon stimulation, we used a previously published gene signature[6], and we used the 'AddModuleScore' function from Seurat to quantify its expression per cell. Cells were classified as interferon stimulated if the module score was higher than 0.5, and significance was determined by a Mann–Whitney $U$-test on module scores, which was corrected for the multiple testing hypothesis using the Bonferroni approach.

### Integration of five COVID-19 studies

Transcriptomic data from refs. 5,6,31–33 were processed using the single-cell analysis Python workflow Scanpy[46]. Each dataset was individually filtered following best practices outlined in ref. 47 (between 200 and 3,500 genes per cell, less than 10% mitochondrial genes expressed per cell, genes expressed in fewer than 3 cells, other parameters at default). The gene sets were reduced to their intersection before combining datasets. Cells came from a total of 602 individuals, with 325 patients with acute COVID-19, 110 patients convalescing from COVID-19,

114 healthy participants and 53 patients in hospital without COVID-19 (controls) (Supplementary Table 1d). This resulted in an integrated embedding containing 946,584 T cells with resolved TCR from 494 samples, made up of 455 donors of which 240 were patients with acute COVID-19, 82 were patients convalescing from COVID-19, 88 healthy participants and 45 patients in hospital without COVID-19 (Supplementary Table 1e). The total number of donors in the integrated object is smaller, as only samples with matching V(D)J sequencing data were kept. A probabilistic scVI model (2 hidden layers, 128 hidden nodes, 20-dimensional latent space, negative binomial gene likelihood, other parameters at default[48]) was trained on the data to map cells to a shared latent space and visualized using UMAP.

### Identification of activated TCR clonotype groups using Cell2TCR
To identify TCR clonotype groups, we used tcrdist3 (ref. 49) with the provided human references to compute a sparse representation of the distance matrices for all identified TRA and TRB CDR3 sequences, with the radius parameter set to 150. We then summed the distances for TRA and TRB to obtain a combined distance matrix. Next, we iterated over possible TCR distance thresholds between 5 and 150 with increments of 5 to compute TCR clonotype groups at each threshold. We then generated a distance adjacency graph of TCRs from different T cells with a distance lower than the threshold, which was clustered to identify TCR clonotype groups using leiden[41] clustering through the igraph package[50], at a resolution of 1 and using the RBConfigurationVertexPartition partition. To find the optimal distance threshold at which only TCRs that recognize the same antigen are grouped together, we quantified clonotype group contamination at each threshold using two approaches. First, we assumed that T cells that were annotated as naive should not participate in an expanded clonotype group, and quantified the proportion of naive T cells in each clonotype group to determine the largest threshold at which we observed minimal participation of naive T cells. Second, we assumed that CD4+ T cells and CD8+ T cells should never be part of the same TCR clonotype group, so we set out to quantify the proportion of CD4+ and CD8+ mixing in each clonotype group to find the largest threshold at which mixing is minimal. Both approaches revealed the same optimal threshold of 35, at which both naive T cell participation and CD4+ and CD8+ mixing is minimal, which we then used for downstream analyses. To identify activated TCR clonotype groups, we assumed that these groups should include activated T cells and that we should at least detect multiple independent TCR clonotypes that seemed to be raised against the same antigen at the same time. We therefore selected clonotype groups that contained at least one participating activated T cell and that contained at least two unique CDR3 nucleotide sequences.

### Identification of activated BCR clonotype groups
To identify BCR clonotype groups that were activated during infection, we used a similar approach as described above for T cells. Instead of using tcrdist to compute distances, we used the Levenshtein distance and iterated over possible thresholds between 1 and 20 to find an optimal threshold by quantifying naive B cell participation. This revealed that a Levenshtein distance of 2 is optimal to identify BCR clonotype groups that only contain B cells that recognize the same antigen. To identify activated BCR clonotype groups, we assumed that these groups should include antibody secreting B cells (plasmablasts and plasma cells) and that we should at least detect multiple independent BCRs clonotypes that seem to be raised against the same antigen at the same time. We therefore selected clonotype groups that contained at least one participating antibody secreting B cell and that contained at least three unique CDR3 nucleotide sequences.

### Generation of V(D)J logos
TCR and BCR logos were generated by providing the CDR3 amino acid sequences of each clonotype group to the ggseqlogo R package[51] or the logomaker Python package[52]. When clonotype groups contained CDR3 amino acid sequences of variable lengths, we selected the sequences with the most frequently occurring length within each group for visualization purposes only.

### GLMMs of cell-state compositional changes over time
The relative abundance of cells per cell type in each sample was modelled using a GLMM with a Poisson outcome. When technical replicates were available (most of the PBMC samples), these were modelled as separate samples. We modelled participant identifiers, days since inoculation and sequencing library identifiers (of multiplexed libraries), as random effects to overcome collinearity between these factors. The effect of each clinical or technical factor on cell-type composition was estimated by the interaction term with the cell type. The glmer function in the lme4 package implemented on R was used to fit the model. The standard error of the variance parameter for each factor was estimated using the numDeriv package. The conditional distribution of the fold change estimate of a level of each factor was obtained using the ranef function in the lme4 package. The log-transformed fold change is relative to the pre-inoculation time point (day −1). The significance of the fold change estimate was measured by the local true sign rate, which is the probability that the estimated direction of the effect is true, that is, the probability that the true log-transformed fold change is greater than 0 if the estimated mean is positive (or less than 0 if the estimated mean is negative). We calculated $P$ values using a two-sample $Z$-test using the estimated mean and standard deviation of the distribution of the effect (log-transformed fold change). $P$ values were converted into FDRs using the Benjamini–Hochberg method.

### Gaussian processes regression and latent variable models to infer time since viral exposure
To infer time from cell-state abundance, we first generated a logistic regression model using CellTypist[42] to predict PBMC or nasopharyngeal cell states based on the highly detailed manually annotates cell states presented in this work. CellTypist models were trained and used under default parameters, with check_expression set to false, balance_cell_type set to true, feature_selection set to true, and max_iter set to 150. We next built a predictive model to infer time since viral exposure using the PBMC data presented in this work as a training dataset. We used the above mentioned publicly available PBMC data from five studies as a test dataset to predict time since viral exposure. Because we were specifically interested in comparing time since viral exposure to reported time since onset of symptoms in varying disease severities, we excluded samples for which these features were unknown. To ensure that the cell-state proportions in the training and test dataset were similar, we used our CellTypist model on both datasets to predict relative cell-state frequencies, which were used as input for our time prediction model. To account for participant-to-participant heterogeneity and continuous variation in the timeline of immune responses, we first constructed a Gaussian process latent variable model[53] to smooth the time since viral exposure in the training dataset. We applied the Pyro implementation of this model[54] across all predicted cell state abundances, and restricted the model to 2,000 iterations and a single latent variable that was initialized on the square root transformed time since inoculation. This resulted in an accurate recapitulation of the mean time since inoculation while smoothing outliers. We next used each predicted cell state as a task input to generate a multi-task Gaussian process regression model[55] to predict the smoothened time since inoculation using GPyTorch[56]. We used the Adam optimizer and allowed for as many iterations for the loss in marginal log likelihood to reach zero. We next predicted the cell state compositions across the entire tested timeline (day −1 to day 28) and compared these cell state compositions to those in our query dataset as predicted by our CellTypist model. Last, we selected the time point at which predicted cell-state

composition had the lowest mean squared error compared with the observed cell-state composition.

## Matching clonotype groups to antigen–TCR database

We computed the fold change enrichment of SARS-CoV-2-specific TCRs in activated T cell populations compared with other T cell populations. After 10 random draws of $n = 5,000$ unique clones of both populations, the median fold change = 4.99, median $P = 0.00044$.

## Bulk TCR sequencing and processing

Total RNA was extracted from whole blood samples collected in Tempus Blood RNA tubes (Thermo Fisher, 4342792) using the manufacturer's protocol. TCR α and β genes were sequenced using a pipeline that introduces UMIs attached to individual cDNA molecules using single-stranded DNA ligation. The UMI enables correction for sequencing error PCR bias, and provides a quantitative and reproducible method of repertoire analysis. Full details for both the experimental TCR sequencing library preparation[57,58] and the subsequent TCR annotation (V, J and CDR3 annotation) using Decombinator (v.4)[59] are published. The Decombinator software is freely available at GitHub (https://github.com/innate2adaptive/Decombinator).

## Memory formation analysis

T cell phenotypes (naive, activated, effector and memory) were recorded for an antigen-specific TCR clone at different time points throughout infection. TCR clones were filtered by having an activated label at least once, being observed in at least two samples, one of which had to be at day 28. Unique TCR clones are distinguished by colour and numbered with their clone_id identifier. Error bands are drawn when the same clone appeared with several distinct cell-type labels, and the size of the error band informs their relative ratios.

## Quantifying TCR diversity restriction in phenotypic clusters using coincidence analysis

To quantify the diversity of TCRs found within different phenotypic clusters, we determined the probability with which two distinct clonotypes within a cluster share an identical CDR3 amino acid sequence[60]. For visualization, we normalized these probabilities by the same quantity calculated over the complete data regardless of phenotype. This ratio of probability of coincidences provides a stringent measure of convergent functional selection of distinct clonotypes that share the same TCR. The analysis is based on clonotypes defined by distinct nucleotide sequences of the hypervariable regions, and does not make direct use of clonal abundance as these can also reflect TCR-independent lineage differences. We focused our analysis on conventional T cells only, considered only cells with at least one valid functional α-chain and β-chain, and kept only a single chain for each cell in which there were multiple chains. We performed the analysis both on the α-chain and β-chain separately, as well as on paired α and β-chains, in each instance requiring exact matching of the CDR3 amino acid sequences.

## Modelling infection outcome on *HLA-DQA2* expression

To test whether cell-type-specific expression of *HLA-DQA2* at the day before inoculation was predictive of the infection outcome of the challenge experiment, we performed logistic regression modelling using the 'glm' R package. For each cell type shown, we fitted whether or not a sustained infection would occur on the mean expression and fraction of cells expressing *HLA-DQA2* at day −1. For cross-validation, we used the 'roc' R package and performed five 1:1 test-train splits.

## Multi-flow re-analysis

Samples used to assess MAIT cell activation were collected as part of the prospective healthcare worker study Covidsortium. Participant screening, study design, sample collection and sample processing have previously been described in detail[61]. Participants with available

PBMC samples who had PCR-confirmed SARS-CoV-2 infection (Roche cobas diagnostic test platform) at any time point were included as cases. A subset of consecutively recruited participants without evidence of SARS-CoV-2 infection on nasopharyngeal swabs and who remained seronegative by both Euroimmun antiS1 spike protein and Roche anti-nucleocapsid protein throughout follow-up (16 weeks of weekly PCR and serology) were included as uninfected controls. The study was approved by a UK Research Ethics Committee (South Central–Oxford A Research Ethics Committee, ref. 20/SC/0149). All participants provided written informed consent.

Multiparametric flow cytometry was performed as described previously and data related to immune subsets other than MAIT cells were previously published[14]. PBMCs were plated in 96-well round-bottomed plates ($0.5-1 \times 10^6$ per sample) and washed once in PBS (PBS; Thermo Fisher) then stained with Blue fixable live/dead dye (Thermo Fisher) for 20 min at 4 °C in PBS. Cells were washed again in PBS and incubated with saturating concentrations of monoclonal antibodies against markers to be stained on the cell surface, diluted in 50% Brilliant violet buffer (BD Biosciences) and 50% PBS for 30 min at 4 °C. After surface antibody staining, cells were resuspended in fix/perm buffer (eBiosciences, Foxp3/Transcription Factor staining buffer kit, fix perm concentrate diluted 1:3 in fix/perm diluent) for 45–60 min at 4 °C. Cells were then washed in 1× perm buffer (10× perm buffer Foxp3/Transcription Factor staining buffer kit diluted to 1× in ddH₂O) and saturating concentrations of intranuclear targets (Ki67) were stained in 1× perm buffer for 30–45 min, 4 °C. Cells were washed twice in PBS then analysed by flow cytometry using a LSR II flow cytometer (BD Biosciences). Flow cytometry data were analysed using FlowJo (v.10.7.1 for mac, Tree Star). Single stain controls were prepared with cells or anti-mouse IgG beads (BD Biosciences). Fluorescence minus one controls (FMOs) were used for gating (see ref. 14 for FMOs and detailed gating related to these stains). Note that the frequency of MAIT cells did not differ between controls or PCR⁺ as previously reported[14].

## Immunofluorescence confocal microscopy

As previously described[62], SARS-CoV-2 and mock-infected human nasal epithelial cultures grown at an air–liquid interface were fixed using 4% (v/v) paraformaldehyde for 30 min, permeabilized with 0.2% Triton-X (Sigma) for 15 min and blocked with 5% goat serum (Sigma) for 1 h before overnight staining with primary antibody at 4 °C. Secondary antibody incubations were performed the next day for 1 h at room temperature. Cultures were then incubated with AlexaFluor 555 phalloidin and DAPI (Sigma) for 15 min before mounting with Prolong Gold Antifade reagent (Life Tech). Samples were washed with PBS-T after each incubation step. Images were captured using a LSM710 Zeiss confocal microscope and rendered using Nikon NIS Elements. Human nasal epithelial cell cultures from three individual donors (one child <12 years old, one adult 30–50 years old and one adult >70 years old) were stained and 4 technical repeats used per donor (mock and SARS-CoV-2 infection conditions). Representative images of immunofluorescence staining, taken 72 h after infection, of nasal epithelial cell cultures from the older adult and the child can be seen in Extended Data Fig. 5b and Extended Data Fig. 9a, respectively.

## Transmission electron microscopy

Cultured human nasal epithelial cells that were either SARS-CoV-2-infected or mock-infected were fixed with 4% paraformaldehyde 2.5% glutaraldehyde in 0.05 M sodium cacodylate buffer at pH 7.4 and placed at 4 °C for at least 24 h, as previously described[62,63]. The samples were incubated in 1% aqueous osmium tetroxide for 1 h at room temperature before subsequently en bloc staining in undiluted UA-Zero (Agar Scientific) for 30 min at room temperature. The samples were dehydrated using increasing concentrations of ethanol (50, 70, 90 and 100%), followed by propylene oxide and a mixture of propylene oxide and araldite resin (1:1). The samples were embedded in araldite and

left at 60 °C for 48 h. Ultrathin sections were acquired using a Reichert Ultracut E ultramicrotome and stained using Reynold's lead citrate for 10 min at room temperature. Images were taken on a JEOL 1400Plus transmission electron microscope equipped with an Advanced Microscopy Technologies (AMT) XR16 charge-coupled device camera and using the software AMT Capture Engine. Human nasal epithelial cell cultures from three individual donors (one child <12 years old, one adult 30–50 years old and one adult >70 years old) at 72 h after infection (mock and SARS-CoV-2 infected) were processed and imaged. Representative images 72 h after infection from SAR-CoV-2-infected nasal epithelial cell cultures from the older adult (>70 years) are shown in (Fig. 2), with additional images from the child (<12 years), younger adult (30–50 years) and older adult (>70 years) can be seen in Extended Data Fig. 9b.

### Serum antibody assays

As previously described[7], serum samples from each participant were taken and the antibody titre measured using two assays. In brief, the SARS-CoV-2 anti-spike IgG concentrations were determined by ELISA (using Nexelis) and reported as ELU ml$^{-1}$ (Supplementary Table 1p). Neutralizing antibody titres for live SARS-CoV-2 virus (lineage Victoria/01/2020) were determined by microneutralization assay at the UK Health Security Agency and reported as the 50% neutralizing antibody titre ($NT_{50}$). The LLOQ was 58 and 50.2 ELU ml$^{-1}$, respectively, for the microneutralization assay and the spike protein IgG ELISA. For the median (IQR) per infection group, see the summary study metadata table in Supplementary Table 1g.

### Reporting summary

Further information on research design is available in the Nature Portfolio Reporting Summary linked to this article.

### Data availability

The data presented in this study can be explored and analysed interactively through our COVID-19 Cell Atlas web portal (https://covid19cellatlas.org). The cell-state annotation model is available at the CellTypist model repository (https://www.celltypist.org/models) under the name 'COVID19_HumanChallenge_Blood'. A reference for our multi-task Gaussian process regression model to infer time since viral exposure on PBMC data is available at our GitHub repository (https://github.com/Teichlab/COVID-19_Challenge_Study). The raw sequencing data are available under controlled access at the European Genome-Phenome Archive under accession number EGAD00001012227. Processed bulk RNA-seq data are available at ArrayExpress (accession number E-MTAB-12993). Single-cell count matrices with metadata are available at the COVID-19 Cell Atlas web portal as h5ad files.

### Code availability

Cell2TCR is available at our GitHub repository (https://github.com/Teichlab/Cell2TCR). Code that was used for data analysis is available at our GitHub repository (https://github.com/Teichlab/COVID-19_Challenge_Study) and marked with the 'Release for Nature publication' version.

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

**Acknowledgements** We are grateful to S. Casserlya and S. Regmi for assistance with collecting samples; T. Porter, A. Oszlanczi, Y. Wood and S. Eckert for library preparation; staff at 10x Genomics and Illumina, R. Dabelić, J. Betley and D. Bentley for assistance and support; Y. Guo and staff at the Flow Cytometry Translational Technology Platform Manager at UCL Cancer Institute for support and guidance with MACS for the Dextramer assays; staff at the University College London CL3 facility at the Paul O'Gorman building and staff at the Sanger Institute Core Sequencing facility for their assistance; and A. L. D. N. Pinto and T. Burgoyne for their expertise and support in acquiring the images using transmission electron microscopy. This research was funded in whole, or in part, by Wellcome Trust grants 206194, 220540/Z/20/A, 211276/Z/18/Z and 224530/Z/21/Z (to M.Z.N., C.C. and S.A.T.), and by Action Medical Research (GN2911, to M.Z.N. and K.B.M.). M.Z.N. acknowledges funding from a MRC Clinician Scientist Fellowship (MR/W00111X/1) and from the Chan Zuckerberg Initiative. M.Z.N. and J.L.B. acknowledge funding from the Rutherford Fund Fellowship allocated by the MRC UK Regenerative Medicine Platform2 (MR/5005579/1). S.M.J. is supported by a CRUK programme grant (EDDCPGM\100002), and a MRC Programme grant (MR/W025051/1). S.M.J. receives support from the Longfonds BREATH Consortia, MRC UKRMP2 Consortia. This work was partly undertaken at UCLH/UCL who received a proportion of funding from the Department of Health's NIHR Biomedical Research Centre's funding scheme. R.C.C. reports grants from UK Research and Innovation/Medical Research Council (UKRI/MRC), Asthma and Lung UK, the Rosetrees Trust and the National Institute for Health and Care Research University College London Hospitals Biomedical Research Centre. M.Y. was funded by The Jikei University School of Medicine and Action Medical Research (GN2911). K.B.W. acknowledges funding from University College London, Birkbeck MRC Doctoral Training Programme. L.M.D. is supported by the European Union's Horizon 2020 research and innovation programme under the Marie Skłodowska-Curie grant agreement no. 955321. L.K. is supported by an EMBO Long-Term Fellowship (ALTF 120-2023). M.N. acknowledges funding from the Wellcome Trust (207511/Z/17/Z) and by NIHR Biomedical Research Funding to UCL and UCLH. R.H. is a NIHR Senior Investigator. M.N.J.W. and C.M.S. acknowledge funding by UKRI/ BBSRC (BB/V006738/1) and the NIHR Great Ormond Street Hospital Biomedical Research Centre. Research at the Netherlands Cancer Institute is supported by institutional grants of the Dutch Cancer Society and of the Dutch Ministry of Health, Welfare and Sport. J.M.B. was funded by the Wellcome

Sanger Institute through the Sanger Prize. The authors gratefully acknowledge support from the UK Vaccine Taskforce of the Department of Business, Energy and Industrial Strategy of Her Majesty's Government (BEIS). We thank the following organizations for their invaluable contributions to development and implementation of the SARS-CoV-2 human challenge project: the Royal Free London NHS Foundation Trust, Human Infection Challenge Network for Vaccine Development (HIC-Vac), NIHR Clinical Research Network staff at the Royal Bolton Hospital, and ISARIC4C Investigators (https://isaric4c.net/about/authors/) for providing the clinical material for challenge virus production. C.C. is supported by the NIHR Imperial Biomedical Research Centre (BRC) award to Imperial College Healthcare NHS Trust and Imperial College London. ISARIC4C is funded by the National Institute for Health Research (NIHR; award CO-CIN-01), the Medical Research Council (MRC; grant MC_PC_19059), the NIHR Health Protection Research Unit in Emerging and Zoonotic Infections at University of Liverpool in partnership with Public Health England (PHE), in collaboration with Liverpool School of Tropical Medicine and the University of Oxford (NIHR award 200907), Liverpool Experimental Cancer Medicine Centre provided infrastructure support for this research (grant reference C18616/A25153) and NIHR Health Protection Research Unit in Respiratory Infections (NIHR award 200927). The views expressed are those of the authors and not necessarily those of the NHS, the NIHR, DHSC or BEIS. This publication is part of the Human Cell Atlas (https://www.humancellatlas.org/publications). Illustrations in Fig. 1 and Extended Data Fig. 10 were created using BioRender (https://www.biorender.com). For the purpose of Open Access, the author has applied a CC BY public copyright license to any Author Accepted Manuscript version arising from this submission.

**Author contributions** M.Z.N. and S.A.T. conceived, set up, directed this study and provided funding. C.C. set up the clinical study and co-ordinated sampling. K.B.W. optimized digestion protocols, processed samples for 10x and CITE-seq, isolated DNA for genotyping, performed Dextramer experiments and assisted with data analyses and interpretation. R.G.H.L. performed and led the data analyses. L.M.D. assisted with data analyses and implemented Cell2TCR in Python. L.K., J.M.B., R.E., K.P., W.S., N.H. and D. Sumanaweera advised on and assisted with data analyses. R.G.H.L., K.B.W., L.M.D., K.B.M., M.Z.N. and S.A.T. interpreted the data and wrote the manuscript. L.R., A.W.-C., L.M., R.K., L.B. and E.P. performed the single-cell sequencing library preparations. M.Y., J.L.B. and J.A.-H. assisted with CITE-seq and 10x sample processing. V.H.T., S.M.J. and R.C.C. provided student supervision to K.B.W. and P.M. H.R.W. processed blood samples. D. Scobie, B.C. and A. Mayer provided bulk TCR-seq data and advised on the data analysis. P.M. collected nasopharyngeal samples for optimization of digestion protocols. M.H., M.N. and R.H. assisted in the set-up of the study. M.N. provided bulk RNA-seq data for validation. B.K., M.K., A.C., A.B., A. Mann and M.M. oversaw sample collection and provided clinical data. M.N.J.W., S.E. and C.M.S. provided electron microscopy and immunohistochemistry images. L.S. performed multi-flow re-analysis for validation.

**Competing interests** R.G.H.L., L.M.D., R.E. and S.A.T. are inventors on a filed patent that is related to the detection and application of activated T cells. In the past 3 years, S.A.T. has received remuneration for scientific advisory board membership from Sanofi, GlaxoSmithKline, Foresite Labs and Qiagen. S.A.T. is a co-founder and holds equity in Transition Bio and Ensocell. From 8 January 2024, S.A.T. is a part-time employee of GlaxoSmithKline. R.E. is a co-founder and equity holder in Ensocell. P.M. is a Medical Research Council (MRC)-GlaxoSmithKline EMINENT clinical training fellow with project funding unrelated to the topic of this work and receives co-funding from the National Institute for Health Research (NIHR) University College London Hospitals (UCLH) Biomedical Research Centre. P.M. reports consultancy fees from SOBI, AbbVie, UCB, Lilly, Boehringer Ingelheim and EUSA Pharma, all unrelated to this study. S.M.J. has received fees for advisory board membership in the last three years from Bard1 Lifescience. He has received grant income from GRAIL Inc. He is an unpaid member of a GRAIL advisory board. He has received lecture fees for academic meetings from Cheisi and AstraZeneca. His wife works for AstraZeneca. R.C.C. has research collaborations with Chiesi Chiesi Farmaceutici S.p.A. and GSK and receives consulting fees from Vicore, outside the submitted work. A. Mann, A.C., M.K., M.M. and A.B. are full-time employees at hVIVO Services. The other authors declare no competing interests.

**Additional information**
**Correspondence and requests for materials** should be addressed to Rik G. H. Lindeboom, Marko Z. Nikolić or Sarah A. Teichmann.

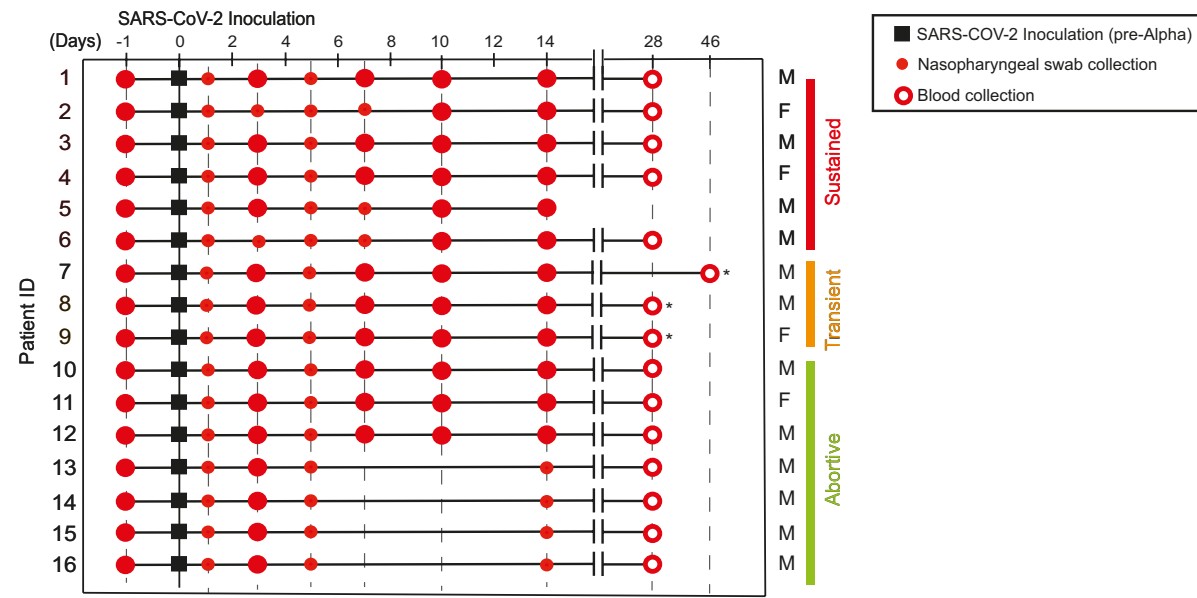

a

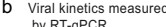

b Viral kinetics measured by RT-qPCR

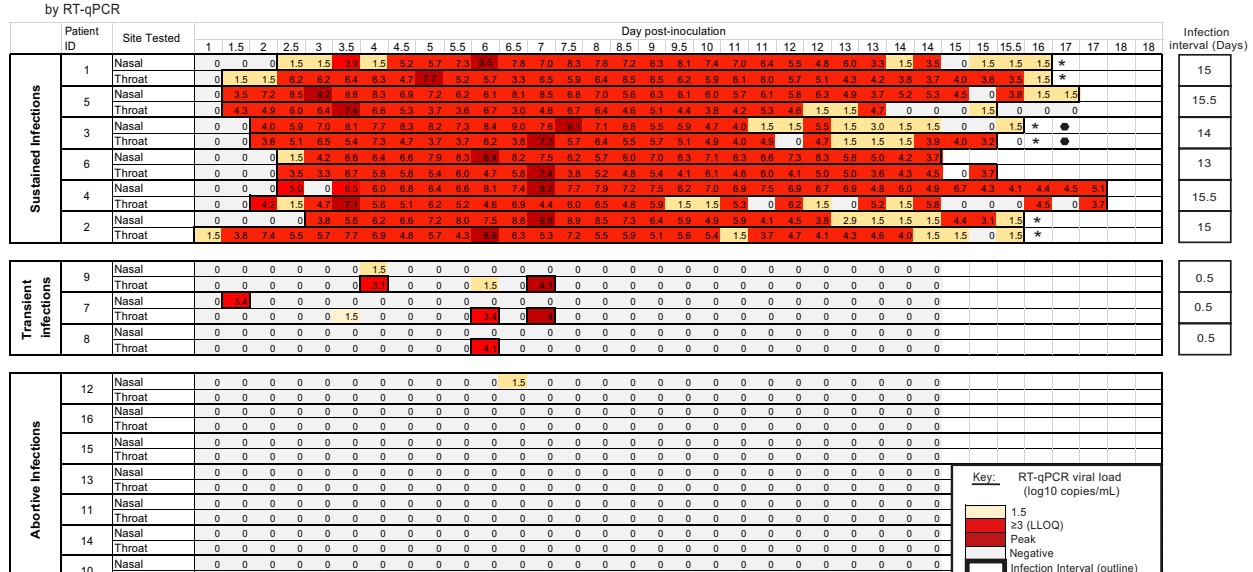

c Viral kinetics measured by FFA cultures

**Extended Data Fig. 1** | See next page for caption.

**Extended Data Fig. 1 | Overview of Single-Cell Human SARS-CoV-2 Challenge Study cohort.** (**a**) Timeline of the samples collected from each of the 16 participants enrolled in our single-cell profiling study. Sample collections are shown relative to the date of SARS-CoV-2 inoculation (day 0). Samples are shown by infection group (sustained, transient and abortive), with their sex (self-identified). *Indicates participants who were either vaccinated (participant 9) or reported to have developed a community infection, before or immediately after blood samples were taken on day 28 (participants 7 and 8). See 'Study participant and design' section in the methods for more details. Visualization of the nasal (mid-turbinate) and throat (pharyngeal) viral kinetics via swabs. Shown for each participant as measured (twice daily at 12 h intervals) via (**b**) RT-qPCR and (**c**) quantitative culture by focus forming assay (FFA), with values shown. The lower limit of quantification (LLOQ) for RT-qPCR was $3 \log_{10}$ copies per ml, with positive detections less than the LLOQ assigned a value of $1.5 \log_{10}$ copies per milliliter and undetectable samples assigned a value of $0 \log_{10}$ copies copies per milliliter. In the FFA the LLOQ was $1.27$ FFU ml$^{-1}$; viral detection less than the LLOQ was assigned $1 \log_{10}$ FFU ml$^{-1}$; and undetectable samples were assigned $0 \log_{10}$ FFU ml$^{-1}$. Patients were identified as testing positive if they had at least one RT-qPCR test where the viral load was able to be quantified ($\geq$LLOQ). Six participants were seen to present multiple, sequential, positive RT-qPCR results and were classified as having a sustained infection. Three participants were seen to have standalone positive results and were classified having transient infections. Seven participants never presented a single RT-qPCR test result $\geq$ LLOQ and these were classified as abortive infections. FFA tests were only performed for patients identified as having sustained infections. Infection intervals for each participant were calculated based on the first and last values across the nose and throat, where positive tests below the LLOQ were counted if they occurred <2 days of a quantifiable ($\geq$ LLOQ) test result. *Indicates where the patient was discharged from quarantine prior to testing negative. The black octagon highlights patients that were still reporting positive results at day 28 post-inoculation.

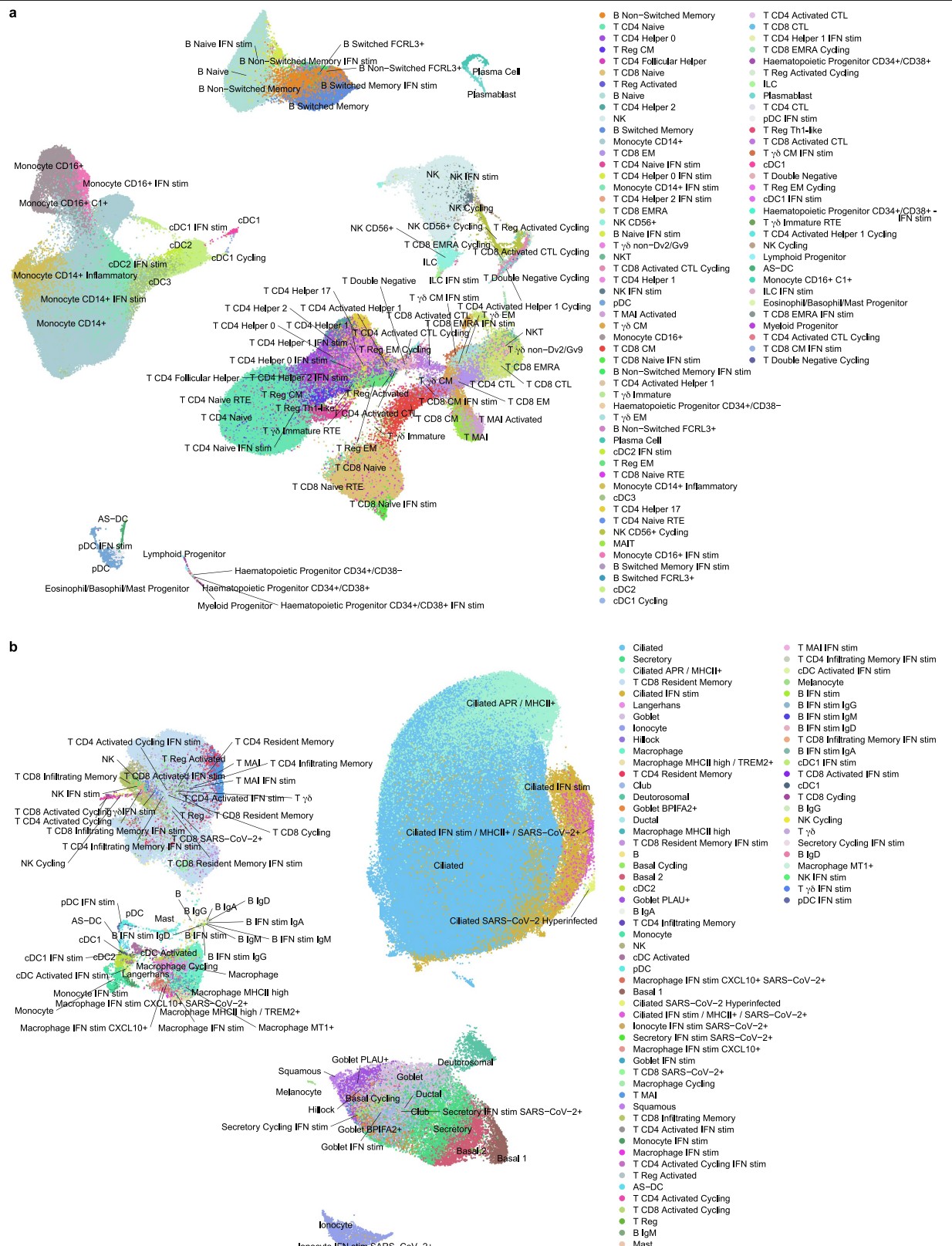

**Extended Data Fig. 2 | All identified and annotated cells. (a)** UMAP of all PBMCs, color-coded and labeled by detailed cell state annotation. Subsets of B cells with differential immunoglobulin chain usage are not shown in full detail for clarity. **(b)** UMAP of all nasopharyngeal cells, color-coded and labeled by detailed cell state annotation.

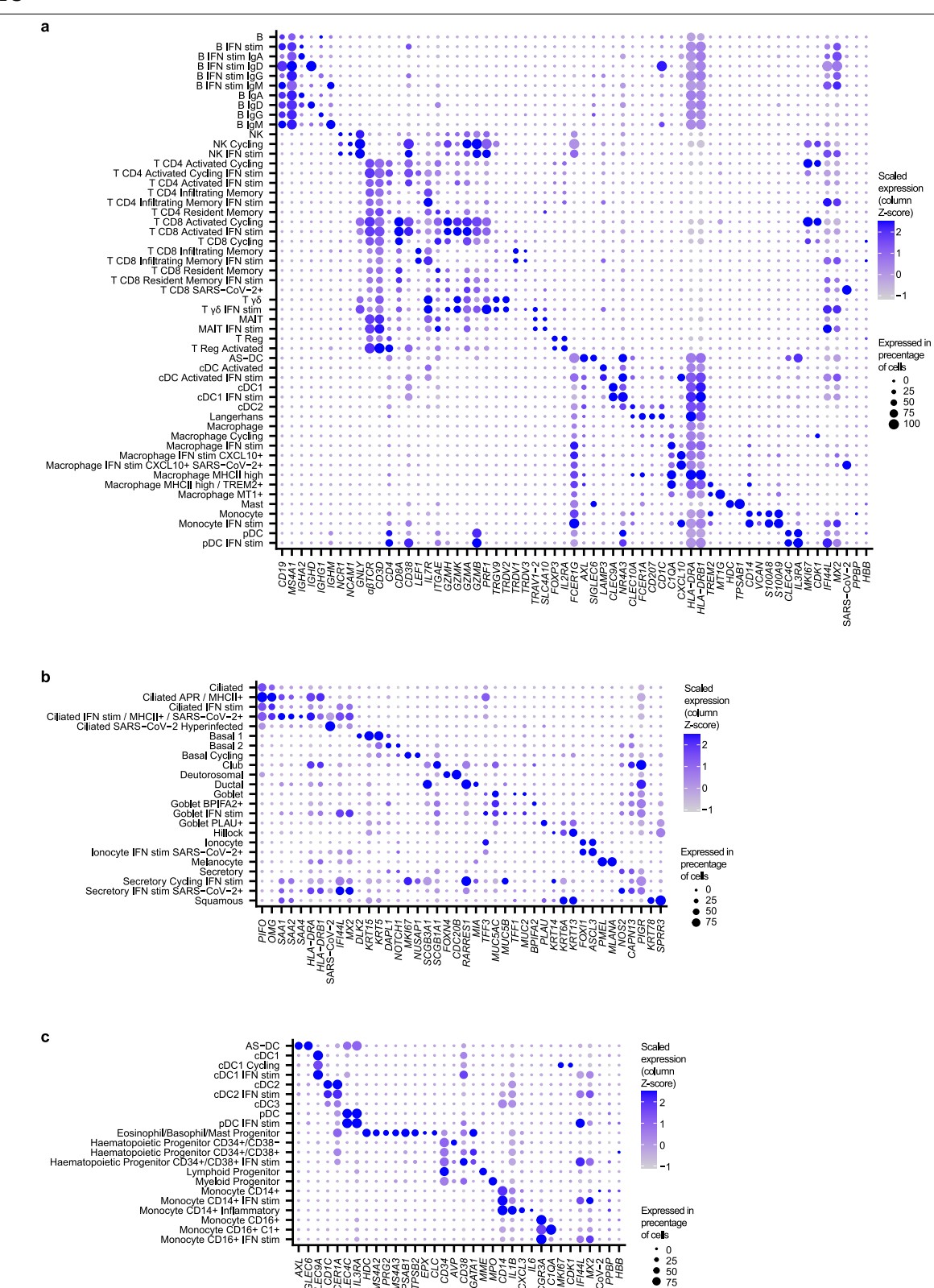

**Extended Data Fig. 3 | Marker gene expression used for annotation.** Marker gene expression of cell states annotated in (**a**) nasopharyngeal immune cells, (**b**) nasopharyngeal epithelial cells, (**c**) myeloid and progenitor PBMCs.

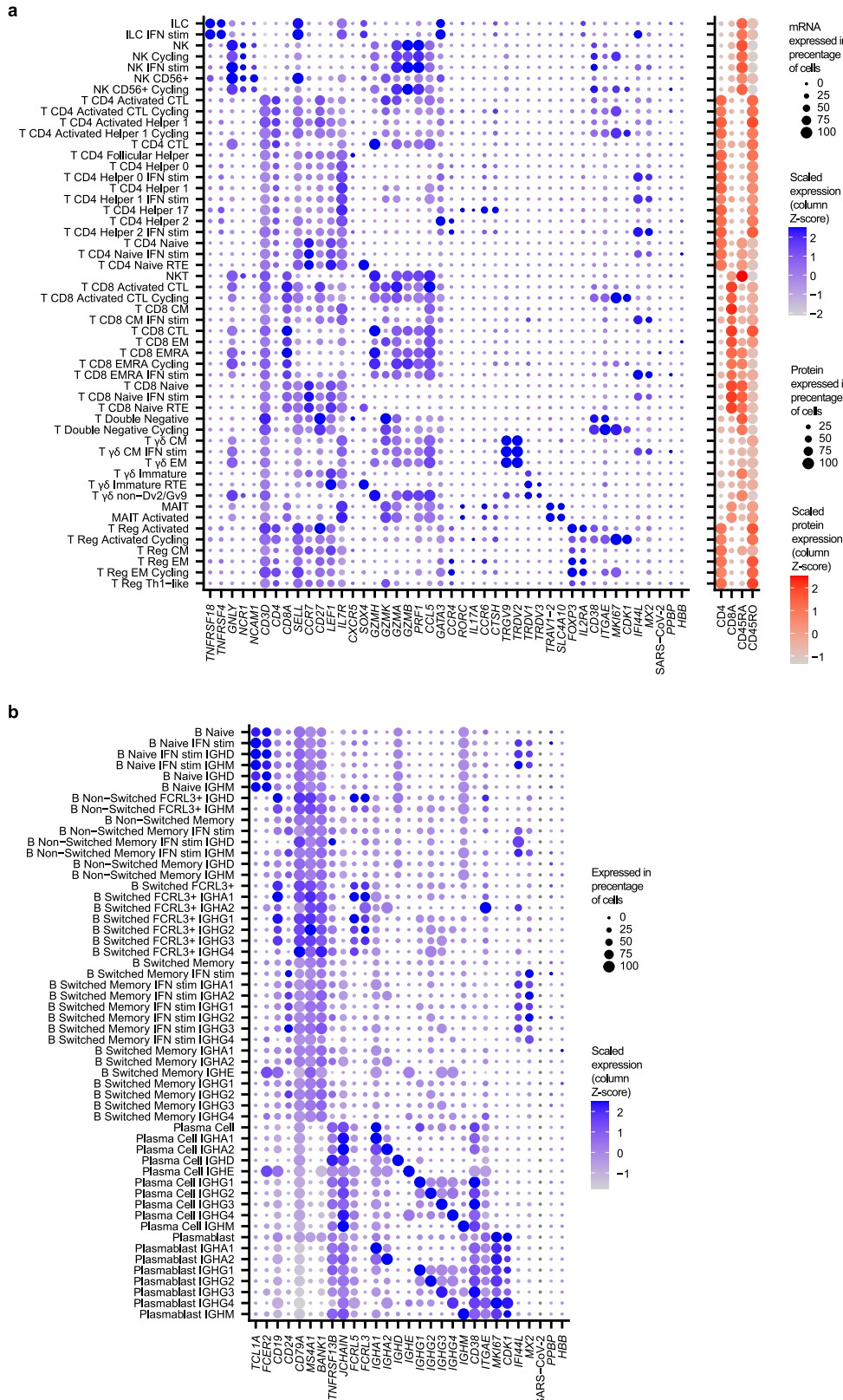

**Extended Data Fig. 4 | Marker gene expression used for annotation of PBMCs.** Marker gene expression of cell states annotated in (**a**) T, NK and ILC cells in PBMCs, (**b**) B cells in PBMCs.

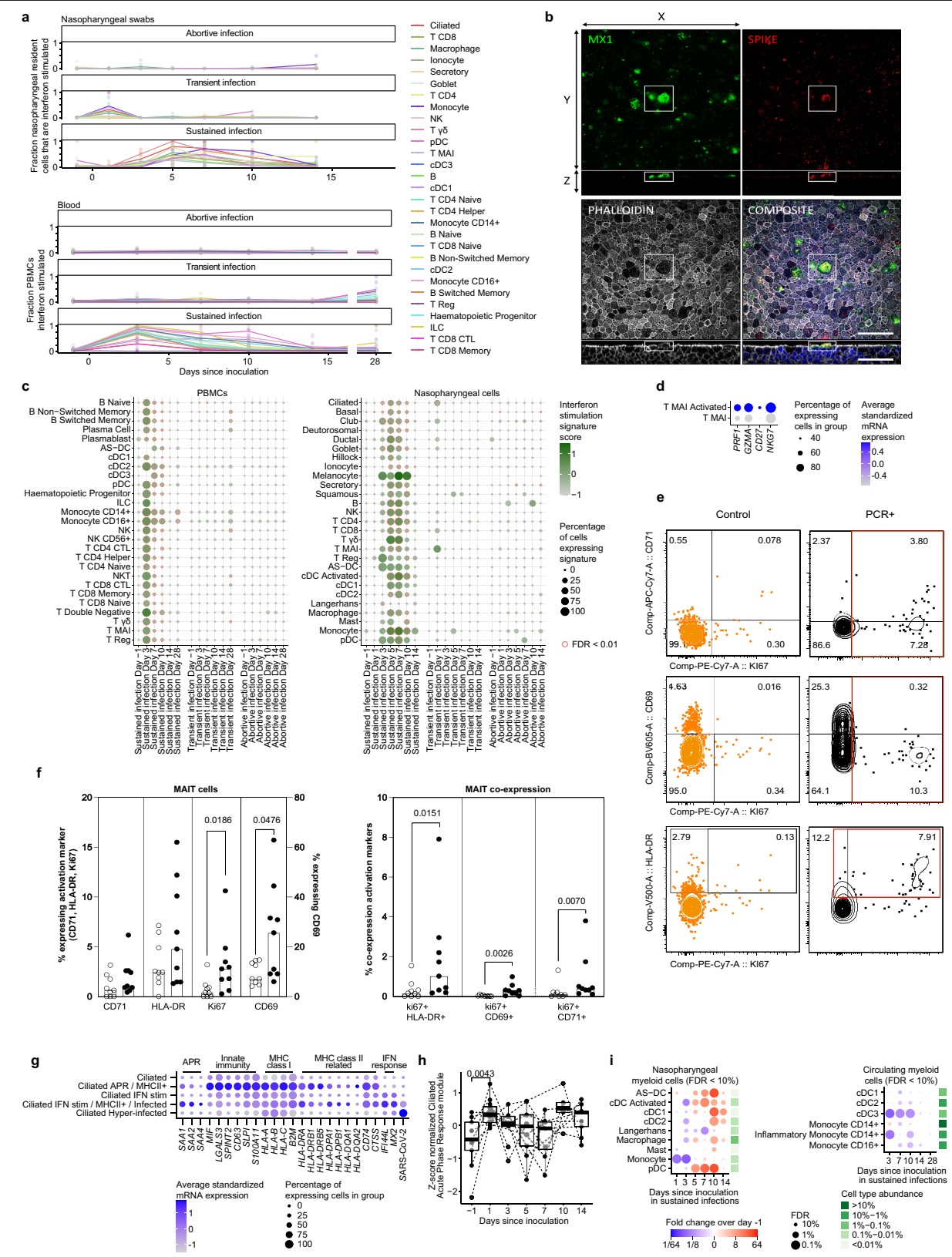

**Extended Data Fig. 5** | See next page for caption.

**Extended Data Fig. 5 | Temporal response states.** (**a**) Line plot showing the mean proportions of interferon stimulated cells over time since inoculation within cell types with a distinct and annotated cluster of interferon stimulated cells for nasopharyngeal cells (top) and PBMCs (bottom). (**b**) Representative Immunofluorescence confocal image of SARS-CoV-2 infected human nasal epithelial cultures grown at air-liquid interface at 72 h post infection. Image shown as a maximum intensity projection of the z-stack. Cells are stained with antibodies for MX1 (green), SARS-CoV-2 spike protein (red), phalloidin (white) and DAPI (blue). White box indicates an area of high colocalisation of MX1 and spike protein staining. Scale bar represents 50μm. (**c**) Dotplot visualizing the mean expression of interferon stimulated genes across cell types and time since inoculation for every participant, for PBMCs (left) and nasopharyngeal cells (right). Red circles indicate significance that was calculated with a Mann Whitney U test compared to the other time points, followed by Bonferroni correction. (**d**) Marker gene expression of activated MAIT cells. (**e**) Representative flow cytometry plots showing activation marker expression (Ki67, CD71, CD69 and HLA-DR) by mucosal associated invariant T cells (MAITs; gated as lymphocytes/single-cells/live-cells/CD3+/CD161++TCR-Vα7.2+) from one non-infected control (left; orange) and one SARS-CoV-2 infected individual at the time of the first positive PCR infection (right; black). Numbers indicate percent positive for each marker including double positives. (**f**) Summary data for single marker (left) or co-expression (right) of activation markers by n = 116,386 total peripheral MAIT cells from n = 18 individual participants; n = 9 uninfected controls (open circles) and n = 9 individuals with co-incident infection (closed circles). P value shown for two-sided Mann-Whitney-U test. Bars, median.(**g**) Marker gene expression of response states observed in ciliated cells. (**h**) Boxplot validating relative changes in acute phase response ciliated cell signature expression in our validation cohort of bulk RNA-seq data of nasopharyngeal swabs from sustained infection cases, n = 61. P value shown from the comparison pre-infection to day 1 post-inoculation was determined using a two-sided Mann-Whitney U test. In all box plots, the central line is the median, the box shows the IQR and the whiskers are extreme values upon removing outliers. (**i**) Dotplot as in Extended Data Fig. 7, showing changes in myeloid cell type abundances compared to pre-infection in sustained infection cases that significantly change on at least one time point compared to pre-infection, for nasopharyngeal (bottom) and circulating myeloid cells (top). The size of the circle denotes the false discovery rate (FDR) The green color scale of the adjacent heatmap depicts the proportion of each cell type relative to all cells.

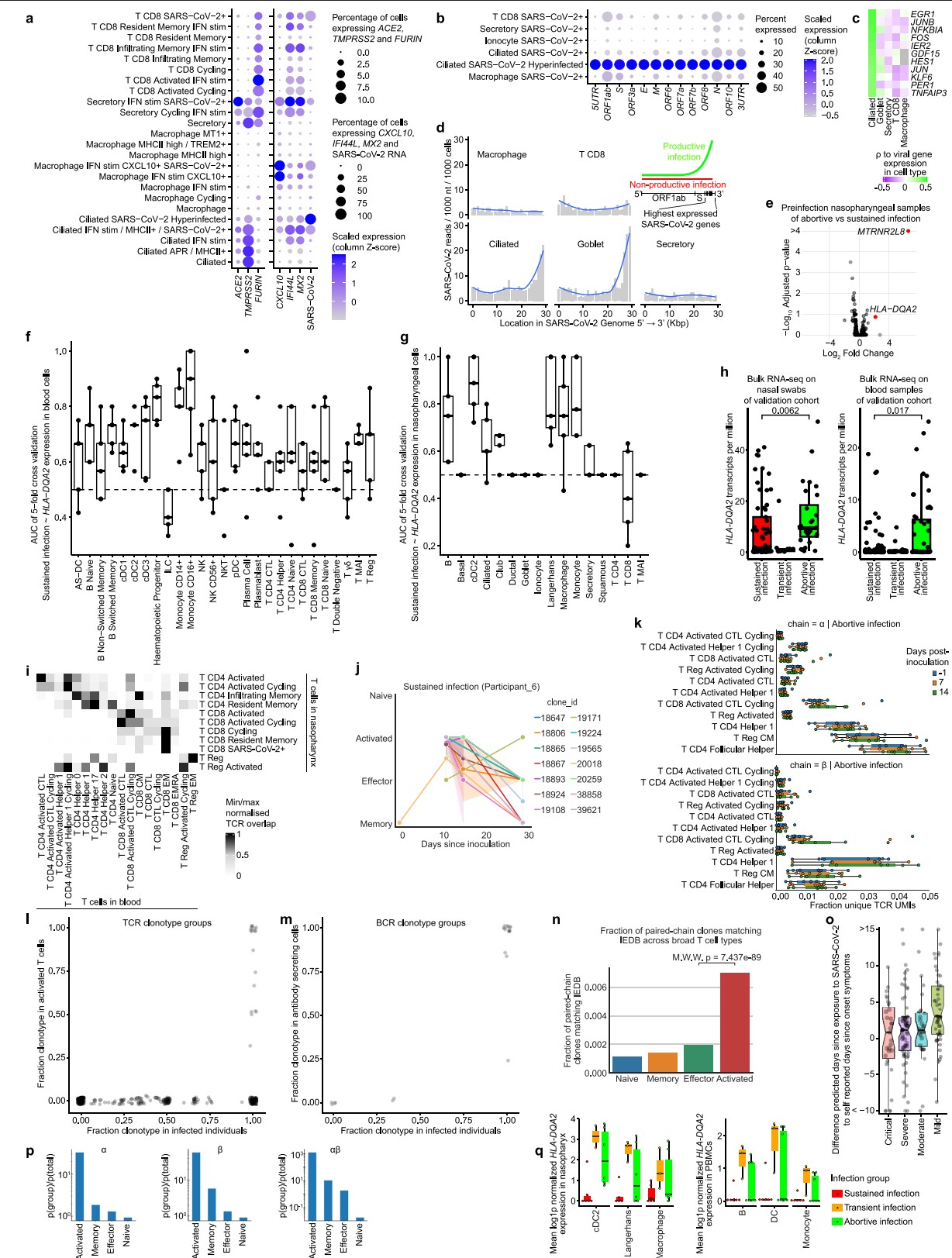

**Extended Data Fig. 6** | See next page for caption.

**Extended Data Fig. 6 | Temporal response states and activated T cells.**
(**a**) Dotplot visualizing the mean expression of viral entry factors (*ACE2, TMPRSS2, FURIN*) and SARS-CoV-2 induced (interferon signalling related) genes (*CXCL10, ILI44L, MX2*), and viral reads. (**b**) Expression of viral genes by genomic region for each cell type with viral reads. (**c**) Heatmap of Spearman correlations between host gene expression and number of viral reads per cell, split by cell type. Shown genes have the highest correlation with viral reads in ciliated cells. (**d**) Barplots showing the distribution of detected viral reads over the SARS-CoV-2 genome in the five most highly infected cell types. The blue line represents a LOESS fit over the data. The top-right inset illustrates a uniform read distribution versus a 3′ biased read distribution. (**e**) Volcano plot of a differential gene expression analysis comparing pre-inoculation nasopharyngeal cells (day -1) from subsequent abortive infection cases to sustained infection cases. Adjusted P values were calculated with a two-sided Wald test while accounting for sex and cell type. (**f**) Boxplot showing the predictive power of circulating cell type-specific *HLA-DQA2* expression in predicting before inoculation (day -1) if a participant subsequently becomes sustained infected. Five-fold cross validation using a 1:1 test-train split is shown in a logistic regression model, based on the mean *HLA-DQA2* expression and fraction of *HLA-DQA2* expressing cells per cell type. (**g**) Boxplot as in (f), but showing the predictive power of *HLA-DQA2* expression in nasopharyngeal cell types. (**h**) *HLA-DQA2* expression in our validation bulk RNA-seq datasets including all timepoints, split by infection group, for blood (n = 216) and nasal (n = 100) samples. P values were determined using a two-sided Mann-Whitney-U test. (**i**) TCR repertoire overlap of nasopharyngeal and circulating conventional T cells, stratified by cell state. We only considered the beta TCR chain to identify overlapping T cell clones and included T cells without a detected TRA sequence. (**j**) Memory formation analysis in an individual with sustained SARS-CoV-2 infection. Unique TCR clones are distinguished by color and numbered with their clone_id identifier. A shaded area is drawn when the same clone appeared with several distinct cell type labels, and the size of the shaded area informs their relative ratios. (**k**) TCR bulk data with matched single-cell labels as in Fig. 3g, but showing the fraction of unique TCR UMIs on abortive infections for activated and other T cells. No particular changes are observed across the three time points sampled. n = 4123 T cells examined over 29 unique samples. (**l**) The fraction of activated T cells that participate in TCR clonotype groups versus the fraction of cells in each group that originate from participants with sustained infections, which reveals that clonotype groups that contain activated T cells are exclusively populated by T cells from sustained infections. Clonotype groups are defined based on TCR distance as described in detail in the methods, and can include T cells from multiple participants and several T cell subtypes. (**m**) Scatterplot as in (l), but showing BCR clonotype groups and the fraction of antibody secreting B cells instead of activated T cells. (**n**) Fraction of unique paired-chain clones matching SARS-CoV-2 entries in Immune Epitope Database (IEDB) across all T cell clones within that broad T cell compartment. Significance level after two-sided Whitney-Mann testing shown for activated vs effector T cells (putative SARS-CoV-2 fraction 3.5 times higher in activated T cells, p = 7.437 ∗ $10^{-89}$). (**o**) Temporal inference on PBMCs from publicly available COVID-19 patients (n = 210), showing the difference between predicted time since viral exposure and reported time since onset of symptoms, split by reported severity. (**p**) Coincidence analysis of TCR sequence diversity restriction in phenotypic subsets. Fraction of clonotype pairs within each phenotypic cluster that share identical CDR3 amino acid sequences (but distinct nucleotide sequences) normalized by the same statistics calculated across all clonotypes, for alpha, beta, and both chains together. The ratio of within cluster versus overall sequence coincidence probabilities is a measure of the breadth of epitopes targeted by the different clonotypes within a cluster[60]. (**q**) Boxplot showing the pre-infection expression of *HLA-DQA2* (n = 16 participants) in nasopharyngeal (left) and circulating (right) professional antigen presenting cell types, across participants and the infection groups. In all box plots, the central line and the notch are the median and its approximate 95% confidence interval, the box shows the IQR and the whiskers are extreme values upon removing outliers.

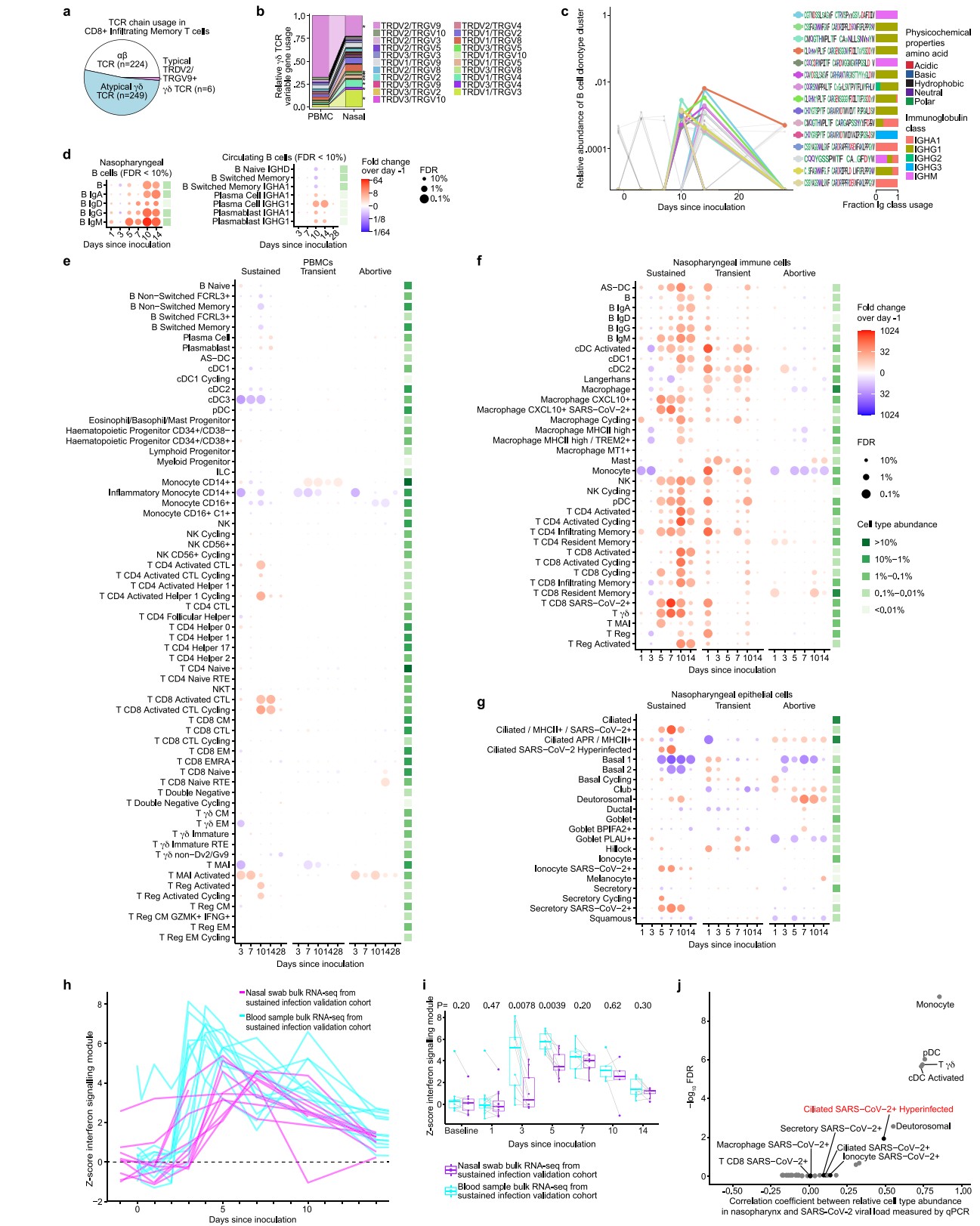

**Extended Data Fig. 7** | See next page for caption.

**Extended Data Fig. 7 | Detailed temporal dynamics in cell state abundances.**
(**a**) Proportion of CD8+ infiltrating T cells that use αβ TCRs, typical Dv2/Gv9 γδ TCRs, or atypical γδ TCRs is shown. (**b**) The relative immune repertoire composition of γδ T cells in circulation and nasopharynx after challenge are shown in the left and right bars, respectively. γδ chain pairs that are significantly more or less abundant between circulation and nasopharynx (p < 0.05) are highlighted with an asterisk. Exact uncorrected P values are 0.02 for TRDV2_TRGV9, TRDV1_TRGV4, TRDV3_TRGV4, and TRDV1_TRGV3, and 0.03 for TRDV3_TRGV5, TRDV1_TRGV10, TRDV3_TRGV8, and TRDV1_TRGV5, and were determined using a two-sided Mann-Whitney-U test. (**c**) Plot as in Fig. 3h, but showing BCR clusters. Immunoglobulin class usage within each activated BCR cluster is shown in the rightmost bars. (**d**) Dotplot as in Fig. 3f, showing the fold changes in B cells in sustained infections. Legend for mean cell type proportions (f). (**e**) Fold changes in abundance of cell states in PBMCs. Detailed annotation of interferon stimulated subsets and immunoglobulin class specific cell states are not shown for clarity. Immune cell abundances were scaled to the total amount of detected PBMCs in every sample prior to calculating the fold changes over days since inoculation compared to pre-infection (day -1) by fitting a GLMM on scaled abundances. The mean cell type proportions over all cells and samples is shown in the green heatmap right of the dotplot to aid the interpretation of changes in cell type abundances. (**f**) Dotplot as in (e), but showing nasopharyngeal immune cells. Immune cell abundances were scaled to the total amount of detected epithelial cells in every sample. (**g**) Dotplot as in (e), but showing nasopharyngeal epithelial cells. (**h**) Linegraph validating the relative expression dynamics over time since inoculation of the type I interferon signalling signature from[12] in sustained infection cases from our validation bulk RNA-seq datasets. (**i**) Boxplots showing bulk RNA-seq measurements of type I interferon signalling in blood and nasal swabs over time as shown in (h), but only focussing on samples with a paired blood and nasal measurement to perform paired analyses. Uncorrected P values of a paired two-sided Mann-Whitney-U test comparing nasal and blood samples at each time point are shown at the top. Preinfection baseline nasal and blood samples were collected at the day before and the same day as the inoculation, respectively. In all box plots, the central line is the median, the box shows the IQR and the whiskers are extreme values upon removing outliers. (**j**) Correlation analysis of relative cell type abundance and viral load as determined by qPCR. Peason correlation coefficients are shown on the X axis. Minus $\log_{10}$ transformed p values shown on the Y axis were corrected for multiple testing by the Benjamini-Hochberg procedure. Only infected cell types or cell types with an FDR < 0.01 are labeled. Dots from infected cell types were coloured black.

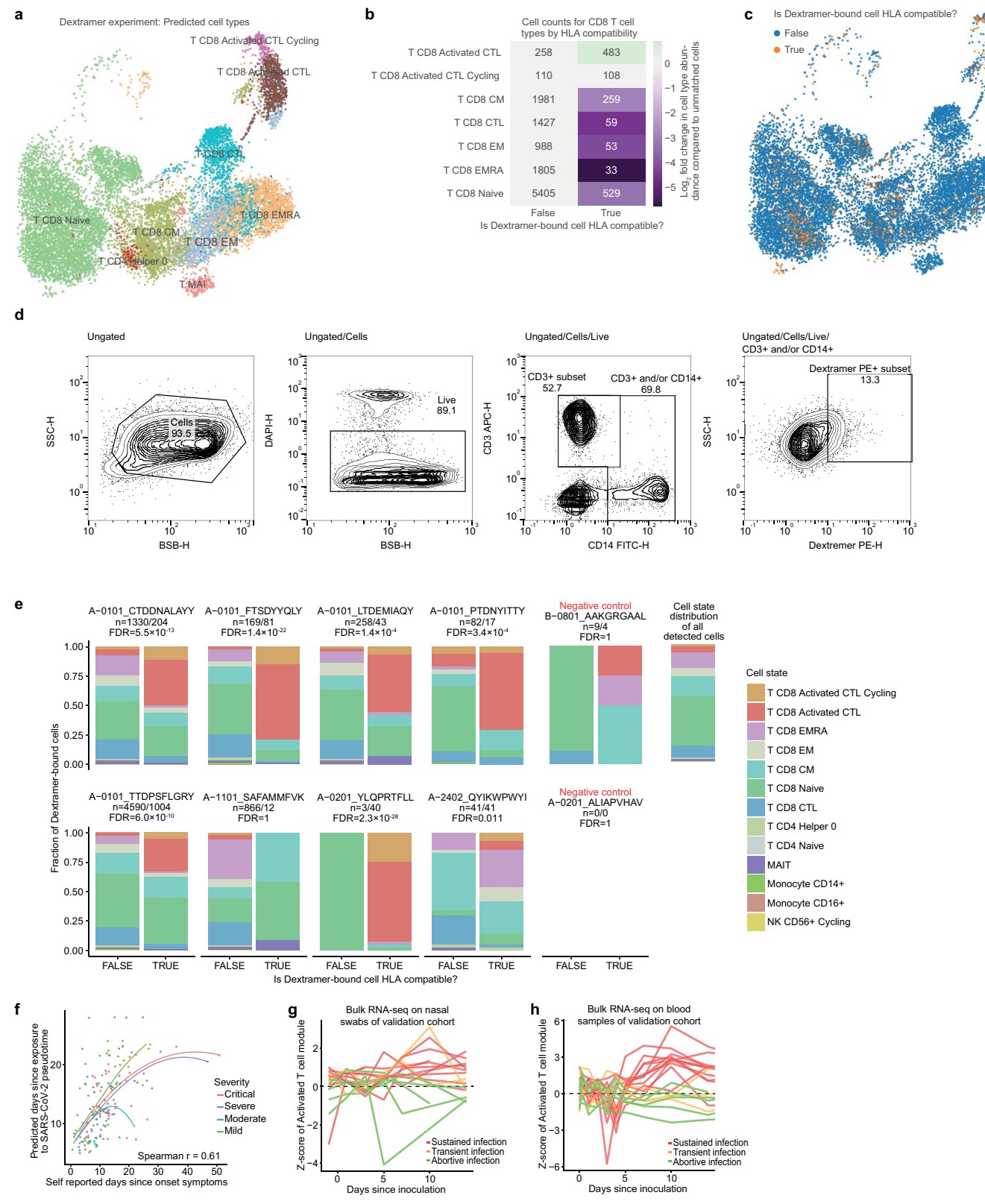

**Extended Data Fig. 8** | See next page for caption.

**Extended Data Fig. 8 | Validation of antigen-specific activated T cells.**
(**a**) UMAP of all CD8+ T cells from the Dextramer assay, with cell types predicted by CellTypist model trained on previous PBMC data. Activated T cells form a distinct cluster. (**b**) Cell counts for CD8+ T cell types by HLA compatibility of donor with the highest-bound Dextramer. Only Activated T cells have positive log2 fold change for HLA-matched Dextramers. (**c**) UMAP as in (a), colored by HLA compatibility, again showing enrichment of activated T cells amongst HLA compatible pairs. (**d**) Gating strategy used to enrich SARS-CoV-2 antigen specific T cells via MACSQuant Tyto cell sorting. Cells were sequentially stained with a multi-allele panel of dCODE dextramer- PE complexes, with the addition of anti-human CD3-APC and CD14-FITC FACS antibodies as references to help us identify T cell specific binding. Debris and cell aggregates were gated out first using BSB-H (backscatter blue laser-height) SSC-H (side scatter-height). From the cells, DAPI+ dead cells were excluded. T cells (CD3+) and monocytes (CD14+) were then gated for (CD3+ and\or CD14+ population) and the sort gate defined from this population as all PE-dCODE Dextramer® positive cells. This lenient sorting strategy was decided upon in order to collect enough cells for 10×5' single-cell analysis downstream and to ensure we were capturing all SARS-CoV-2 antigen specific cells. Any non-specific binding (e.g. to monocytes) and background noise could then be removed computationally. (**e**) Proportions of activated T cells bound to Dextramers loaded with selected SARS-CoV-2 antigens. The total amount of bound cells to each Dextramer is shown, color-coded by predicted cell state. If barcodes from several Dextramers were detected to be bound to the same cell, we only selected the Dextramer with the highest signal as bound. As a control to separate background and real binding, cells are separated based on the HLA haplotype compatibility with the tested Dextramer. Only Dextramers with at least 10 HLA matched bound cells are shown. FDR corrected p values were determined by a Fisher-exact test comparing the proportion of HLA matched activated T cells in the Dextramer bound cells to the proportion of unbound HLA matched activated T cells. N represents the number of cells in each bar. The right-most bar represents the overall distribution of cell types across all Dextramer experiments. (**f**) Predicted time since viral exposure is plotted against reported time since onset of symptoms. Lines represent LOESS fits of the data split and color coded by reported severity. (**g**) Linegraph validating the relative expression dynamics across time of the activated T cell signature shown in Fig. 4b, in our bulk RNA-seq validation dataset of nasal swabs. (**h**) Linegraph as in (g), but showing bulk RNA-seq blood samples.

**a** Control (Mock infection)

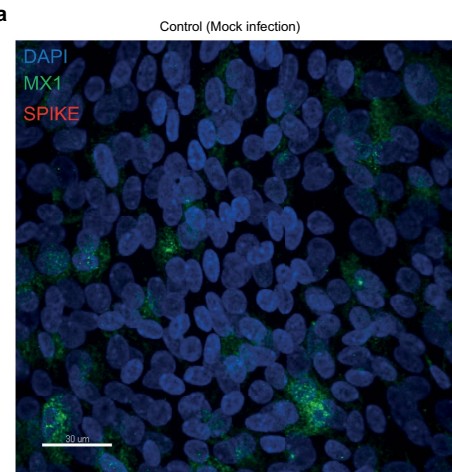

DAPI
MX1
SPIKE

30 µm

**b**

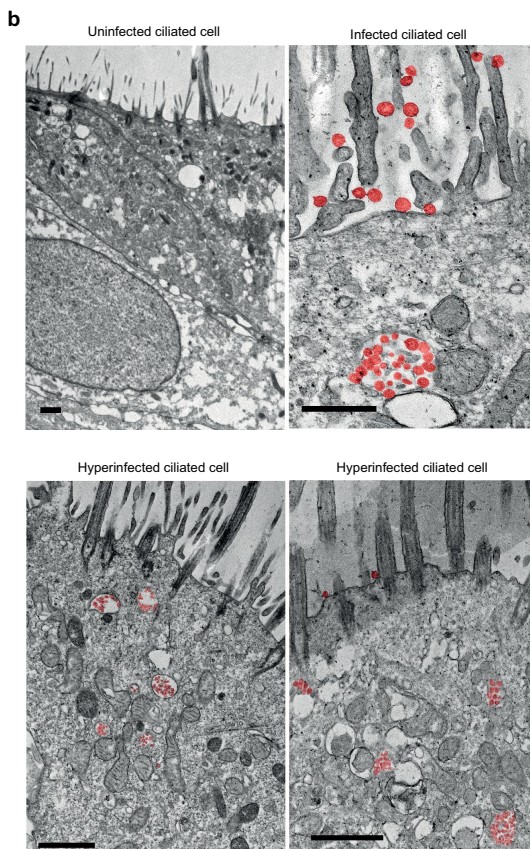

Uninfected ciliated cell    Infected ciliated cell

Hyperinfected ciliated cell    Hyperinfected ciliated cell

**Extended Data Fig. 9 | Controls for microscopy data. (a)** Representative immunofluorescence confocal image of mock infected pediatric human nasal epithelial cultures grown at air-liquid interface at 72 h post-infection. Image shown as a maximum intensity projection of the z-stack. Cells are stained with antibodies for MX1 (green), SARS-CoV-2 spike protein (red) and DAPI (blue). Scale bar represents 30 µm. **(b)** Representative transmission electron micrographs of an uninfected ciliated cell (top left) and infected ciliated cell (top right) or hyper-infected ciliated cells (bottom panels). SARS-CoV-2 viral particles are false colored with red to aid visualization. Images taken using SARS-CoV-2 infected human nasal epithelial cultures grown in air-liquid interface 72 h post-infection. Scale bar represents 1 µm.

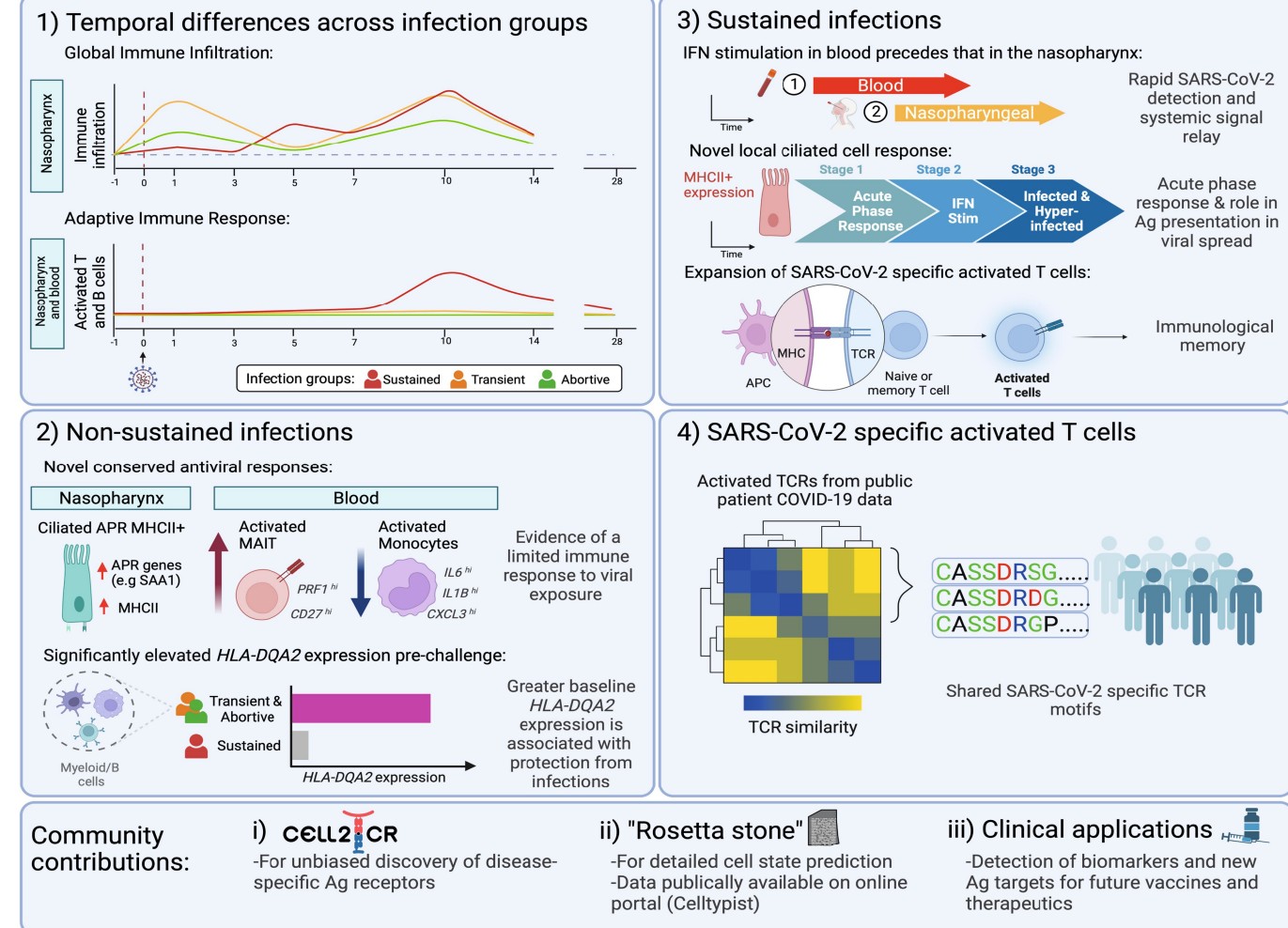

**Extended Data Fig. 10 | Temporally resolved epithelial and immune response in SARS-CoV-2 infections.** Summary figure highlighting the key finding from the study. These includes; 1) distinct temporal differences in the cellular dynamics observed between the different infection groups; 2) several novel conserved antiviral responses and higher baseline expression of *HLA-DQA2* in participants who were exposed to the virus but who did not go on to develop a sustained infection; 3) novel characteristics of sustained infection, with a rapid relay observed in the blood compared to the site of inoculation, a dynamic local ciliated response occurring early on during infections (pre-symptoms) and a temporally restricted, distinct, SARS-CoV-2 specific activated T cell population which leads to immunological memory; and 4) the identification of public motifs in SARS-CoV-2 specific activated T cells. In addition, our work provides community tools for inference of specific TCR motifs (Cell2TCR) in activated T cells, a detailed publicly available reference database underpinning the detection of future biomarkers and antigen (Ag) targets for therapeutic applications. Schematic created with BioRender.com.

Marko Z Nikolić  
Rik GH Lindeboom

# Reporting Summary

## Statistics

For all statistical analyses, confirm that the following items are present in the figure legend, table legend, main text, or Methods section.

| n/a | Confirmed | |
|---|---|---|
| ☐ | ☒ | The exact sample size (*n*) for each experimental group/condition, given as a discrete number and unit of measurement |
| ☐ | ☒ | A statement on whether measurements were taken from distinct samples or whether the same sample was measured repeatedly |
| ☐ | ☒ | The statistical test(s) used AND whether they are one- or two-sided<br>*Only common tests should be described solely by name; describe more complex techniques in the Methods section.* |
| ☐ | ☒ | A description of all covariates tested |
| ☐ | ☒ | A description of any assumptions or corrections, such as tests of normality and adjustment for multiple comparisons |
| ☐ | ☒ | A full description of the statistical parameters including central tendency (e.g. means) or other basic estimates (e.g. regression coefficient) AND variation (e.g. standard deviation) or associated estimates of uncertainty (e.g. confidence intervals) |
| ☐ | ☒ | For null hypothesis testing, the test statistic (e.g. *F*, *t*, *r*) with confidence intervals, effect sizes, degrees of freedom and *P* value noted<br>*Give P values as exact values whenever suitable.* |
| ☒ | ☐ | For Bayesian analysis, information on the choice of priors and Markov chain Monte Carlo settings |
| ☒ | ☐ | For hierarchical and complex designs, identification of the appropriate level for tests and full reporting of outcomes |
| ☐ | ☒ | Estimates of effect sizes (e.g. Cohen's *d*, Pearson's *r*), indicating how they were calculated |

*Our web collection on statistics for biologists contains articles on many of the points above.*

## Software and code

Policy information about availability of computer code

**Data collection**  
Single-cell RNA-seq and CITE-seq data from PBMCs was jointly aligned against the GRCh38 reference that 10X Genomics provided with CellRanger 3.0.0, and alignment was performed using CellRanger 4.0.0. CITE-seq antibody-derived tag (ADT) barcodes were aligned against a barcode reference provided by the supplier, which we annotated to add informative protein names and made available in our GitHub repository. Single-cell RNA-seq data from nasopharyngeal swab samples were aligned against the same reference using STARSolo 2.7.3a, and post-processed with an implementation of emptydrops extracted from CellRanger 3.0.2. To detect viral RNA in infected cells, we added 21 viral genomes including pre-Alpha SARS-CoV-2 (NC_045512.2) to the above mentioned reference genomes for RNA-seq alignment, as described in Yoshida et al, Nature, 2022. Single cell alpha/betaTCR and BCR data was aligned using CellRanger 4.0.0 with the accompanying GRCh38 VDJ reference that 10X Genomics provided. Single cell gamma/delta TCR data was aligned against the GRCh38 reference that 10X Genomics provided with CellRanger 5.0.0, using CellRanger 6.1.2.

**Data analysis**  
Both single cell RNA-seq and ADT-seq data were corrected using SoupX 1.5.2 (Young and Behjati, 2020) to remove free-floating and background RNAs and ADTs. To correct ADT counts, SoupX 1.5.2 parameters soupQuantile and tfidfMin parameters were set to 0.25 and 0.2, respectively, and lowered by decrements of 0.05 until the contamination fraction was calculated using the autoEstCont function. SoupX on RNA data was performed using default settings. To confidently annotate SARS-CoV-2 infected cells, we used SoupX corrected viral RNA counts to remove false positives due to freely floating SARS-CoV-2 virions. To profile the distribution of viral reads, we removed PCR duplicates from the aligned BAM files that STARSolo produced with MarkDuplicates in picard (https://broadinstitute.github.io/picard/), and tallied the location within the SARS-CoV-2 genome using the start of each sequencing read. Aligned single cell RNA-seq data was imported from the filtered_feature_bc_matrix folder into Seurat V4.1.0 for processing, keeping only cells with at least 200 RNA features detected. Nasopharyngeal and PBMC cells with more than 50% and 10% of the counts coming from mitochondrial genes were excluded, respectively. SoupX corrected gene expression and ADT counts were normalized by dividing it by the total counts per cell and multiplying by 10 000, followed by adding one and a natural-log transformation (log1p).

Each PBMC sample was pooled twice into two unique pools containing up to four PBMC samples per pool, followed by CITE-seq and single cell VDJ sequencing as described above. Souporcell V2.0 (Heaton et al. 2020) was used to demultiplex each pools based on the genotype differences between the mixed samples. Souporcell analyses were performed with the skip_remap parameter enabled and using the common SNP database that was provided by the software. We used two complementary approaches to confidently assign participant identity to each souporcell cluster. First we compared the cluster genotypes with SNP array derived genotyping data, generated for all participants and performed using the Affymetrix UK Biobank AxiomTM Array kit by Cambridge Genomic Services (CGS). Second, the combinations of samples within each pool was unique, enabling assignment of participant identity based on the presence of unique participant-specific combinations of identical genotypes in two separate pools. This multiplexing and replication strategy furthermore enabled us to distinguish library specific batch effects from participant specific effects in downstream analyses.

Aligned single cell BCR and alpha/beta TCR sequencing data was imported in scirpy to obtain a cell by TCR or BCR formatted table, which was then added to Seurat objects containing gene expression data. Aligned single cell gamma/delta TCR data was reannotated using Dandelion V0.2.4. TCR sequences were compared to human SARS-CoV-2 specific entries from https://www.iedb.org/ fetched on 24.08.2023.

All custom code developed in this study is publicly available at: https://github.com/Teichlab/COVID-19_Challenge_Study, with the 'Release for Nature publication' version marking the last commit (90e64cb) before submission.

Other bioinformatics analyses used the following packages with version:

Python: python (3.9.16), ipykernel (6.14.0), numpy (1.23.5), pandas (1.5.3), scanpy (1.9.3), celltypist (1.3.0), tcrdist3 (0.2.2), igraph (0.10.4), leidenalg (0.9.1), matplotlib (3.7.1), seaborn (0.11.2), logomaker (0.8), celltcr (0.1), statannotations (0.5.0), scipy (1.10.1),

R: R (4.0.4), Seurat (4.0.1), tidyverse (1.3.1), ggplot2 (3.3.6), harmony (1.0), ComplexHeatmap (2.6.2), sceasy (0.0.6), reticulate (1.18), SoupX (1.5), rvcheck (0.2.1), cardelino (1.4.0), randomcoloR (1.1.0.1), ggh4x (0.2.8), circlize (0.4.15), readr (1.4.0), lme4 (1.1-29), Matrix (1.3-2), numDeriv (2016.8-1.1), Rsamtools (2.6.0), GenomicAlignments (1.26.0), msigdbr (7.5.1), fgsea (1.28.0), glmmSeq (0.1.1), future (1.21.0), igraph (1.2.6), leiden (0.3.7), ggseqlogo (0.2), patchwork (1.1.1)

For manuscripts utilizing custom algorithms or software that are central to the research but not yet described in published literature, software must be made available to editors and reviewers. We strongly encourage code deposition in a community repository (e.g. GitHub). See the Nature Portfolio guidelines for submitting code & software for further information.

# Data

Policy information about availability of data

All manuscripts must include a data availability statement. This statement should provide the following information, where applicable:

- Accession codes, unique identifiers, or web links for publicly available datasets
- A description of any restrictions on data availability
- For clinical datasets or third party data, please ensure that the statement adheres to our policy

The data presented in this study can be explored and analyzed interactively through our COVID-19 Cell Atlas web portal (https://covid19cellatlas.org). The cell state annotation model is available in the CellTypist model repository (https://www.celltypist.org/models) under the name 'COVID19_HumanChallenge_Blood'. A reference for our Multi Task Gaussian Process Regression model to infer time since viral exposure on PBMC data is available at our GitHub repository (https://github.com/Teichlab/COVID-19_Challenge_Study). The raw sequencing data is available under controlled access at the European Genome-Phenome Archive under accession number EGAD00001012227. Processed bulk RNAseq data is available at ArrayExpress (accession number: E-MTAB-12993). Single-cell count matrices with metadata are available at https://www.covid19cellatlas.org/ as h5ad files.

# Human research participants

Policy information about studies involving human research participants and Sex and Gender in Research.

| Reporting on sex and gender | No sex- or gender-based analyses were performed. This study is based on 16 participants, which is not an appropriate sample size to confidently look for sex- or gender-related effects. |
| --- | --- |
| Population characteristics | Sero-negative (no evidence of COVID-19 infection or previous vaccination) healthy male and female volunteers 18-30 years of age (inclusive) with no known risk factors for severe COVID-19. |
| Recruitment | Screening of potential participants took place in two stages with an initial screening visit, followed by a study specific remote consultation to go through the full study participant information following adequate time for the informed consent form (ICF) and participation in the study to be considered. Screening visits took place between Day -90 to Day -2. Potential participants were screened under a separate study-specific screening protocol using a screening ICF and advertising material that was approved by the Research Ethics Committee (REC) and Health Research Authority (HRA). Screening activities under the separate screening protocol continued up until subjects sign the study specific consent. Recruitment was done through a number of channels: <br>• Approved advertising, including social media <br>• hVIVO volunteer database (Volunteers already registered with any other hVIVO database may be contacted to determine their interest in participating in SARS-CoV-2 research.) <br>• Referral <br>• Organic search (e.g. via Google or other search engines) <br><br>The participant sample was biased by the age criteria (18-30 years) and requirement to be healthy with no co-morbidities or known risk factors for severe COVID-19 based on clinical history, blood tests and radiology. There was potential self-selection |

bias as participation was voluntary and instigated by the volunteers. Due to these factors, direct extrapolation of the results to young children, older adults, those with pre-existing conditions and minority groups may not be possible.

Ethics oversight

This study was conducted in accordance with the protocol, the Consensus ethical principles derived from international guidelines including the Declaration of Helsinki and Council for International Organizations of Medical Sciences (CIOMS) International Ethical Guidelines, applicable ICH Good Clinical Practice guidelines, applicable laws and regulations. The screening protocol and main study were approved by the UK Health Research Authority – Ad Hoc Specialist Ethics Committee (reference: 20/UK/2001 and 20/UK/0002).

Note that full information on the approval of the study protocol must also be provided in the manuscript.

# Field-specific reporting

Please select the one below that is the best fit for your research. If you are not sure, read the appropriate sections before making your selection.

☒ Life sciences    ☐ Behavioural & social sciences    ☐ Ecological, evolutionary & environmental sciences

For a reference copy of the document with all sections, see nature.com/documents/nr-reporting-summary-flat.pdf

# Life sciences study design

All studies must disclose on these points even when the disclosure is negative.

| | |
|---|---|
| Sample size | No sample size calculation was performed. As these are scarce samples, we collected and analyzed as many samples from the two quarantine groups of participants we had access to, which is how the presented sample size was established. |
| Data exclusions | No samples were excluded from analysis. |
| Replication | All available samples from two distinct quarantine groups were analyzed. We analyzed the dataset comparing three infection groups (6, 3 and 7 participants per group), looking at changes in PBMCs and nasopharyngeal swabs over time (13-9 samples per participant, which always included all possible samples we were able to obtain). Each PBMC sample was measured twice, each nasopharyngeal sample was measured once. All attempts at replication were successful. |
| Randomization | None. Participants were not allocated in groups, but all received the same treatment. |
| Blinding | Blinding was not relevant as all participants received the same treatment and were not allocated to groups. |

# Reporting for specific materials, systems and methods

We require information from authors about some types of materials, experimental systems and methods used in many studies. Here, indicate whether each material, system or method listed is relevant to your study. If you are not sure if a list item applies to your research, read the appropriate section before selecting a response.

## Materials & experimental systems

| n/a | Involved in the study |
|---|---|
| ☐ | ☒ Antibodies |
| ☒ | ☐ Eukaryotic cell lines |
| ☒ | ☐ Palaeontology and archaeology |
| ☒ | ☐ Animals and other organisms |
| ☐ | ☒ Clinical data |
| ☒ | ☐ Dual use research of concern |

## Methods

| n/a | Involved in the study |
|---|---|
| ☒ | ☐ ChIP-seq |
| ☒ | ☐ Flow cytometry |
| ☒ | ☐ MRI-based neuroimaging |

# Antibodies

Antibodies used

For CITE-seq: 137 TotalSeq-C Human Cocktail, V1.0 antibodies (BioLegend, cat. # 99814399905). The reagents that were provided were a pre-diluted commercial panel.

For Dextramer® SARS-CoV-2 antigen specific enrichment via MACSQuant Tyto cell sorting cells were stained with: anti-human CD14 conjugated to FITC (clone: HCD14 , Biolegend cat. # 325603); anti-human CD3 conjugated to APC (clone: UCHT1, Biolegend cat #300458 ); PE-dCODE Dextramer® (10x) - Gold, SARS-CoV-2 Multi Allele Panel -XL from Immudex. The latter consists of 44 SARS-CoV-2 antigen specific dCODE™ Dextramer® regents, including a 29 MHC I dCODE Dextramer® reagents (Cat #WA05973dXG PE 50 fBC0587, WA05972dXG PE 50 fBC0588, WB05939dXG PE 50 fBC0589, WB05824dXG PE 50 fBC0590, WC05754dXG PE 50 fBC0591, WD05981dXG PE 50 fBC0592, WD05754dXG PE 50 fBC0593, WF05952dXG PE 50 fBC0594, WF06031dXG PE 50 fBC0595, WH05842dXG PE 50 fBC0596, WB02666dXG PE 50 fBC0597, WI03233dXG PE 50 fBC0598, WA06027dXG PE 50 fBC0599, WA06028dXG PE 50 fBC0600, WA06081dXG PE 50 fBC0601, WA05846dXG PE 50 fBC0602, WA06029dXG PE 50 fBC0603,

WB05948dXG PE 50 fBC0604, WB06025dXG PE 50 fBC0605, WB05762dXG PE 50 fBC0606, WC06082dXG PE 50 fBC0607, WC05978dXG PE 50 fBC0608, WD06030dXG PE 50 fBC0609, WD06083dXG PE 50 fBC0610, WD05982dXG PE 50 fBC0611, WF05984dXG PE 50 fBC0612, WH06032dXG PE 50 fBC0613, WH05888dXG PE 50 fBC0614, WH05879dXG PE 50 fBC0615)  plus an additional 15 MHC II dCODE Dextramer® reagents (Cat # FA10157DXG PE 25 FBC0351, FA10160DXG PE 25 FBC0352, FA10161DXG PE 25 FBC0353, FA10162DXG PE 25 FBC0354, FA10164DXG PE 25 FBC0355,FA10165DXG PE 25 FBC0356, FA10167DXG PE 25 FBC0357, FA10168DXG PE 25 FBC0358, FA10169DXG PE 25 FBC0359, FA10170DXG PE 25 FBC0360, FA10171DXG PE 25 FBC0361, FA10172DXG PE 25 FBC0362, FA10173DXG PE 25 FBC0363, FA10175DXG PE 25 FBC0364, FA10002DXG PE 25 FBC0365).

| Validation | All antibodies employed were commercial antibodies. |
|---|---|

137 TotalSeq-C Human Cocktail, V1.0 antibodies validation:
Proteogenomics quality tested. This panel has been optimized on human PBMCs. Full validation results can be downloaded at the suppliers website: https://www.biolegend.com/en-us/products/totalseq-c-human-universal-cocktail-v1-0-19736

anti-human CD14 conjugated to FITC validation for flow cytometry (FC):
FC quality tested. Each lot of this antibody is quality control tested by immunofluorescent staining with flow cytometric analysis. For flow cytometric staining, the suggested use of this reagent is 5 µl per million cells in 100 µl staining volume or 5 µl per 100 µl of whole blood.
Application References: McMichael A, et al. 1987. Leucocyte Typing III. Oxford University Press. New York.; Knapp W, et al. Eds. 1989. Leucocyte Typing IV. Oxford University Press. New York.; Schlossman S, et al. Eds. 1995. Leucocyte Typing V. Oxford University Press. New York.

anti-human CD3 conjugated to APC validation for flow cytometry (FC):
FC quality tested. Each lot of this antibody is quality control tested by immunofluorescent staining with flow cytometric analysis. For flow cytometric staining using the µg size, the suggested use of this reagent is ≤0.25 µg per million cells in 100 µl volume. It is recommended that the reagent be titrated for optimal performance for each application. For flow cytometric staining using the test sizes, the suggested use of this reagent is 5 µl per million cells in 100 µl staining volume or 5 µl per 100 µl of whole blood.
FC application References: Thakral D, et al. 2008. J. Immunol. 180:7431.; Yoshino N, et al. 2000. Exp. Anim. (Tokyo) 49:97.

# Clinical data

Policy information about clinical studies

All manuscripts should comply with the ICMJE guidelines for publication of clinical research and a completed CONSORT checklist must be included with all submissions.

| Clinical trial registration | Clinicaltrials.gov NCT04865237 |
|---|---|
| Study protocol | Study protocol is described in Killingley et al, Nature Medicine, 2022. |
| Data collection | The study was conducted at the Queen Mary BioEnterprises (QMB) Innovation Centre, London, UK (outpatient screening and follow-up visits) and Royal Free London NHS Trust, London, UK (in-patient quarantine). The first date of participant enrollment was 6th March 2021 and the last was 8th July 2021.<br>Data collection occurred at:<br>Study specific screening Day -90 to Day -2<br>Quarantine Phase Day -2 to Day 14 (+ extended days)<br>Follow up visits Day 28 (+/-3 days), Day 90 (+/- 7 days), Day 180 (+/- 14 days), Day 270 (+/- 14 days) and Day 360 (+/- 14 days) |
| Outcomes | Primary Objective / Endpoint:<br>• To identify a safe and infectious dose of wild type SARS-CoV-2 in<br>healthy volunteers, suitable for future intervention studies, that:<br>• has an acceptable safety profile as measured by:<br>o Occurrence of Adverse Events (AEs) within 30 days<br>post-viral challenge (Day 0) up to Day 28 follow up.<br>o Occurrence of Serious Adverse Events (SAEs) from<br>the viral challenge (Day 0) up to Day 28 follow up.<br>• induces laboratory confirmed infection in ≥50% of participants<br><br>Secondary Objectives / Endpoints:<br><br>Objective: To further assess SARS-CoV-2 viral infection rates in upper respiratory samples in healthy volunteers, by inoculum dose<br>Endpoints: To assess the incidence of laboratory confirmed infection rates using a) mid turbinate samples, b) throat swabs, and c) both mid turbinate and throat swabs, as defined by:<br>• Variant 2:  Occurrence of at least two quantifiable (≥LLOQ) RT-PCR measurements, reported on 2 or more consecutive timepoints, starting from 24 hours post-inoculation and up to discharge from quarantine.<br>• Variant 3: Occurrence of at least two detectable (≥LLOD) RT-PCR measurements, reported on 2 or more consecutive timepoints, starting from 24 hours post-inoculation and up to discharge from quarantine.<br>• Variant 4: Occurrence of at least one quantifiable (≥LLOQ) SARS-CoV-2 viral cell culture measurement, starting from 24 hours post-inoculation and up to discharge from quarantine.<br><br>Objective: To assess the incidence of symptomatic SARS-CoV-2 infection, in healthy volunteers, by inoculum dose<br>Endpoints: To assess the incidence of lab-confirmed symptomatic SARS-CoV-2 infection using a) mid turbinate samples, b) throat swabs, and c) both mid turbinate and throat swabs, defined as:<br>• Variant 1: |

o Occurrence of at least two quantifiable (≥LLOQ) RT-PCR measurements, reported on 2 or more consecutive timepoints, starting from 24 hours post-inoculation and up to discharge from quarantine, AND
o Either one or more positive clinical symptoms of any grade from two different categories in the symptom scoring system (Upper Respiratory, Lower Respiratory, Systemic), or one Grade 2 symptom from any category
• Variant 2:
o Occurrence of at least two detectable (≥LLOD) RT-PCR measurements, reported on 2 or more consecutive timepoints, starting from 24 hours post-inoculation and up to discharge from quarantine, AND
o Either one or more positive clinical symptoms of any grade from two different categories in the symptom scoring system (Upper Respiratory, Lower Respiratory, Systemic), or one Grade 2 symptom from any category
• Variant 3:
o Occurrence of at least one quantifiable (≥LLOQ) SARS-CoV-2 viral cell culture measurement, starting from 24 hours post-inoculation and up to discharge from quarantine, AND
o Either one or more positive clinical symptoms of any grade from two different categories in the symptom scoring system (Upper Respiratory, Lower Respiratory, Systemic), or one Grade 2 symptom from any category

Objective: To assess the SARS-CoV-2 viral dynamics in upper respiratory samples (AUC, peak, duration, incubation period) in healthy volunteers, by inoculum dose
Endpoints: To assess the viral dynamics using a) mid turbinate samples, and b) throat swabs, as measured by:
• Area under the viral load-time curve (VL-AUC) of SARS-CoV-2 as determined by qRT-PCR, starting from 24 hours post-inoculation and up to discharge from quarantine.
• Peak viral load of SARS-CoV-2 as defined by the maximum viral load determined by quantifiable (≥LLOQ) qRT PCR measurements, starting from 24 hours post-inoculation and up to discharge from quarantine
• Duration of SARS-CoV-2 quantifiable (≥LLOQ) qRT PCR measurements, starting from 24 hours post-inoculation and up to discharge from quarantine. Duration is defined as the time (hours) from the first quantifiable of the two viral quantifiable positives used to assess infection until first confirmed undetectable assessment after their peak measure (after which no further virus is detected).
• Incubation period of SARS-CoV-2 qRT PCR measurements. Incubation period is defined as the time (hours) from inoculation to the first quantifiable of the two viral quantifiable positives used to assess infection, starting from 24 hours post-inoculation and up to discharge from quarantine.
The above endpoints will also be evaluated using quantitative cell culture.

Objective: To assess the SARS-CoV-2 induced symptoms, in healthy volunteers, by inoculum dose
Endpoints:
• Sum total symptoms diary card score: sum total clinical symptoms (TSS) as measured by graded symptom scoring system, starting one day post-viral challenge (Day 1) up to discharge from quarantine
• Area under the curve over time (TSS-AUC) of total clinical symptoms (TSS) as measured by graded symptom scoring system (categorical and visual analogue scales), starting one day post-viral challenge (Day 1) up to discharge from quarantine.
• Peak symptoms diary card score: peak total clinical symptoms (TSS) as measured by graded symptom scoring system (categorical and visual analogue scales, starting one day post-viral challenge (Day 1) up to discharge from quarantine
• Peak daily symptom score: Individual maximum daily sum of Symptom score starting one day post-viral challenge (Day 1) up to the end of quarantine.
• Number (%) of participants with Grade 2 or higher symptoms

Objective: To assess the incidence of SARS-CoV-2 illness, in healthy volunteers, by inoculum dose
Endpoints: The incidence of:
• Upper Respiratory Tract illness (URT)
• Lower Respiratory Tract illness (LRT)
• Systemic illness (SI)
• Febrile illness (FI)
• Proportion of Subjects with Grade 3 symptoms on any occasion at any time from the last assessment on Day 0 to quarantine discharge
• Proportion of Subjects with Grade 2 or higher symptoms on any occasion at any time from the last assessment on Day 0 to quarantine discharge
• Proportion of Subjects with Grade 2 or higher Symptoms on two separate occasions at any time from the last assessment on Day 0 to quarantine discharge
• Proportion of Subjects with any symptom (grade >=1) on any occasion at any time from the last assessment on Day 0 to quarantine discharge
• Proportion of Subjects with any symptom (grade >=1) on two separate occasions at any time from the last assessment on Day 0 to quarantine discharge

