## [Peer Review File · Nature]

Manuscript Title: Human SARS-CoV-2 challenge uncovers local and systemic response dynamics

Reviewer Comments & Author Rebuttals

Reviewer Reports on the Initial Version:

Referees' comments:

Referee #1 (Remarks to the Author):

This is a human challenge study designed to identify the local and systemic immune correlates of different mild outcomes of timed infection with pre-Alpha SARS-CoV-2. The authors used primarily single cell transcriptomics on cells obtained from nasopharyngeal swabs and blood samples to characterize immune response kinetics during infection, and created a detailed temporal atlas of these responses. Among their cohort of 16 participants, they identified three infection outcomes (sustained, transient and abortive) and carried out detailed and comprehensive analyses of cell-type specific activation and gene expression following exposure to SARS-CoV-2. In those characterized as having sustained infection, the authors demonstrated the following:

- Interferon signaling can be detected in the blood prior to at the site of viral inoculation
- Delayed nasopharyngeal immune infiltration compared to those with transient infection
- Changes in nasopharyngeal resident and circulating myeloid compartments during infection, including a notable decrease in inflammatory monocytes across all infection cohorts
- Activation across all infection cohorts of MAIT cells expressing PRF1
- Ciliated cells in the nasopharynx supported productive infection, while local T cells and macrophages did not (although they were able to acquire viral genome)
- Infected epithelial cells may present SARS-CoV-2 antigen to CD4+ T cells via MHCII
- Activated CD4+ T cells expand by day 10 post-inoculation and are capable of expressing cytotoxic genes
- Activated B cells expand and mature between days 10-14 post-inoculation and are characterized by mostly IgG1- and IgA1-secreting plasmablasts
- Identification of an atypical gamma-deltaTCR-expressing CD8+ T cell population
- Regulatory T cell types associated with immune tolerance and resolution of infection expanded by day 14 with a strong upregulation of IL-10

Despite several intriguing findings from this approach, the authors should take care not to overgeneralize their results, given the very small sample number

evaluated in this study (16, with only 6 developing a sustained infection) and the fact that all participants were in a very young age bracket (18-29). The limitations of this study with regard to interpreting more severe outcomes of SARS-CoV-2 infection were nicely stated in the discussion. The paper could also

benefit from a brief discussion of age-related dynamics in SARS-CoV-2 infection, as these results may be different in an older cohort.

The use of the word, “resolve,” in the title seems overly definitive, given this small cohort.

The authors should consider discussing how the dose used in this study (10 TCID₅₀) compares with a physiologically relevant real-world exposure.

An internal control of baseline prior to infection is used for subjects in this study. However, this doesn't account for innate immune responses that are generated simply by putting any foreign particle in your nose. Could the abortive infection group simply be people who responded to the stimulus in the absence of infection? Do PBMCs of transiently or abortively infected people

exhibit any recall immune response to SARS-CoV-2 antigen? How do the immune response dynamics during SARS-CoV-2 infection compare to those following intranasal vaccination?

Measurement of viral load by PCR is limited because it measures the presence of viral genome without being a good indicator of infectious virus. The authors do show focus-forming assay results of subjects with sustained infection in the supplemental data. The paper would benefit from moving some discussion of this to the main text, as well as including FFA data from individuals who were transiently infected to help show that PCR positivity was not a false result.

Additionally, an RNA-seq analysis at single cell resolution that includes data on viral genome or antigen expression would be helpful to better understand the cohorts depicted in Figure 1.

Figure 1 comments:

- Labels on panels b and d obscure visualization of the colors. May be better to use more general labels in the image to reduce amount of text, since legend is very clear
- This figure is missing data that show evidence of infection in any of these cells (e.g. presence of SARS-CoV-2 N transcripts) – see above comment

Figure 2 comments:

- Validation of the IFN response at the protein level would be helpful, as SARS-CoV-2 has numerous proteins that shut down translation of host transcripts. Therefore, even if an ISG is expressed at the RNA level, it may not ever be translated in an infected cell.
- Myeloid subsets (monocytes, activated DCs) reduced at site of inoculation at day 3 (trafficking to LNs) – also observed in transient & abortive infections
- Do inflammatory monocytes in transient/abortive infection have same activation marker expression as those in sustained infection? Would be useful to have a flow cytometric characterization of these cells to more comprehensively evaluate markers.
- Fig 2e and 2f – the figure legend could use more information, as it does not clarify what sample is being analyzed (sustained infection? one participant or

pooled data?)

It is surprising to learn that CD8+ T cells were infected in this study, albeit not productively. Please add references for the previous studies mentioned on this topic in line 213.

What is ACE2/TMPRSS2 expression on the cells that are infected? (ciliated, goblet, CD8, macrophages). This study could benefit from a flow cytometric analysis of these cells (ACE2, activation markers, ISG upregulation vs. SARS-CoV-2 antigen).

The authors should consider rewording the heading, “Hyper-infected ciliated cells are the main source of SARS-CoV-2 and produce anti-inflammatory molecules,” and toning down the definitive language of lines 234-242.

- It is difficult to determine whether ciliated cells are the “main source,” of virus when the only cells from the nasopharynx that were analyzed are those collected by a nasopharyngeal swab.
- The authors have not shown any evidence that ciliated cells, despite being infected, are transmitting virus to other cells.
- The term “hyper-infected” seems rather arbitrary and is not well supported. Low cell viability (lines 805-806) or asynchronous infections could impact the ability to identify infection in some ciliated cells. In Fig 2f, is productive infection happening in all ciliated, or is it just a subset of cells? RNA-seq may not be enough to fully characterize infected cells.

The authors suggest that nasopharyngeal epithelial cells may be expressing SARS-CoV-2 antigen to CD4+ T cells via MHCII. However, their analysis is based on RNA-seq, but not protein evaluation. Also, what is functioning as costimulation in these cells?

A global note. The figure legends often do not contain sufficient details about the experiments being depicted. In many cases, it is difficult to tell which samples are being analyzed or which technical approach is being utilized. The Figure 3 legend is particularly difficult to decipher.

Figure 3 comments:

- Figs 3a and 3e is very difficult to tell which labels are referring to which colors. The authors may consider incorporating a figure legend, as they did in Fig 1, and removing these labels.
- The resolution is very low for the clonotype cluster sequences in Figs 3k and 3l, and they are difficult to read.

The authors did not test SARS-CoV-2 specificity of antibody response, yet they state, “Timing of production of antibodies is in line with B cell responses observed in vaccination studies,

suggesting that these antibody secreting B cells at day 10 after inoculation are SARS-CoV-2 specific.” The authors should consider performing neutralization and/or SARS-CoV-2 binding assays to determine quality and actual specificity of antibodies from these cells. Timing of the B cell response alone is not sufficient to determine specificity.

The authors rely heavily on data from sequencing for their studies. Overall, the paper would benefit from more functional analyses of the relevant cell populations. For example, ELISPOT assays could be used to validate activation of T cells in response to specific peptide stimuli.

Intracellular cytokine staining and flow cytometry could also be used to look for multifunctionality in these cells, which could be useful in the case that any

of the populations mentioned exhibit overlap.

Fig 4 comments

- In the Fig 4d legend, it should be made clear whether these are CD4 or CD8 epitopes.

- In Fig 4d, ORF1ab is extremely broad. Are the authors able to comment on which viral Nsps these epitopes map to?

Lines 441-443 – activated clonotype groups shared between patients. SARS-CoV-2-reactive T cell epitopes have been reported in individuals who were not exposed to SARS-CoV-2 (PMID 32753554). These were attributed to the development of potentially cross-reactive T cell responses in response to common cold coronaviruses. The presence of shared activated clonotypes identified in this study may therefore not be enough to determine antigen specificity against SARS-CoV-2. The authors should pursue more functional studies of T cells and/or validation of the specificity of these clonotypes.

Line 468 and throughout. “Challenge” is often used to refer to the 2nd inoculation in order to test immunity after initial exposure. Is this the correct terminology for this study?

Lines 469 and 475 “stuffy nose” is very colloquial. Would a term such as “nasal congestion” or “rhinitis” be more appropriate here?

Line 504 – The phrase, “the COVID-19 and broader infection diseases community,” is not clear.

The authors should elaborate on what implications their study might have for the “broader infection diseases community” and rephrase this sentence to be clearer.

Lines 506-508; 529-530 are overstated.

- It is not clear from the data whether the infection in transiently or abortively infected cohorts was prevented or never took. Will there be any follow-up on these cohorts to determine whether there is antigen-specific recall upon subsequent reinfection or vaccination?

- A correlation between MAIT cell activation and decreases in inflammatory monocytes, and SARS-CoV-2 infection is not necessarily a causation. The authors should revise this sentence or consider adding additional mechanistic studies of these cell populations.

- Consideration should be made of the innate immune response to any foreign object placed in the nose. From these data, it is not possible to conclude that this response is “new and distinct.”

Line 537. The authors suggest that the lymphatic system is the most likely route for activation of the antiviral response during infection. This statement does not take into consideration the fact that there are circulating immune cells at any given time in the blood, especially professional antigen-presenting cells, which are the most likely to produce the interferon response to infection. Please revise.

Line 543 is overstated. The authors offer no evidence that these cells are producing virus, only that they contain viral RNA.

The Discussion could benefit from mention of SARS-CoV-2 proteins that can inhibit the innate immune response, especially as it relates to IFN signaling delay at the site of infection.

The Fig 5 legend should be elaborated on in order to effectively describe the contents of the figure to a reader. The current legend is not specific or

descriptive enough.

Line 712 typo. "Of note after their were discharged from quarantine..." should be "they," not "their."

Line 850 typo. Immudex should be Immudex

Line 865 typo. "Microcentrifuge" should be "microcentrifuged"

Line 1034 please update Figure #s

Line 1076. There is an extra space after GpyTorch

Line 1109. There is an extra space after sequence

Line 1217 typo. "an" should be "a"

Line 1227 typo. Should be "there are no data"

Supplemental Fig 1 comments

- Could any of the transiently infected cohort results have been false positives? From this figure, it looks like only one test was taken at a given time point. The authors state that an FFA was not performed on these samples; however, that result could help to validate whether the positive was real.
- Generally, it would be nice to see a graph of CT values for these patients. Not only would that help in defining the kinetics of infection, but it could also provide more support to the authors' assertion that some of their cohort were indeed transiently infected.
- There is a typo next to the asterisk at the bottom of the figure. "Pateint" should be Patient.

Referee #2 (Remarks to the Author):

In their manuscript "Human SARS-CoV-2 challenge resolves local and systemic response dynamics" Lindeboom et al. describe in a time-resolved manner the local and systemic response to SARS-CoV-2. This well written manuscript provides exciting new insights into the early immune response following infection to SARS-CoV-2 and the dynamics of the infection. The major strength of this study is the opportunity to study the immune response against SARS-CoV-2 in adult participants that were not exposed to the virus and not vaccinated before. Together with the collection of nasal swabs and peripheral blood samples on the day before inoculation followed by a series of further samples afterwards and different single cell sequencing approaches they created a unique data set allowing to contribute substantial new knowledge to the field.

General comments

Following inoculation of a low SARS-CoV-2 dose, 6 out of 16 participant from the cohort developed a sustained infection with mild symptoms, three

individuals remained symptom-free, but showed sporadic positive PCR results (transient infection), and 7 participants obviously prevented infection at all and remained negative regarding virus PCR and symptoms (abortive infection). The authors describe the dynamic of the immune response in a very detailed manner for the group with sustained infection. Thus, the majority of the presented results is based on samples of only 6 participants and might be underpowered. Furthermore, the entire study appears to be more or less descriptive, without answering the most relevant questions. Data from the day -1 samples (before inoculation) were mainly used as baseline, without considering them as an important basis determining the outcome of the infection – whether an individual patient developed a sustained infection or was able to prevent it. Comparing the day-1 cell population distribution and gene expression profiles between the three groups would offer the unique opportunity to describe the specific immune status that may protect against SARS-CoV-2 infection. Unfortunately, such a comparison is missing so far, but have to be done.

A second major flaw is the so far not considered comparison between the sustained and transient groups regarding the early innate anti-viral response. Earlier published studies reported the relevance of the early innate immune response in particular in infected epithelial cells to clear the infection. This includes the expression of pattern recognition receptors able to bind the virus as well as the activation of down-stream targets to initiate an effective interferon-response. Figure 6a seems to give some evidence that in the transient group this anti-viral-response is developed very early in an appropriate manner which is potentially the reason that these participants were able to clear the infection without symptom development. Unfortunately, this really important result was not further followed in this manuscript. The early expression of the innate anti-viral response including the expression of the relevant pattern recognition receptors should be shown and compared between the three groups.

More detailed comments

In the introduction (line 65) it is stated that it was not possible so far to study the early phases of SARS-CoV-2 exposure in humans. In fact, there are earlier published studies providing first results on the early compared to the late response to SARS-CoV-2 infection (e.g. Loske et al. Nat Biotechnol 2022 Mar;40(3):319-324). The authors should discuss their results in comparison to earlier studies.

Six out of 16 participant inoculated with SARS-CoV-2 remained negative for virus PCR as well as any symptoms. How this result can be explained? Potential causes and mechanisms should be investigated based on the exciting data set available from the participants, described and discussed.

Line 119. The authors state that their data set contained all expected cell-types. However, neutrophils are missing. Neutrophils has been found in nasopharyngeal samples of healthy as well as SARS-CoV-2 infected individuals and were also described to play an important role in COVID-19. They should be present also in the nasopharyngeal samples and should be also considered in the data analyses. In the UMAP in Figure 1b neutrophils are missing, not clear why.

Figure 2

2a) How the interferon stimulation signature score was determined? Information is missing in the methods section and should be added. In Figure 2a the same dot plots should be added for transient infection (which is probably the more important information). I would expect that in ciliated and secretory cells the interferon stimulation signature is earlier and stronger in the transient group compared to the sustained group, which would be a quite important result. Figure 6a should show these plots for abortive samples. Currently, Figure 6a shows a comparison between the three groups, but based on the proportions of Interferon-stimulated cells. This is not comparable to the information provided in Figure 2a and it is difficult to distinguish the response in the different cell types. It seems that in particular in ciliated cells there is a strong early interferon response which could be one of the reasons for a quick virus elimination and prevention of sustained infection. Please describe also how the proportion of interferon-stimulated cells was determined.

2b) It seems to be not appropriate for focus only on sustained infection to study the early immune response to SARS-CoV-2 (see also line 168-170). As shown in Figure 1f and 6a, the earliest response occurred in the transient group and should be described more in detail or at least in comparison to the sustained group. To focus in particular on the early response, data from day one post-inoculation should be included for the nasopharyngeal samples

2c) Line 182-183 A significant decrease of inflammatory monocytes was observed in all groups – Fig. 2c shows an increase from day 3 on. This is in contrast to Figure 2b showing in fact a decrease in circulating myeloid cells from day 3 on and is very confusing. Not sure whether 2c is helpful at all.

2i) Lines 246-248: How these genes were selected? Most significant? Why genes involved in virus recognition and induction of an innate anti-virus response (such as DDX58, IFIH1, STAT1...) are not shown here? What about gene coding for chemokines that recruit immune cells? From the shown data in Figure 2i one could not be convinced that the hyper-infected ciliated cells show in fact an anti-inflammatory gene expression status. Furthermore, the genes shown in Extended Data Fig 6c do not support an attenuation of the interferon response in hyper-infected ciliated cells. To support this assumption a larger number of ISGs should be shown here.

2g) How the fraction of acute phase response gene (APR) positive cells was determined? Please describe this in the methods section. Which gene set was used for classification?

Line 246-248. Ciliated cells exhibit a unique response to high viral amounts. Here a set of genes is mentioned including ERG1, NFKB1A.... How these genes were selected (most significant once?)? What about the genes involved in the innate anti-viral response? What about chemokines involved in the recruitment of inflammatory immune cells? What makes you sure about the anti-inflammatory potential of these cells?

Line 268-271. Why a focus was set on APR+ and MHCII+ cells? Do these pathways were identified following enrichment analyses or differential gene expression analyses?

Figure 3b. Is this data from the sustained infection group only? If so, please explain this in the legend

Line 342-343. From Figure 3b it is not obvious that CD4+ T cells express high amounts of cytotoxicity genes, in particular PRF1. A comparison of the cytotoxicity gene expression should be shown (with significance) between day -1 and the later time points, as well as between the three groups.

It is speculated that the activation of cytotoxic CD4+ T cells is potentially a result of MHC class II presentation by infected ciliated cells. This is only a hypothesis. Without providing any supportive experimental data such a speculation should be avoided. I would suggest to remove this speculative part.

Figure 3j. Not clear which data are shown here. From the sustained group? Please add a more detailed description to the legend

Line 411-413. Activated T cells has been described already in earlier studies in nasopharyngeal samples. Please adapt the wording and cite this earlier studies.

Figure 4b. Please provide a definition of the activated T cells

Figure 4f. This Figure refers to the earlier published COVID-19 studies where the case numbers are provided in Extended Table 1d and e. However, the case numbers of different disease severities can not be found anywhere, please add. Furthermore, for the COVID-19 patient data, the information on the sampling time-point (days post infection) is missing. In general, the COVID-19 patient data set is poorly described

In general, a statistical analysis is completely missing. Which cell population significantly increase or decrease over time. Which gene/gene sets are differentially expressed over time? Significance data should be added to each of the Plots.

Discussion

The authors state in the discussion (line 505-508) that multiple response states have been detected that precede the onset of clinical manifestation and that this represents a newly discovered immune response that emerges when an individual is exposed to SARS-CoV-2, but manages to prevent the onset of viral spread. This is not correct. In fact, different response states are described very detailed for individuals with sustained infection. Information which response states prevent viral spread (transient and abortive groups) is completely missing.

Not clear which feature could be used as biomarkers (line 508 discussion). The results section does not provide data on potential biomarkers. For many different cell populations and gene sets an up or down has been described in a descriptive manner. No idea which of those feature could serve as biomarkers. I would suggest to remove this.

In the discussion it is further mentioned, that a major advantage of this challenge experiment is that also abortive and transient infections can be studied to unravel immune signatures preventing the infection. This would be in fact an important issue, but unfortunately is not really addressed in this manuscript.

Figure 5.1. Instead the time course of immune infiltration here the induction of the innate immune response both in immune and epithelial cells would be of interest.

In line 524-525 it is said that during sustained infections that lead to COVID-19, a novel acute phase response was observed in ciliated cells together with an activation of MAIT cells and a decrease of inflammatory monocytes in blood. This is exactly shown in Figure 5.2. but under the headline of “antiviral responses in non-sustained groups”. If this is really data from non-sustained groups, it is not clear to which data/Figures this summary refers to. At least one chapter should be added to the results part describing in detail the immune and epithelial responses in non-sustained groups. It is further mentioned, that these two responses (shown in 5.2) are the only immune responses that are also observed in participants with transient and abortive infections. Again, where in the manuscript this data is shown? How the authors have proved that these are the only responses in the non-sustained groups? How the comparison between the sustained and non-sustained groups has been done?

Figure 5.3 3 Three stages of the local ciliated cell response are mentioned here. Where in the manuscript these three stages and their time course are described?

Figure 5. Community contributions iii) What kind of vaccine targets and therapeutics have been identified? If none of the individuals has developed a severe disease course, would it be possible at all to suggest therapeutics based on the data presented here?

Line 534-538. It is further mentioned in the discussion that in sustained infection the activation of interferon signaling in blood precedes activation at the site of inoculation. According to the results presented in Figure 2a this seems to be not true. In this Figure the IFN stimulation occurred both in blood and nasopharyngeal cells at day 3 and started even earlier (day 1) in the nasopharyngeal cells.

Line 539. The relatively slow immune response at the site of infection is in contrast to the immediate immune infiltration observed in transient infection.

Again, this sentence seems to be incorrect. As Extended Data Figure 6a shows, there is a very fast interferon-response in transient infection.

Line 550. It is not only possible but have been already shown in many early studies that patients that require hospitalization exhibit an perturbed interferon response and an exacerbated inflammatory response. This sentence should be corrected accordingly. Furthermore I would suggest to discuss the limitation of being able to study only mild disease course a little bit more.

The participants have been inoculated with a very low virus dose. It should be discussed how representative the observed results are for an infection occurring under natural conditions with potentially much higher virus concentrations.

Furthermore, a comparison of the observed response dynamics under experimental conditions with the response pattern described in COVID-19 patients with more severe disease courses would be helpful.

Line 556. Activated conventional T cells have already been described in earlier single cell studies focusing on more severe COVID-19 course. I would suggest to remove this sentence.

The conclusion should be more cautious and should say, that this is a detailed time-resolved description of sustained infection with mild disease course.

Referee #3 (Remarks to the Author):

The manuscript by Lindeboom and collaborators outlines a description of the immune response to SARS-CoV-2 through scRNAseq, CITE-seq, TCR, and BCR sequencing. For this, investigators exposed healthy unvaccinated human participants to pre-Alpha-SARS-CoV-2 in a controlled environment with subsequent collection of longitudinal nasopharyngeal and blood samples. The manuscript describes local immune infiltration post-exposure, systemic induction of interferon response, decrease in the frequency of nasopharyngeal myeloid cells post-exposure in patients who developed a sustained infection, systemic activation of MAIT cells, SARS-CoV-2 ssRNA detection in various cell types including CD8+ T cells, restriction of SARS-CoV-2 replication to ciliated and goblet cells, MHC class II expression in infected epithelial cells, clonal expansion and activation of SARS-CoV-2-specific T cells, presence of non-

TRDV2/TRGV9-expressing g/d T cells, and clonal expansion of B cells. Additionally, the authors present the Cell2TCR approach for the detection of clonotype clustering.

The study offers a unique opportunity to analyze patients prior to exposure, in the days between exposure and development of symptoms and even in the first days of symptoms, as well as exposed patients who avoided sustained infection. These are all aspects of public interest and importance. The manuscript covers a wide dataset translating great investment and effort from the group and describes a broad array of more or less well-established immune processes. However, three years after the start of the pandemic, from the immune response description standpoint, the current presentation of the data brings limited uniquely novel advances and there is an opportunity to elevate this study to meet its potential. Additionally, choices in the organization of the manuscript and interpretation of the data need improvement. Still, the group's extensive effort and work to thoroughly provide longitudinal and single-cell level descriptions of host responses to SARS-CoV-2 infection is heroic.

Major concerns

1. One of the unexpected findings that the authors state is the "Widespread systemic interferon response precedes response at site of inoculation" (149) However, it appears that ISG responses are indeed starting in NP cells at day 3 also (Fig. 2a). Measuring IFN levels in the nasopharynx and plasma would be important to support these findings and claims. Also, why is day 1 missing from PBMC in Fig. 2a? This information is needed to support the authors' claim.
2. Related to the above, for the statement "immediate response to SARS-CoV-2 infection includes informing circulating immune cells before tissue-resident cells through interferon signaling" (162-163) - is it possible that local response is delayed in sustained infection instead?
3. Another unexpected finding is the presence of viral RNA genomes within CD8 T cells in the nasopharynx (Fig. 2e). Figure 1f shows that there is no increase in CD8 T cell number in any of the infected individuals within the nasopharynx at day 7. Does this mean that CD8 T cells that are resident in the NP are being infected? What do their TCR sequences look like – are they the ones identified to be specific to SARS-CoV-2 antigens? How about other transcripts – what type of CD8 T cells are being infected, and any hint of activation/cycling/cell death in the RNA signature? This is an important opportunity to learn more about the potential impact of SARS-CoV-2 infection on CD8 T cell function and fate.
4. Another major claim of this study is the reduction in circulating inflammatory monocytes in all exposed individuals, regardless of the sustained nature of the infection. However, they don't seem to be migrating into the nasopharynx to fight off infection (Fig. 2b). Where are they going?
5. Authors do not explore differences in viral load between patients. In a setting that stands out for the controlled initial viral load, there is a remarkable opportunity to analyze which baseline or early features correlate with lower or higher viral loads.
6. To me, the most interesting finding of this study is that about half of the exposed individuals do not get productively infected. What are the factors that determine this? The authors can mine information from their day -1 samples to examine this issue. Also, factors predictive of transient infection phenotype would be important to determine.
7. Fig 2i shows spearman correlation analysis of SARS-CoV-2 reads and gene expression – it does not show upregulation of the mentioned genes per se, as

implied in the text (245-248). “suggests that SARS-CoV-2 is capable of inducing” (248-249) In addition to not showing upregulation, the data also does not account for SARS-CoV-2 directly inducing changes in gene expressions.

8. While I understand that these human challenge studies were approved by the ethics committee, I wonder whether any of the subjects developed long-term sequelae. This is an important outcome that should be considered for future proposed studies. Please comment on this aspect.

9. Statistical analyses are lacking in many figures.

10. A detailed description of the participants’ demographics should be included.

Minor concerns

- Lack of clarity: Phrasing choices, clarity of data, and proper reference of previous work mentioned should be improved. (ex., “up to 100% of some cell types at a given time” (154),). Authors are not consistently clear in how statistical significance is defined throughout the different analysis. (ex. “highly significant” (308), “slight increase” (472)).
- Figure legends should always state patients and time points included, which statistical analysis was used, and what each dot represents.
- The manuscript does not adequately acknowledge the extent of previous work in the field. Authors claim that “our understanding of the cellular disease dynamics remains limited” (29), but there has been extensive work from naturally infected individuals. The authors should also reference previous work in “Studies by us and others” (49), “In contrast to previous studies” (213), “we and others” (284).
- Markers used for grouped analysis (ex. ISG score in Fig 2a) should be more clearly stated.
- The word “strikingly” is overused.
- Cell annotation process could be more clearly addressed – from Extended Figure 3, it seems like not all cells annotated as CD4+ express CD4 and that not all infected cells had detectable SARS-CoV-2.
- Please be more specific in the statements. “all immune cell types significantly infiltrate the site of inoculation after exposure to SARS-CoV-2” (132-133). However, abortive infection patients were also exposed to SARS-CoV-2 and did not display this pattern.
- Line 208 – Are all 2512 cells from the sustained infection group? Was there no detection of SARS-CoV-2 in cells from the other groups?
- Fig 2g. I found this figure difficult to understand. The text mentions 5 cell groups, yet there are only four in the figure. The mean values and statistical analysis are not clear. The information is difficult to follow overall.
- “when an individual is exposed to SARS-CoV-2, but manages to prevent the onset of viral spread” (507-508) Were these features not also observed in sustained infection patients?
- “Figure X, X, and X” (1034) should be corrected
- The authors show good commitment to transparency of reporting.

Author Rebuttals to Initial Comments:

Comments referee #1

Comment by the referee	Response by the authors
C1.1. This is a human challenge study designed to identify the local and systemic immune correlates of different mild outcomes of timed infection with pre-Alpha SARS-CoV-2. The authors used primarily single cell transcriptomics on cells obtained from nasopharyngeal swabs and blood samples to characterize immune response kinetics during infection, and created a detailed temporal atlas of these responses. Among their cohort of 16 participants, they identified three infection outcomes (sustained, transient and abortive) and carried out detailed and comprehensive analyses of cell-type specific activation and gene expression following exposure to SARS-CoV-2. In those characterized as having sustained infection, the authors demonstrated the following: - Interferon signaling can be detected in the blood prior to at the site of viral inoculation	R1.1. We thank the reviewer for the kind words and constructive feedback. Based on your comments, we have made key improvements to our manuscript such as addressing the generalizability issue that arose due to our cohort size by including data from an additional 20 challenged participants as a validation cohort, and from in vitro and patient samples, including flow and microscopy data. In addition, we have improved the discussion of existing literature, the flow and conciseness of our manuscript, clarity of our figures and legends, and we have included additional analyses on the observed cellular infections. Below, is a detailed response to the comments, which have significantly improved the quality of our manuscript.

- Delayed nasopharyngeal immune infiltration compared to those with transient infection
- Changes in nasopharyngeal resident and circulating myeloid compartments during infection, including a notable decrease in inflammatory monocytes across all infection cohorts
- Activation across all infection cohorts of MAIT cells expressing PRF1
- Ciliated cells in the nasopharynx supported productive infection, while local T cells and macrophages did not (although they were able to acquire viral genome)
- Infected epithelial cells may present SARS-CoV-2 antigen to CD4+ T cells via MHCII
- Activated CD4+ T cells expand by day 10 post-inoculation and are capable of expressing cytotoxic genes
- Activated B cells expand and mature between days 10-14 post-inoculation and are characterized by mostly IgG1- and IgA1-secreting plasmablasts

 - Identification of an atypical gamma-deltaTCR-expressing CD8+ T cell population - Regulatory T cell types associated with immune tolerance and resolution of infection expanded by day 14 with a strong upregulation of IL-10 	
C1.2. Despite several intriguing findings from this approach, the authors should take care not to overgeneralize their results, given the very small sample number evaluated in this study (16, with only 6 developing a sustained infection) and the fact that all participants were in a very young age bracket (18-29). The limitations of this study with regard to interpreting more severe outcomes of SARS-CoV-2 infection were nicely stated in the discussion. The paper could also benefit from a brief discussion of age-related dynamics in SARS-CoV-2 infection, as these results may be different in an older cohort.	R1.2. We thank the reviewer for the support for our approach, and understand that our sample size might seem limiting. In the revised manuscript, we have addressed the potential limitations of a small cohort size by including a validation bulk RNAseq dataset of 20 additional challenged individuals, and by performing experimental validation experiments, to confirm our key observations. Additionally, there are some unique aspects to our experimental setup that make it statistically highly robust compared to existing COVID-19 studies, which we have outlined in the last three paragraphs below. First, to validate our key findings, we have added a bulk RNAseq validation dataset that includes nasal swabs and blood samples of an additional 20 challenged individuals. This validation dataset is unique and highly complementary to our single cell genomics dataset: while the use of bulk instead of single-cell-resolved RNA-seq restricts us to only observing global dynamics over time, it did give us the opportunity to increase our temporal resolution to daily measurements post-inoculation. This not only allowed us to confirm our surprising observation that the interferon response in blood precedes the local interferon response at the site of inoculation, but it also allowed us to pinpoint the exact timing of these responses. Remarkably, Reviewer Figure R1.2.a and the revised Extended Data Figure 7h, reveal that a drastic activation of the interferon response in circulating immune cells happens between day 2 and 3 post-inoculation, while the local interferon response is more gradually activated and peaks between day 5 and day 7 post-inoculation. In addition, we used this newly acquired validation dataset to confirm our activated T cell observations and some of the new analyses that are included in this revision (Extended Figures 5h, 6h, 6k, 8g, 8h). Bulk RNA-seq allows us to capture global effects that affect most cell types,

although gene expression dynamics in relatively rare cell populations, such as MAIT cells, are hard to deconvolute from the ensemble measurements of bulk assays. Therefore, we sought to experimentally validate this observation. Whilst additional samples from seronegative challenged individuals are not available anymore and are impossible to generate, we were able to obtain *ex vivo* PBMC FACS data from PCR+ and PCR- health care workers at the time of PCR positivity vs. PCR negative tested concomitantly (COVIDsortium prospective health care worker study: Chandran et al. *Cell Rep Med* 2022: <https://doi.org/10.1016/j.xcrm.2022.100557>). Using this data, we were able to confirm our observation that MAIT cells show increased activation markers immediately after exposure (see **Reviewer Figure R1.2.b** and **Extended Data Fig. 5e,f**). Finally, our revised manuscript now includes *in vitro* data such as electron microscopy and immunofluorescence microscopy of air-liquid interface cultures (**Reviewer Figure R1.2.c, Fig. 2g, Extended Data Fig. 5b**) to validate the heterogeneity in viral infection of ciliated cells, and the activation of interferon signaling in infected cells.

Reviewer Figure R1.2.a: Linegraph validating and refining the relative expression dynamics over time since inoculation of the type I interferon signaling signature from Rosenheim et al. *bioRxiv* 2023 (<https://doi.org/10.1101/2023.06.01.23290819>) in sustained infection cases from our validation bulk RNA-seq datasets.

Reviewer Figure R1.2.b: Representative flow cytometry plots showing activation marker expression (Ki67, CD71, CD69 and HLA-DR) by mucosal associated invariant T cells (MAITs; gated as lymphocytes/single-cells/live-cells/CD3+/CD161++ TCR-Va7.2+) from one non-infected control (left; orange) and one SARS-CoV-2 infected individual at the time of the first positive PCR infection (right; black). Numbers indicate percent positive for each marker including double positives. (f) Summary data for single marker (left) or co-expression (right) of activation markers by peripheral MAIT cells from nine uninfected controls (open circles) and nine individuals with coincident infection (closed circles). p value shown for Mann-Whitney test. Bars, median. This has been added to **Extended Data Fig. 5e-f**.

Reviewer Figure R1.2.c. Transmission electron micrographs of SARS-CoV-2 infected and hyper-infected ciliated cells. Images were taken using differentiated nasal epithelium airway-liquid interface (ALI) *in vitro* cultures 72 hour post infection with SARS-CoV-2 virus. ALI cultures were generated from healthy adults and infected as previously described (Woodall et al. *bioRxiv* 2023: <https://doi.org/10.1101/2023.01.16.524211>). Viral particles were falsely coloured with red to aid visualisation here. Scale bar represents 1 μ m. Panels have been added to the main Figures (see **Figure 2g**) in the manuscript. Panels have been added to the **Extended Data Figure 9b**.

Regarding the size of our study, we would like to emphasise that in contrast to our cohort size (now expanded from 16 to 36), our sample size is very large and in fact much larger than many of the other single-cell-resolved COVID-19 studies that have been published. Our detailed longitudinal sampling, together with the multi-organ and multi-modal profiling, means that we analysed more than 181 samples, of which many were measured in replicate, and in total we sequenced more than 700 libraries for this study. We note that this has been one of the largest challenge studies on respiratory diseases that has been executed to date.

But most importantly, our experimental set-up of a challenge study is fundamentally different from the dozens of patient cohorts that have been published so far, and allows for much more accurate research even with a smaller cohort. Patient cohorts need to be stratified for clinical features that impact the observed immune response. This means that a patient cohort should include a few people with the same disease severity, age group, timepoint since-infection, ethnicity, standard-of-care, viral strain, viral load, infection history, serology, comorbidities, coinfections etc. If one divides even the size of the biggest patient cohorts by all possible combinations of categories, it is extremely difficult to control for these variables even with hundreds of participants. And many of these critical aspects are in fact often unknown and therefore impossible to stratify for. In contrast, the experimental setup of a challenge study completely controlled for all of these aspects, both in terms of strict admission criteria and screening, and in terms of sampling over the course of infection. Because we have such a homogenous participant population, a detailed multi-modal and multi-organ time-series, and together with a unique day -1 baseline sample to compare to, this makes our dataset clean and strong, even as compared to the largest patient cohorts described thus far. The proof for our statistical rigour is in the pudding, where even though it is 2023 now, we were able to discover fundamental novel insights into the immune response to COVID-19.

Because of the above mentioned reasons, we believe that our strict age limits are a key strength of our work, as it allows us to accurately control for the large impact that age has on immune composition and responses, as we and others have shown previously. As we outlined in our Discussion section of the text, we fully agree that caution should indeed be taken when extrapolating our results to children, elderly, and severe disease.

C1.3. The use of the word, “resolve,” in the title seems overly definitive, given this small cohort.	R1.3. We agree that the title may be overly definitive and have adapted to: “Human SARS-CoV-2 challenge uncovers local and systemic response dynamics”
C1.4. The authors should consider discussing how the dose used in this study (10 TCID50) compares with a physiologically relevant real-world exposure.	R1.4 The natural infection dose of SARS-CoV-2 remains unknown (Killingley et al. Nat Med 2022: https://doi.org/10.1038/s41591-022-01780-9). Early in the pandemic, a World Health Organization Advisory Group published expert consensus guidelines recommending a starting dose of 10² TCID50 (Levine et al. Clin Infect Dis 2020: https://doi.org/10.1093/cid/ciaa1290). Based on in vitro data in primary human airway epithelial cells and Syrian hamsters, the dosage for the Human COVID-19 Challenge study was started ten-fold lower than this at 10 TCID50 and was found sufficient to meet the 50–70% target infection rate. Compared to a prospective household contact study, a similar secondary attack rate of ~38% was observed, suggesting that the model may replicate a similar, if not slightly higher exposure than naturally acquired infections events (Singanayagam et al. Lancet Infect Dis 2022: https://doi.org/10.1016/S1473-3099(21)00648-4). We now discuss the relation between our experimental dose and natural infections in the main text of our revised manuscript.
C1.5 An internal control of baseline prior to infection is used for subjects in this study. However, this doesn’t account for innate immune responses that are generated simply by putting any foreign particle in your nose. Could the abortive infection group simply be people who responded to the stimulus in the absence of infection? Do PBMCs of transiently or abortively infected people exhibit any recall	R1.5a Whilst no placebo control challenge arm was included in this particular study, due to ethical constraints and extensive costs, previous non-COVID-19 human challenge studies such as Jha et al. J Allergy Clin Immunol 2021 (https://doi.org/10.1016/j.jaci.2020.07.012) reported no significant changes in the levels of nasal mediators in placebo controls. It is therefore unlikely that a significant immune response is mounted due to presence of a foreign particle in the nose. Furthermore, great care was taken when designing and manufacturing the dose inoculum virals, which were produced in a very clean manner in accordance with the current good manufacturing practices (cGMP) (Killingley et al. Nat Med 2022: https://doi.org/10.1038/s41591-022-01780-9). Individual dose inoculum vials were

immune response to SARS-CoV-2 antigen? How do the immune response dynamics during SARS-CoV-2 infection compare to those following intranasal vaccination?

generated through the dilution of a master virus bank with a sucrose diluent and underwent extensive quality testing (i.e. looking at identity, infectivity, and contaminant/adventitious agent tests) as part of the GMP manufacturing release process prior to challenges. Thus the responses observed are highly likely the result of the virus and not the vehicle or any contaminants within the vials.

Lastly, this is supported by the extent of which we see the same response within all participants across all 3 infection groups, not just locally, but also within the circulating immune compartment across all participants, for example the increase in circulating MAIT cells and reduction in inflammatory monocytes). This supports the idea that these immune responses are a product of the exposure to the virus upon challenge, rather than just simply people responding to stimuli in the absence of an infection.

R1.5b Question: Do we see any recall immune response to SARS-CoV-2 in the PMBCs of transient and abortive infections?

Within our PBMC dataset we do not see significant responses that would be indicative of a recall immune response to SARS-CoV-2 in the transient and abortive infections. Of note, we do detect T cell activation, in one of the transient infected participants, but no seroconversion. While we note this in our manuscript, we cannot draw any conclusions on this based on the sporadic nature of this observation.

R1.5c Question: How do the immune responses compare to intranasal vaccinations?

We think this is an interesting point, but unfortunately beyond the scope of this study and to the best of our knowledge has yet to be resolved in humans at the single cell level. When future studies involving the development of new mucosal vaccines on seropositive individuals challenged with SARS-CoV-2 (e.g. CEPI https://cepi.net/wp-content/uploads/2023/07/CfP_BetaCoVCHIM-call-text-2023-0711-final-for-publication-for-EC.pdf) produce new scRNAseq data, our study will provide the only available reference dataset for comparison

	to seronegative individuals.
C1.6 Measurement of viral load by PCR is limited because it measures the presence of viral genome without being a good indicator of infectious virus. The authors do show focus-forming assay results of subjects with sustained infection in the supplemental data. The paper would benefit from moving some discussion of this to the main text, as well as including FFA data from individuals who were transiently infected to help show that PCR positivity was not a false result.	R1.6 In this challenge study, only the participants that consecutively tested positive via qPCR were taken on to be further tested via FFA. Whilst we agree that the measurement of viral load via PCR does not necessarily translate to a good indication of infectious virus, qPCR is a much more sensitive method compared to FFA tests. With a good lower limit of detection, we are confident with our qPCR results, although we concede to the reviewer's point that we cannot comment on the "infectivity" within the transient infection group here. In addition, there are significant and clearly distinct transcriptional differences across all 3 infection groups.
C1.7 Additionally, an RNA-seq analysis at single cell resolution that includes data on viral genome or antigen expression would be helpful to better understand the cohorts depicted in Figure 1.	R1.7 We provide several analyses related to SARS-CoV-2 in this study. On the one hand, we have quantified the amount of viral reads, split by cell type and by time point, in Fig. 2d. Note that viral reads were only detected in cells from the sustained infection cohort and from nasopharyngeal swabs. This analysis showed that ciliated cells were preferentially infected by SARS-CoV-2, and a hyperinfected ciliated cell state could be distinguished (Fig. 2d and 2f). We also included analyses by electron microscopy that show that the infected and hyperinfected ciliated cell states can be distinguished (Fig 2g, Extended Figure 9b and). Expression of viral entry and related genes is now also shown (Extended Data Fig 6a). We then quantified which regions of the viral genome were preferentially expressed per cell type, allowing us to distinguish between productive and non-productive viral replication (Extended Data Fig 6b and 6d). This analysis showed that infected ciliated and goblet cells mostly expressed transcripts from the 3' end, where the structural proteins of SARS-CoV-2 required for virion production are located (Kim et al. Cell 2020: https://doi.org/10.1016/j.cell.2020.04.011), while this bias was absent in infected CD8+ T cells, secretory cells and macrophages, indicating that not all infected cell types can be hijacked for viral replication.

Finally, we address the issue of SARS-CoV-2 specificity by showing the time-restricted presence of activated T cells (and plasma B cells) with shared receptor motifs that are enriched in SARS-CoV-2 specific sequences. We were also able to assign viral epitopes to 40% of shared TCR motifs, showing that NSP3 is most often targeted by TCR motifs (22% of motifs), followed by Spike protein (8%, see **Reviewer Figure R1.7 below**). Furthermore, we performed dextramer experiments with SARS-CoV-2 peptides, where we showed a significant enrichment of activated T cell types across several peptides (**Extended Data Fig. 8b, 8c, 8d**). Taken together, these data support that SARS-CoV-2 infects and successfully replicates in several cell types of the nasopharynx in the sustained infection cohort and that a SARS-CoV-2 specific adaptive immune response targeting a subset of viral antigens is successfully mounted in those same individuals.

Reviewer Figure R1.7: Distribution of viral antigens for all shared, activated TCR motifs of the challenge and public dataset. Around 40% of motifs had a match with the SARS-CoV-2 specific database IEDB, with the most common targets being NSP3 and Spike protein.

C1.8. Figure 1 comments:

R1.8 Based on this comment, we have evaluated removing labels. However, we believe that these UMAPs become very hard to interpret without the on-figure labels, as similar colours can be hard to distinguish. As shown in

- Labels on panels b and d obscure visualization of the colors. May be better to use more general labels in the image to reduce amount of text, since legend is very clear

Reviewer Figure R1.8, this is especially an issue for the PBMC panels, where the number of cell types is larger. We hope this illustrates why we would like to keep the labels.

Reviewer Figure R1.8: UMAP showing PBMC cell types without labels on figure.

C1.9 - This figure is missing data that show evidence of infection in any of these cells (e.g. presence of SARS-CoV-2 N transcripts) – see above comment

R1.9 We have now included an additional figure in **Extended Data Figure 6b** and **Reviewer Figure R1.9**, in which we delineate the viral gene expression into individual viral genes across the infected cell clusters. This indeed shows that the expression of the N transcript is amongst the highest of the viral genes in different subsets.

	 Reviewer Figure R1.9: Dotplot showing the expression of viral genes in the identified cell types.
C1.10 Figure 2 comments: - Validation of the IFN response at the protein level would be helpful, as SARS-CoV-2 has numerous proteins that shut down translation of host transcripts. Therefore, even if an ISG is expressed at the RNA level, it may not ever be translated in an infected cell.	R1.10 There is a wealth of studies that have examined the interferon response to infection by SARS-CoV-2 that correlate robust cellular interferon responses with improved clinical outcomes (e.g. Hadjadj et al. Science 2020: https://doi.org/10.1126/science.abc6027). Our own group has also shown the induction of interferon response genes at the RNA level in epithelial cells is mirrored by upregulation of ISG at the protein level (Woodall et al. bioRxiv 2023: https://doi.org/10.1101/2023.01.16.524211). In addition, we now present in vitro immunofluorescence data to show that the interferon stimulated gene MX1 can be co-expressed with the viral spike protein (Extended Data Fig. 5b, Extended Data Fig. 9a), further supporting the notion that virally infected cells induce an interferon response at both RNA and protein levels.
C1.11 - Myeloid subsets (monocytes, activated DCs) reduced at site of inoculation at day 3 (trafficking to LNs) – also observed in transient & abortive infections	R1.11 We indeed also observe a significant decrease in nasopharyngeal monocytes in abortive (and sustained) infections, but not in transient infections (Extended Data Figure 7e-g). However, changes in the other nasopharyngeal myeloid cell types are not significant. We have noted this observation now in the revised text (Supplementary Note). Of note, because the interpretation of our observations in the myeloid compartment is speculative, and based on other reviewer comments to focus our manuscript on our key findings, we decided to relocate these discussions to the Supplementary Note that is now attached to the revised manuscript.

C1.12 - Do inflammatory monocytes in transient/abortive infection have same activation marker expression as those in sustained infection? Would be useful to have a flow cytometric characterization of these cells to more comprehensively evaluate markers.

R1.12 Due to participant welfare and ethical considerations, we were only able to obtain a single nasopharyngeal swab per participant per time point, which makes these samples very precious and hence we chose to perform single cell RNA-seq instead of flow cytometry, as it gives unbiased genome-wide insights regarding the transcription of all known and unknown markers at a cell-type and response-state resolved manner. Additionally, just like flow cytometric assays, our CITE-seq assay gives cell surface protein expression data of 130 cell type and activation markers. To address this question, we performed differential gene and protein expression analysis to compare inflammatory monocytes between our different infection groups. This analysis did not yield any significant changes between infection groups that are relevant for immune activation (**Reviewer Figure R1.12**). This is expected, because in our effort to annotate cell types and response states, we perform a very granular clustering based on transcriptome-wide (marker) gene and protein expression, and we use differential expression to identify any further subsets that could be functionally distinct and relevant. If there were any functional differences between abortive and sustained inflammatory monocytes, we would have expected these to form a distinct subcluster that we would have subsequently annotated as different.

Reviewer Figure R1.12: Differential expression analysis comparing inflammatory monocytes from sustained and

	abortive infections. In this comparison, we controlled for time point, sequencing library, sex and patient ID.
C1.13 - Fig 2e and 2f – the figure legend could use more information, as it does not clarify what sample is being analyzed (sustained infection? one participant or pooled data?)	R1.13 We agree that this was vague and have addressed this now. Throughout the figures, we now provide this information in either the figure labels or figure legends.
C1.14 It is surprising to learn that CD8+ T cells were infected in this study, albeit not productively. Please add references for the previous studies mentioned on this topic in line 213.	R1.14 We agree that this is a remarkable observation, especially considering the lack of viral entry genes on these cells (Extended Data Fig. 6a). While previous single cell RNA-seq studies have observed infected nasopharyngeal T cells, we could not find any previous publication that separated productive from unproductive infections in T cells. Post-mortem lung studies observed SARS-CoV-2 infection of T cells, and in vitro studies have shown SARS-CoV-2 is capable of infecting T cells in an ACE2-independent manner (Wang et al. Sig Transduct Target Ther 2020: https://doi.org/10.1038/s41392-020-00426-x ; Shen et al. Sig Transduct Target Ther 2022: https://doi.org/10.1038/s41392-022-00919-x ; Brunetti et al. Elife 2023: https://doi.org/10.7554/eLife.84790). We now included these references in the revised text.
C1.15 What is ACE2/TMPRSS2 expression on the cells that are infected? (ciliated, goblet, CD8, macrophages). This study could benefit from a flow cytometric analysis of these cells (ACE2, activation markers, ISG upregulation vs. SARS-CoV-2 antigen).	R1.15 Quantifying the expression of viral entry genes reveals that ciliated cells express the highest amount of viral entry genes (Extended Data Figure 6a, Reviewer Figure R1.15), which could explain their relatively high infection rate. Interestingly, infected secretory cells have higher expression of ACE2 compared to non-infected secretory cells. In contrast, ACE2 and TMPRSS2 expression cannot explain the observed SARS-CoV-2+ myeloid and T cells. This, together with the lack of productive infections (Extended Data Figure 6d), could suggest that the presence of virions in these immune cells reflect an antiviral engulfment to contain the viral spread. While this is expected behaviour of macrophages, it's not a typical characteristic of a T cell. Of note, SARS-CoV-2 infection, which is also dependent on ACE2 for entry, has also been found at pathological levels in T cells (Wang et al. Sig Transduct Target Ther 2020: https://doi.org/10.1038/s41392-020-00426-x ; Shen et al. Sig Transduct Target Ther 2022: https://doi.org/10.1038/s41392-022-00919-x ; Brunetti et al. Elife 2023: https://doi.org/10.7554/eLife.84790). In addition, others have discovered ACE2-independent SARS-CoV-2 infection of T cells (Shen et al. Sig Transduct Target Ther 2022: https://doi.org/10.1038/s41392-022-00919-x). While we refrain from excessive speculations

around this matter, we now discuss the viral entry genes in the revised text and extended data figures.

Reviewer Figure R1.15: Dotplot of viral entry genes, antiviral immune response markers, and viral RNA.

C1.16 The authors should consider rewording the heading, “Hyper-infected ciliated cells are the main source of SARS-CoV-2 and produce anti-inflammatory molecules,” and toning down the definitive language of lines 234-242.	R1.16 We agree that this is too definitive, and have reworded the heading (now: “Hyper-infection and anti-inflammatory response of ciliated cells”) and toned down the language here (now: “...uncovering a possible role for this subset of ciliated cells as major virion producers”).
C1.17 - It is difficult to determine whether ciliated cells are the “main source,” of virus when the only cells from the nasopharynx that were analyzed are those collected by a nasopharyngeal swab. - The authors have not shown any evidence that ciliated cells, despite being infected, are transmitting virus to other cells.	R1.17 As mentioned in C1.16, we have toned down the language and we agree with the reviewer that we have not shown any evidence that ciliated cells are actually transmitting virus to other cells. We do show that ciliated cells are susceptible to proliferative viral infection (Extended Data Figure 6d), and to further substantiate the role of hyper-infected ciliated cells in fueling the viral spread and disease, we have now performed a correlation analysis between the viral load detected by qPCR, and the relative amount of different nasopharyngeal cell types and subsets (Extended Data Figure 7i). This revealed that the hyper-infected ciliated cells are the only infected cellular subset that significantly correlates with viral load.
C1.18- The term “hyper-infected” seems rather arbitrary and is not well supported. Low cell viability (lines 805-806) or asynchronous infections could impact the ability to identify infection in some ciliated cells. In Fig 2f, is productive infection happening in all ciliated, or is it just a subset of cells? RNA-seq may not be enough to fully characterize infected cells. The authors suggest that nasopharyngeal epithelial cells may be expressing SARS-CoV-2 antigen to CD4+ T cells via MHCII. However,	R1.18 Based on the transcriptome profile, we performed stringent filtering of high-quality cells. Low cell viability and poor technical quality can be easily identified in single cell RNA-seq by a high relative contribution of mitochondrial reads, low UMI counts, low expression of cell type specific markers, and high amounts of background signal. As shown in Extended Data Figure 3b and 5g, hyper-infected ciliated cells have clearly distinct upregulation of cell type marker genes, including exclusive markers such as NFKBIA (Extended Data Figure 6c). In addition, their distinct expression of interferon, acute phase response markers, and MHC class II genes, together with their distinct clustering by leiden clustering, indicates that they are functionally distinct. In addition to their high quality, it is important to note that while a typical cell that is found to be infected contains ~10 viral reads, hyper-infected ciliated cells contain on average more than 1000 viral reads. Considering that cells only contain 1000-2000 unique reads, this is an incredibly high number that surpasses any other RNA found in a single cell. This also means that the difference between infected ciliated and hyper-infected ciliated cells is not just a mere gradual increase in viral reads, it is a step change that delineates a clearly distinct group with distinct markers. To further confirm the presence of a high degree of variability in the amount of infection of ciliated cells, we have performed

their analysis is based on RNA-seq, but not protein evaluation. Also, what is functioning as costimulation in these cells?

electron microscopy (**Figure 2g**). This indeed distinguished hyperinfected from infected ciliated cells, with the former harbouring many more viral particles compared to the latter. Asynchronous infections could indeed occur, but we don't think that the hyper-infected subset is particularly affected by this. The bias towards highly expressed viral genes (**Extended Data Figure 6d**) is a strong indication that most viral reads in these cells originate from productive infection events, and not just from large uptake of infecting virions.

While we cannot quantify abortive *versus* productive infections at single cell level, we can see that both productive infections occur in both the infected and hyper-infected subsets of ciliated cells. We have now clarified this in the main text.

We thank the reviewer for raising the interesting point about costimulation and for giving us the opportunity to discuss our findings in a broader context, as well as provide additional data to support our hypothesis. MHCII expression in airway epithelial cells has been previously described (Madisson et al. *Nat Genet* 2023; <https://doi.org/10.1038/s41588-022-01243-4>; Wosen et al. *Front Immunol* 2018; <https://doi.org/10.3389/fimmu.2018.02144>) and validated at the protein level (Shenoy et al. *Nat Commun* 2021; <https://doi.org/10.1038/s41467-021-26045-w>; Kaneko et al. *Clin Immunol* 2022; <https://doi.org/10.1016/j.clim.2022.108991>). In addition, MHCII+ epithelial cells were shown to co-cluster with CD4 T cells during COVID-19 (Kaneko et al. *Clin Immunol* 2022; <https://doi.org/10.1016/j.clim.2022.108991>). However, the precise interactions between these cell types remain insufficiently understood. Unfortunately, the restricted sample availability in our current study cohort did not allow us to generate further experimental validations (see R1.12). Therefore, we performed additional computational analysis to investigate cellular crosstalk between MHCII+ ciliated and activated CD4 T cells using CellPhoneDB (Efremova et al. *Nat Protocols* 2020; <https://doi.org/10.1038/s41596-020-0292-x>). Our analysis indeed revealed a co-stimulatory interaction module (**Reviewer Figure R1.18**), characterised by CD40-CD40LG, several members of the TNF superfamily, and other receptor-ligand pairs. It is tempting to speculate that these co-stimulatory interactions complement MHCII-dependent antigenic stimuli for activating CD4 T cells. However, resolving the functional role and physiological consequences of this cellular crosstalk is beyond the scope of this manuscript and will likely be addressed in future studies.

--	--

	Reviewer Figure R1.18: CellPhoneDB analysis predicts co-stimulatory interactions between MHCII+ ciliated cells and activated CD4 T cells.
C1.19 A global note. The figure legends often do not contain sufficient details about the experiments being depicted. In many cases, it is difficult to tell which samples are being analyzed or which technical approach is being utilized. The Figure 3 legend is particularly difficult to decipher.	R1.19 Thank you for pointing this out, and this has now been addressed throughout the figure legends.
C1.20 Figure 3 comments:  - Figs 3a and 3e is very difficult to tell which labels are referring to which colors. The authors may consider incorporating a figure legend, as they did in Fig 1, and removing these labels. - The resolution is very low for the clonotype cluster sequences in Figs 3k and 3l, and they are difficult to read. 	R1.20 Thank you for this comment, and please see our response to your comment above about Figure 1 (R1.8). We have evaluated how labeling and including legends affects the clarity of these UMAPs, and concluded that labeling makes the figure more interpretable than adding legends when the UMAPs contain many cell states. The message that these panels need to convey is that the activated T cell subsets present distinct cell clusters, which will be harder to show when replacing labels with legends. We have also reduced the number of panels in Fig. 3 to improve the clarity of this figure, and we increased the font sizes of the cluster sequences. Final figures will be high resolution.
C1.21 The authors did not test SARS-CoV-2 specificity of antibody response, yet they state, "Timing of production of antibodies is in line with B cell responses observed in vaccination studies,	R1.21 Thank you for your comment, and serum for these participants was indeed collected and the antibody titre analysed at several time points, both pre and post SARS-CoV-2 inoculation (see Reviewer Figure R1.21 below). We have now included this data as part of the Extended Data Table 1p and as well as a column within the summary study metadata table in Extended Data Table 1g to help support our claim as well as prove the reader with more information.

suggesting that these antibody secreting B cells at day 10 after inoculation are SARS-CoV-2 specific.” The authors should consider performing neutralization and/or SARS-CoV-2 binding assays to determine quality and actual specificity of antibodies from these cells. Timing of the B cell response alone is not sufficient to determine specificity.

Here the antibody titre was measured by two means as suggested by the reviewer and as described in Killingley et al. *Nat Med* 2022 (<https://doi.org/10.1038/s41591-022-01780-9>). The SARS-CoV-2 anti-spike IgG concentrations were determined via ELISA (via Nexelis) and reported as ELU ml⁻¹ (see the top panel of **Reviewer Figure R1.21**). Neutralizing antibody titers for live SARS-CoV-2 virus (lineage Victoria/01/2020) were determined by microneutralization assay at the UK Health Security Agency and reported as the 50% neutralizing antibody titer (NT50) (see the bottom panel of **Reviewer Figure R1.21**).

Here we can see an increase in the SARS-CoV-2 antibody levels in the serum of the sustained infection group only >14 days post-inoculation. This is in line with the temporal B cell response observed within our PBMC single cell dataset, which was restricted to the sustained infection group and observed around day 10. We believe this helps further support our claim that this response is specific to SARS-CoV-2.

Review Figure R1.21. Antibody titers as measured from the participants serum up to 28 days post-inoculation. Graphs showing the SARS-CoV-2 anti-spike IgG concentrations (top) and the neutralising antibody titre (bottom) collected via ELISA (ELU ml⁻¹) and a microneutralization assay (MNA) (reported as 50% neutralizing antibody titre (NT50)) respectively. The lower limit of detection (LLOD) for each as is shown via the purple dotted line; 58 for the microneutralization assay and 50.2 ELU ml⁻¹ for the spike protein IgG ELISA. N = 16 (6 Sustained infections, 7 Abortive infections and 3 Transient infections). Data shown at Day -2 (pre-inoculation), Day 14 and Day 28. Median, IQR. Of note: one of the participants within the Transient infection group reported a community SARS-CoV-2 infection ~day 28 following the initial study quarantine. This was reported both in the method section of the paper and Extended Figure 1a. This may account for the slight elevation seen here in the Transient group at day 28.

C1.22 The authors rely heavily on data from sequencing for their studies. Overall, the paper would benefit from more functional analyses of

R1.22 As explained in R1.12, our unique study and experimental design makes us restricted in the amount of samples that we can obtain, and the amount of cells that we obtain per sample. Because of this, it would be a missed opportunity to focus on individual biomolecules and markers, and pushed us towards using the absolute

the relevant cell populations. For example, ELISPOT assays could be used to validate activation of T cells in response to specific peptide stimuli.	state-of-the-art that would allow us to obtain an as-complete-as-possible picture of the cell type resolved immune responses over time. Our revised manuscript now includes multiple experimental assays to validate our single-cell-transcriptomics-based findings, including FACS, electron microscopy, immunofluorescence, and a validation cohort of an additional 20 participants. In addition, as a reminder, our original and revised manuscript includes specific peptide pull down data using Dextramers that validates the SARS-CoV-2 specificity of the identified activated T cell clones (Extended Data Figure 8).
C1.23 Intracellular cytokine staining and flow cytometry could also be used to look for multifunctionality in these cells, which could be useful in the case that any of the populations mentioned exhibit overlap.	R1.23 As explained in R1.12 and R1.22, due to participant welfare and ethical concerns, we can only obtain small amounts of blood and nasopharyngeal samples per participant per time point, which makes these samples very precious and limits us in performing targeted assays to measure a few cell type markers by flow cytometry. Because of this we chose to perform single cell RNA-seq instead, as it gives unbiased genome-wide insights in the RNA production of all known and unknown markers at a cell-type and response-state resolved manner, including all expressed cytokines. This indeed gives us a unique possibility to study the functionality of these activated T cells in great detail. We revisited our original analysis to focus on cytokine expression of all 297 cytokines that are annotated in KEGG (hsa04060), and Reviewer Figure R1.23 shows all cytokines that were expressed in at least 1% of a conventional T cell subset. This reveals cytokine expression patterns as expected. Interestingly, type II interferon, TRAIL, and CD70 (CD27 ligand) production is increased in the activated subsets, further underscoring their activated and anti-viral phenotype. To aid the further interrogation of our novel cell states, we now also provide the results from differential gene expression analysis of all genes and all cell states in Extended Data Table 1.

	Reviewer Figure R1.23: Cell state specific expression of cytokines from gene set hsa04060 in conventional T cell subsets. Only cytokines that are expressed in at least 1% of at least 1 cell state are shown.
C1.24 Fig 4 comments  - In the Fig 4d legend, it should be made clear whether these are CD4 or CD8 epitopes. - In Fig 4d, ORF1ab is extremely broad. Are the authors able to comment on which viral Nsps these epitopes map to? 	R1.24 This is now incorporated in the new Fig. 4d via column MHC_class of the revised clustermap. Furthermore, the targeted antigen is now more detailed. With regard to the second points, this is now incorporated in the new Fig. 4d, with ORF1ab split into NSPs. When filtering down to TCR motifs shared by 5 individuals or more, NSP3 is the only one of the ORF1ab NSPs that is found across motifs.
C1.25 Lines 441-443 – activated clonotype groups shared between patients. SARS-CoV-2-reactive T cell epitopes have been reported in individuals who were not exposed to SARS-CoV-2 (PMID 32753554). These were attributed to the development of potentially cross-reactive T	R1.25 To address this point, we have performed a dextramer assay on the blood samples, which showed an enrichment of activated T cells (Extended Data Fig. 8b,e). We furthermore intersected the TCR motifs with SARS-CoV-2 specific TCRs from a database and again showed a significant increase in activated T cells compared to other T cell compartments (Extended Data Fig. 6n), and were able to assign a specific SARS-CoV-2 epitope to several of the shared TCR motifs (Fig. 4d, Extended Data Table 1f).

cell responses in response to common cold coronaviruses. The presence of shared activated clonotypes identified in this study may therefore not be enough to determine antigen specificity against SARS-CoV-2. The authors should pursue more functional studies of T cells and/or validation of the specificity of these clonotypes.	
C1.26 Line 468 and throughout. “Challenge” is often used to refer to the 2nd inoculation in order to test immunity after initial exposure. Is this the correct terminology for this study?	R1.26 Yes, a challenge study is a standard term for a clinical trial involving the intentional exposure to a disease-causing agent. We use the term ‘challenge’ as described in previous literature referring to the current human challenge study (Killingley et al. Nat Med 2022: https://doi.org/10.1038/s41591-022-01780-9), and believe this terminology to be sufficiently explanatory for our use.
C1.27 Lines 469 and 475 “stuffy nose” is very colloquial. Would a term such as “nasal congestion” or “rhinitis” be more appropriate here?	R1.27 Thank you for highlighting this. We have now replaced ‘stuffy nose’ with ‘nasal congestion’ in the revised main text.
C1.28 Line 504 – The phrase, “the COVID-19 and broader infection diseases community,” is not clear. The authors should elaborate on what implications their study might have for the “broader infection diseases community” and rephrase this sentence to be clearer.	R1.28 We have now removed this sentence and elaborate on community contributions in Figure 5.

C1.29 Lines 506-508; 529-530 are overstated.	R1.29 Line 506-508 reads [We detect..multiple response states.], “including the activation of MAIT cells and decreases in inflammatory monocytes. Importantly, this represents a newly discovered immune response that emerges when an individual is exposed to SARS-CoV-2 but manages to prevent the onset of viral spread.” The increase in MAIT cells and decrease in monocytes across response groups is clearly shown in Fig 2b-c. While others have described involvement of MAIT cells in COVID-19 patients (e.g. Flament et al. Nat Immunol 2021: https://doi.org/10.1038/s41590-021-00870-z), we are not aware that this response has been described before in individuals who do not develop infection, and will therefore keep this wording.
C1.30 - It is not clear from the data whether the infection in transiently or abortively infected cohorts was prevented or never took. Will there be any follow-up on these cohorts to determine whether there is antigen-specific recall upon subsequent reinfection or vaccination?	R1.30 This data was not collected as individuals from our cohort would have been subsequently naturally infected or been vaccinated since the challenge study, and this would make it impossible to distinguish whether the presence of antigen-specific T cells is due to previous immunity, the challenge itself or a later infection/vaccination. While this type of human recall study for common infections is therefore hard to do in practice, in future, vaccine studies may be amenable to study clonotype expansion and recall, and this is an interesting suggestion in the context of vaccines for rare disease (e.g. Dengue fever vaccination of Western participants).
C1.31- A correlation between MAIT cell activation and decreases in inflammatory monocytes, and SARS-CoV-2 infection is not necessarily a causation. The authors should revise this sentence or consider adding additional mechanistic studies of these cell populations.	R1.31 In human viral challenge studies, observed immune responses relative to the time prior to infection (d-1) are considered to be caused by the infection across many published studies. Hence we believe that our wording and interpretation is in keeping with the current literature. We have also added additional data from an independent cohort, to strengthen the link between SARS-CoV-2 infection and MAIT cell activation (Extended Figure 5 e,f). Furthermore, the results of our symptomatic patients are in line with multiple published studies. e.g. Flament et al. Nat Immunol 2021: https://doi.org/10.1038/s41590-021-00870-z (MAIT), e.g. reviewed in Knoll et al. Front Immunol 2021: https://doi.org/10.3389/fimmu.2021.720109 (monocytes).

C1.32 - Consideration should be made of the innate immune response to any foreign object placed in the nose. From these data, it is not possible to conclude that this response is “new and distinct.”	R1.32 Please refer to R1.5a for a response to this point. Whilst no placebo control challenge arm was included in this particular study, due to ethical constraints and extensive costs, previous non-COVID-19 human challenge studies such as Jha et al. J Allergy Clin Immunol 2021 (https://doi.org/10.1016/j.jaci.2020.07.012) reported no significant changes in the levels of nasal mediators in placebo controls. We therefore do not believe a significant immune response is likely simply due to the presence of a foreign particle in the nose.
C1.33 Line 537. The authors suggest that the lymphatic system is the most likely route for activation of the antiviral response during infection. This statement does not take into consideration the fact that there are circulating immune cells at any given time in the blood, especially professional antigen-presenting cells, which are the most likely to produce the interferon response to infection. Please revise.	R1.33 Circulating immune cells, especially professional antigen-presenting cells, would indeed be possible candidates for the quick activation of the antiviral response. However, it is important to note that because we profiled the transcriptomes of PBMCs in a single cell resolved manner, we had in-depth insights in the interferon production of circulating immune cells. Despite the strong activation and very early temporal dynamics, we do not detect any changes in interferon production by any of the circulating immune populations that could explain this finding (Reviewer Figure R1.33).

	Reviewer Figure R1.33: Interferon mRNA production by nasopharyngeal cells (left) and PBMCs (right). Reads from all interferon type 1 genes (top) or from all interferon type 3 genes (bottom) have been aggregated to make this dotplot. Color scale represents the mean (aggregate) log1p expression value within each cell type at each time point and infection group. The size of each dot represents the percentage of cells with detectable expression.
C1.34 Line 543 is overstated. The authors offer no evidence that these cells are producing virus, only that they contain viral RNA.	R1.34 Line 543 reads: “Here, we found that a small subset of hyper-infected ciliated cells becomes anti-inflammatory and the main source of viral production.” Fig. 2f clearly shows the distinction between cell types containing low viral reads and hyperinfected ciliated cells with >100-fold higher levels of viral reads. Extended Data Fig. 6c shows that this subfraction of cells expresses a distinct anti-inflammatory gene set (e.g. NKFB inhibitor and others). We now quantitate the total fraction of viral reads in hyper-infected ciliated cells (67% of all detectable viral RNA) and also provide microscopy evidence that in vitro-infected ciliated cells give rise to both low level and high level virion-producing cells (Fig. 2g, Extended Data Fig. 9b). This statement in the Discussion section is thus well supported.
C1.35 The Discussion could benefit from mention of SARS-CoV-2 proteins that can inhibit the innate immune response, especially as it relates to IFN signaling delay at the site of infection.	R1.35 The immune suppressive effects of SARS-CoV-2 are already well reviewed in the literature (Minkoff et al. Nat Rev Microbiol 2023: https://doi.org/10.1038/s41579-022-00839-1). None of our results specifically relate to viral inhibition of the innate immune response, and therefore are unable to shed light on this important area of COVID19 research.
C1.36 The Fig 5 legend should be elaborated on in order to effectively describe the contents of the figure to a reader. The current legend is not specific or descriptive enough.	R1.36 Thank you for this comment; we have expanded this legend.

C1.37 Line 712 typo. “Of note after their were discharged from quarantine...” should be “they,” not “their.” Line 850 typo. Immundex should be Immudex Line 865 typo. “Microcentrifuge” should be “microcentrifuged” Line 1034 please update Figure #s Line 1076. There is an extra space after GpyTorch Line 1109. There is an extra space after sequence Line 1217 typo. “an” should be “a” Line 1227 typo. Should be “there are no data”	R1.37 Thank you for pointing out these errors, which we have now corrected in the main text.
C1.38 Supplemental Fig 1 comments - Could any of the transiently infected cohort results have been false positives? From this figure, it looks like only one test was taken at a given time point. The authors state that an FFA was not performed on these samples; however, that result could help to validate whether the positive was real.	R1.38 As mentioned in R1.6 above, unfortunately, only the participants that consecutively tested positive via qPCR were taken on to be further tested via FFA. Whilst we agree that the measurement of viral load via PCR does not necessarily translate to a good indication of infectious virus and it is always most ideal to have both, qPCR is still one of the most sensitive assays we have for the detection of viral load (with FFA test known to be a less sensitive assay in comparison). With a good lower limit of detection, our qPCR results are thus reliable. We do agree with the reviewer’s point that we cannot comment on the “infectivity” within the transient infection group here. It is worth noting that there are significant and clearly distinct transcriptional differences across all 3 infection groups. Participants were qPCR tested twice per day (using both nasal and throat swabs). These tests were conducted

	from day -1 (pre-challenge) right up until the day the participant either tested consecutively negative or left quarantine (as highlighted by the * and ♦ symbol in Extended Data Figure 1b and described in the Virology method section). To help make this clearer, we have added “(daily)” to the figure legend and re-word the text in the Virology method section to read: “Daily samples, were taken from both nose and throat (pharyngeal) in order to assess and quantify the viral kinetics of each participant pre- and post-inoculation.” For the reason stated above, in combination with the fact that multiple standalone positive qPCR tests were seen within ⅓ of these transient participants (see Extended Figure 1b and Extended Data Table 1 a-b) leads us to believe these results are not false positives. This is furthermore supported by the fact we see defined immune responses within these transient participants within our single cell datasets.
C1.39 - Generally, it would be nice to see a graph of CT values for these patients. Not only would that help in defining the kinetics of infection, but it could also provide more support to the authors’ assertion that some of their cohort were indeed transiently infected.	R1.39 We have added CT values to Extended Figure 1b as requested. The data from both the PCR and FFA assays can also be found in Extended Data Table 1a,b,h and i.
- There is a typo next to the asterisk at the bottom of the figure. “Pateint” should be Patient.	R1.40 This has now been corrected. Many thanks.

Comments referee #2

Comment by the referee	Response by the authors
C2.1 In their manuscript “Human SARS-CoV-2 challenge resolves local and systemic response dynamics” Lindeboom et al. describe in a time-resolved manner the local and systemic response to SARS-CoV-2. This well written manuscript provides exciting new insights into the early immune response following infection to SARS-CoV-2 and the dynamics of the infection. The major strength of this study is the opportunity to study the immune response against SARS-CoV-2 in adult participants that were not exposed to the virus and not vaccinated before. Together with the collection of nasal swaps and peripheral blood samples on the day before inoculation followed by a series of further samples afterwards and different single cell sequencing approaches they created a unique data set allowing to contribute substantial new knowledge to the field.	R2.1 We thank the reviewer for the kind words and enthusiasm, and for highlighting that our study is highly unique and contributes substantial new knowledge to the field. Based on your insightful and constructive feedback, we have made major improvements to our manuscript: ● We now include a large validation cohort of 20 challenged participants to increase our sample size from 16 to 36 (Extended Data Figures 5h, 6h, 6k, 7h, 8g, 8h).● We provide additional patient- (Extended Data Fig. 5e,f) and in vitro derived (Fig. 2g, Extended Data Fig. 5b) validation data.● Another key improvement made based on your comments is our expanded analysis of the pre-infection correlates with infection outcome, which revealed a new biomarker (HLA-DQA2) predictive for preventing the onset of a sustained infection (Fig 2h-j, Extended Data Fig. 6f-h).● In addition, we have improved the discussion of existing literature, the flow and conciseness of our manuscript, clarity of our figures, legends and statistics used. Below, you can find our detailed response to your comments, which has significantly improved the quality of our manuscript.
C2.2 General comments Following inoculation of a low SARS-CoV-2 dose, 6 out of 16 participant from the cohort developed a sustained infection with mild symptoms, three	R2.2 Please refer to the R1.2 response above.

individuals remained symptom-free, but showed sporadic positive PCR results (transient infection), and 7 participants obviously prevented infection at all and remained negative regarding virus PCR and symptoms (abortive infection). The authors describe the dynamic of the immune response in a very detailed manner for the group with sustained infection. Thus, the majority of the presented results is based on samples of only 6 participants and might be underpowered.

C2.3 Furthermore, the entire study appears to be more or less descriptive, without answering the most relevant questions. Data from the day -1 samples (before inoculation) were mainly used as baseline, without considering them as an important basis determining the outcome of the infection – whether an individual patient developed a sustained infection or was able to prevent it. Comparing the day-1 cell population distribution and gene expression profiles between the three groups would offer the unique opportunity to describe the specific immune status that may protect against SARS-CoV-2 infection. Unfortunately, such a comparison is missing so far, but have to be done.

R2.3 We thank the reviewer for this great suggestion. As suggested, we performed an in-depth differential gene expression analysis comparing the infection groups at the preinfection time point (day -1) and at our earliest time points post-inoculation. The aim of this analysis was to find genes whose expression was significantly associated with the subsequent disease outcome of the SARS-CoV-2 challenge. This analysis yielded very interesting results where we found that the baseline expression of HLA-DQA2, a non-polymorphic non-canonical subunit of MHC class II, is higher in both the circulating and nasopharyngeal professional-antigen-presenting cells of participants that become abortively or transiently infected, compared to sustained infection cases that become COVID-19 patients (**Figure 2h, Extended Data Figure 6e**). This indicates that HLA-DQA2 expression is a biomarker for protection against productive SARS-CoV-2 infections. Importantly, this represents the first expression-based biomarker for protection against SARS-CoV-2 infection. HLA-DQA2 expression during SARS-CoV-2 infection has previously been associated with milder COVID-19 by a number of studies, which we can validate with additional analyses of COVID-19 datasets (**Reviewer Figure R2.3.a**), which strengthens our observation and indicates that its expression is not only associated with protection against severe COVID-19, but also with protection against the onset of COVID19.

Reviewer Figure R2.3.a: HLA-DQA2 expression in COVID-19 patient data is significantly higher in mild and moderate cases compared to severe and critical cases for four of the five studies. Monocyte HLA-DQA2 log-normalised expression for 5 COVID-19 datasets, stratified by severity for each dataset. Significance determined with Mann-Whitney test.

In addition, we have validated the biomarker potential in our new bulk-RNA-seq based validation cohort (**Extended Data Figure 6h**), and we built cross-validated predictive models of disease outcome based on HLA-DQA2 expression in circulating monocytes (**Extended Data Figure 6f-g**). We have included these exciting new results in the main text and figures, which we've also copied below for clarity:

HLA-DQA2 as possible protective biomarker

Inspired by the identified and potentially protective responses in the non-sustained infection cases immediately after inoculation, we next set out to identify genes whose pre-infection expression levels could predict disease outcome. Interestingly, at the day before viral inoculation, we found that HLA-DQA2 expression in blood immune cells was significantly higher in participants in whom the virus did not succeed in establishing a sustained infection (Fig. 2h-j). HLA-DQA2 is a poorly characterized non-polymorphic class II MHC molecule whose increased expression in blood has been associated with milder COVID-19 progression. Our data suggest that HLA-DQA2 can act as a biomarker for protection against productive SARS-CoV-2 infections, which we confirm using cross-validation and additionally in our independent validation cohort (Extended Data Fig. 6f-h). This is, to our knowledge, the first gene expression-derived predictive biomarker that is not based on acquired immunological memory.

In addition to gene expression based analyses, we also investigated if we could find any pre-existing T cell immune repertoire characteristics that associated with disease outcome. This analysis was initiated based on the hypothesis that pre-existing T cell clones that exhibit cross reactivity with SARS-CoV-2 epitopes can facilitate protection in seronegative people, which is something that others have seen before. To this end, we made use of our deep-sequenced bulk TCR data collected at the day before inoculation, which we intersected with known SARS-CoV-2 specific TCRs and with our newly defined SARS-CoV-2 activated T cell clonotypes. This revealed that there is indeed a tendency towards more pre-existing T cell clones that can potentially recognize SARS-CoV-2 epitopes in the participants with transient or abortive infections (**Reviewer Figure R2.3.b**).

This indicates that protection and disease outcome could be influenced by both cell intrinsic factors such as HLA-DQA2 expression and by pre-existing adaptive immunity as shown here and by others (in addition to protective or predisposing genetics).

Reviewer Figure R2.3.b: Inspection for pre-existing immunity. Analysis of bulk TCR beta chain sequencing data, separated into PCR positive (n=18) and PCR negative samples (n=17) at baseline (day -1) shows that PCR positive individuals (n=18) have significantly fewer TCRs in bulk matching IEDB compared to PCR negative individuals (n=17, $p < 0.1$). Fraction of sequences matching SARS-CoV-2 specific beta chain TCR sequences from IEDB is shown, normalised by total duplicate count for each sample. Significance determined with Mann-Whitney test.

However, the difference in TCR overlap was small and only significant for the beta chain when intersected with activated T cell clonotypes (**Reviewer Figure R2.3.c**), which is why we chose to not include it in the revised manuscript.

Reviewer Figure R2.3.c: Analysis of bulk TCR alpha and beta chain sequencing data, separated into PCR positive (n=18) and PCR negative samples (n=17) at baseline (day -1) shows that PCR positive individuals have significantly fewer TCRs in bulk matching activated T cells compared to PCR negative individuals ($p < 0.005$), but only when considering the TCR beta chain. Fraction of sequences for each sample matching TCR motifs of activated T cells from public COVID-19 patient data is shown. Significance determined with Mann-Whitney test, only significant for TCR beta chain.

C2.4 A second major flaw is the so far not considered comparison between the sustained and transient groups regarding the early innate anti-viral response. Earlier published studies reported the relevance of the early innate immune response in particular in infected epithelial cells to clear the infection. This includes the expression of pattern recognition receptors able to bind the virus as well as the activation of down-stream targets to initiate an effective interferon-response. Figure 6a seems to give some evidence that in the transient group this anti-viral-response is developed very early in an

R2.4 We agree that a thorough comparison of the (early) immune responses between the infection groups is important to do. It is important to note that we had already done such comparisons in the previous submission, which yielded robust and interpretable results about immediate nasopharyngeal infiltration, MAIT activation, and migration of inflammatory monocytes. In our revised manuscript, we now also include HLA-DQA2 as a powerful biomarker associated with transient and abortive infections as described above (R2.3), and we furthermore explicitly clarify that the detectable (novel) early innate antiviral responses are restricted to these four novel observations. In addition, we have now added detailed differential gene expression analyses to our manuscript (**Fig. 2h, Extended Data Fig. 6e**, all DEGs shown in **Extended Data Table 1n-o**), which will enable readers to investigate individual markers and responses over time and between infection groups.

In our revisited analyses of the dynamics between infection groups, we also included a specific

appropriate manner which is potentially the reason that these participants were able to clear the infection without symptom development. Unfortunately, this really important result was not further followed in this manuscript. The early expression of the innate anti-viral response including the expression of the relevant pattern recognition receptors should be shown and compared between the three groups.

analysis on dynamic PRR expression as suggested by the reviewer. Many PRRs are part of the IFN response, and for these we do observe a strong upregulation in sustained infection cases as expected. However, we do not detect any PRRs with unique dynamics in the abortive or transient groups. In **Reviewer Figure R2.4**, we highlight the key PRR proteins involved in the detection of SARS-CoV-2 infection (based on a review by Kumar et al. *NPG Asia Mater* 2021: <https://doi.org/10.1038/s41427-020-00275-8>), and show their expression dynamics across time since inoculation and between infection groups. Similarly to the genome-wide analysis, this subset highlights interferon responsive RIG-I and MDA-5 expression that is specifically upregulated in the sustained infection cases during the later phase of the infection. Importantly, none of these key PRRs appear to be specifically expressed in transient or abortive infection cases and can explain the different outcome of the disease.

	Reviewer Figure R2.4: Expression dynamics of key PRRs involved in detecting SARS-CoV-2 infection across time since inoculation and across infection groups. Mean expression represents the mean log_{1p} expression. The reviewer's observation that former Extended Data Figure 6a indeed hints towards an earlier local IFN response that might be associated with the disease progression is well spotted. We noticed this as well before submission. However, the increase in IFN stimulation at day 1 post-inoculation is not robust as it is only observed in one participant, and we therefore did not further discuss this or make claims about it. The validation data that is now included in our revised manuscript further supports the sporadic nature of this observation.
C2.5 More detailed comments In the introduction (line 65) it is stated that it was not possible so far to study the early phases of SARS-CoV-2 exposure in humans. In fact, there are earlier published studies providing first results on the early compared to the late response to SARS-CoV-2 infection (e.g. Loske et al. Nat Biotechnol 2022 Mar;40(3):319-324). The authors should discuss their results in comparison to earlier studies.	R2.5 Loske et al. Nat Biotechnol 2022 (https://doi.org/10.1038/s41587-021-01037-9) offers valuable insight into the early versus late phase of SARS-CoV-2 infection, which we now cite in our revised manuscript. However, like all of the studies which rely on collecting samples from natural infections, it is limited by the estimated time of infection based upon reported symptoms. For example in the paper the authors define an early infection phase as 0-4 days post-onset of symptoms (dps) and a late infection phase 5–12 dps, with a small group of asymptomatic patients (n=2) collected 0-2 days post first PCR (dpPCR). Included below is a summary of Loske et al. cohort characteristics: SARS-CoV-2 children; Mean dps: 4.5 ± 2.8 and Mean dpPCR; 2.2 ± 2.5 SARS-CoV-2 adults; Mean dps: 5.0 ± 3.7 and Mean dpPCR: 1.8 ± 2.5 Loske et al. report that the key marker genes, RIG-I and MDA-5, are at peak expression at the first time-point that they report. Based on our data, we can now refine the temporal interpretation of this data, which reveals that RIG-I and MDA-5 expression peaks at day 5 after viral exposure (Reviewer Figure R2.4, see previous comment). This highlights the 4-day gap in knowledge that

our study uniquely addresses.

This is in line with multiple studies that estimate that the incubation period of the ancestral SARS-CoV-2 virus is between 4.6 - 6.4 days (Puhach et al. *Nat Rev Microbiol* 2023: <https://doi.org/10.1038/s41579-022-00822-w>), with the onset of symptoms varying between 2-14 days after viral exposure (<https://www.cdc.gov/coronavirus/2019-ncov/symptoms-testing/symptoms.html>). Whilst daily longitudinal sampling of respiratory specimens has revealed that viral RNA can already be detected prior to the onset of symptoms (Ke et al. *Nat Microbiol* 2022: <https://doi.org/10.1038/s41564-022-01105-z>). As there is usually no cause for testing prior to symptoms and the delays obtaining the results of PCR tests, few studies are able to catch/sample from infection at this early time point. Together, this would suggest most human studies to date have missed the earliest response to SARS-CoV-2 infection, with a need to better understand the epithelial and immune responses during this incubation period.

Importantly, within our study we are able to provide a pre-infection comparison for each individual patient and track the dynamics longitudinally across a detailed disease time course both within the blood and nasopharynx, capturing pre-symptomatic cellular responses. For example the upregulation of APR in ciliated cells, the activation of MAIT cells, depletion of some myeloid cells, and the global activation of IFN signalling in blood are observed before or at day 3 post challenge (**Fig. 2**) and prior to the onset of key symptoms, the earliest of which appeared predominantly appeared ~ 4 days post-inoculation (**Fig 4e**). Of note: for a direct comparison to Loske et al., cohorts also differ in terms of disease severity (with inclusion of moderate), demographic (e.g. mean adult age; 39.0 ±10.4).

In addition, we are also able to provide a comparison to two interesting patient subgroups who were challenged, but remained symptom free, going on to develop a transient or abortive infection. Due to the nature of these groups they would be impossible/highly difficult to capture

	and study in humans outside of a human challenge study.
C2.6 Six out of 16 participant inoculated with SARS-CoV-2 remained negative for virus PCR as well as any symptoms. How this result can be explained? Potential causes and mechanisms should be investigated based on the exciting data set available from the participants, described and discussed.	R2.6 As outlined in detail in R2.3, we have included multiple additional analyses to explore this, which led to the identification of HLA-DQA2 as a biomarker of protection, which is now discussed in the main text and discussion of the revised manuscript. Of note, as explained in detail in R1.4 and R2.33, the relatively low infection rate was intentional due to ethical concerns and to make our viral dose physiologically relevant.
C2.7 Line 119. The authors state that their data set contained all expected cell-types. However, neutrophils are missing. Neutrophils has been found in nasopharyngeal samples of healthy as well as SARS-CoV-2 infected individuals and were also described to play an important role in COVID-19. They should be present also in the nasopharyngeal samples and should be also considered in the data analyses. In the UMAP in Figure 1b neutrophils are missing, not clear why.	R2.7 The reduced representation of neutrophils in microfluidics based single cell sequencing is well recognised (Wigerblad et al. J Immunol 2022: https://doi.org/10.4049/jimmunol.2200154) and we now acknowledge this in the discussion. Reasons for this include  ● Neutrophils (and other granulocytes) have relatively low RNA content, expressing a few hundred genes (Hay et al. Exp Hematol 2018: https://doi.org/10.1016/j.exphem.2018.09.004). ● Neutrophils have relatively high levels of RNases and other inhibitory compounds resulting in fewer transcripts detected in GEMs, and less usable sequencing reads (https://kb.10xgenomics.com/hc/en-us/articles/360004024032-Can-I-process-

	neutrophils-or-other-granulocytes-using-10x-Single-Cell-applications-).  • Neutrophils are particularly sensitive to degradation after collection and cannot be isolated via density gradient centrifugation (as is the case for other nucleated blood cells, often referred to as PBMCs). As noted in Ziegler et al. Cell 2021 (https://doi.org/10.1016/j.cell.2021.07.023), on which our dissociation/digestion method was based, the intrinsic fragility of these cell types and the need for cryopreservation within the sample pipeline, granulocytes cell types are under-represented with this technique relative to the known true tissue-resident abundance. In order to study these cell types, further studies would be needed, with a protocol more specifically designed for neutrophils, likely with a specific enrichment step.
C2.8 Figure 2 2a) How the interferon stimulation signature score was determined? Information is missing in the methods section and should be added. In Figure 2a the same dot plots should be added for transient infection (which is probably the more important information). I would expect that in ciliated and secretory cells the interferon stimulation signature is earlier and stronger in the transient group compared to the sustained group, which would be a quite important result. Figure 6a should show these plots for abortive samples. Currently, Figure 6a shows a comparison between the three groups, but based on the proportions of Interferon-stimulated cells. This is not comparable to the information provided in Figure 2a and it is difficult to distinguish the response in the different cell types. It seem that in particular in ciliated cells there is a strong early interferon	R2.8 We apologise for the oversight, we have now included a methods section on how the interferon stimulation signature was defined and quantified. In short, we used the signature defined by Yoshida et al. Nature 2022 (https://doi.org/10.1038/s41586-021-04345-x), the 'AddModuleScore' function was used to quantify an aggregate measure of interferon stimulation per cell, and we classified cells as stimulated if this measure was higher than 0.5. We had originally not included the non-sustained groups in Figure 2 because we did not observe significant interferon stimulation in these individuals, as we mentioned in the main text. However, we agree that it is useful to support this claim by showing the lack of interferon stimulation dynamics over time in the non-sustained infection groups. We have therefore included an extended analysis in Extended Data Figure 5c, and we have now included an indication of significance to the plots to further aid the interpretation of this analysis. Extended Data Figure 5c shows that wide-spread interferon stimulation is absent in transient and abortive infections, including in ciliated cells. The small bump at day 1 is caused by a single individual, which is why we do not comment on it.

response which could be one of the reasons for a quick virus elimination and prevention of sustained infection. Please describe also how the proportion of interferon-stimulated cells was determined.	
C2.9 2b) It seems to be not appropriate for focus only on sustained infection to study the early immune response to SARS-CoV-2 (see also line 168-170). As shown in Figure 1f and 6a, the earliest response occurred in the transient group and should be described more in detail or at least in comparison to the sustained group. To focus in particular on the early response, data from day one post-inoculation should be included for the nasopharyngeal samples	R2.9 We agree that the antiviral responses in non-sustained infections represent a unique and important aspect of our study. Because of this, we performed extensive analyses to identify any responses that could be robustly associated with transient and/or abortive infections. These analyses revealed that immediate immune infiltration of the nasopharynx is associated with transient infections (Fig. 1f), and that both abortive and transient infection groups also display our newly identified dynamics in inflammatory monocytes and activated MAIT cells (Fig. 2b,c). In addition, the absence of key antiviral responses in these groups, such as the interferon and adaptive immune response (Extended Data Fig. 5c), represent unexpected findings related to non-sustained infections. In addition, our revised manuscript now includes a section about HLA-DQA2 (Fig. 2h-j), which is significantly enriched in non-sustained cases and associated with protection. Beyond these key observations, in order to keep the manuscript digestible and adhere to the word limit, we do not discuss or speculate about additional antiviral responses that are not significantly associated with transient and abortive infections.
C2.10 2c) Line 182-183 A significant decrease of inflammatory monocytes was observed in all groups – Fig. 2c shows an increase from day 3 on. This is in contrast to Figure 2b showing in fact a decrease in circulating myeloid cells from day 3 on and is very confusing. Not sure whether 2c is helpful at all.	R2.10 Compared to our baseline measurements before inoculation, the amount of inflammatory monocytes indeed drastically decreases by more than 10-fold on day 3 post-inoculation. After this initial drastic reduction, the amount of inflammatory monocytes indeed gradually goes up over the course of 2 weeks to baseline levels. This same behaviour is shown in both former Figure 2b and 2c. The only difference is that 2b is showing the original cell type abundances, while 2c is only showing the relative fold change in abundance compared to baseline. To avoid confusion and to improve the flow and focus of our manuscript, we have now moved 2c to the supplemental note, where its main purpose is to highlight the global myeloid migration dynamics.
C2.11 2i) Lines 246-248: How these genes were selected? Most significant? Why genes involved in	R2.11 These genes were indeed most significantly associated with viral gene expression, and selected in an unbiased manner as top ranking. Based on literature, we then identified that most

virus recognition and induction of an innate anti-virus response (such as DDX58, IFIH1, STAT1...) are not shown here? What about gene coding for chemokines that recruit immune cells? From the shown data in Figure 2i one could not be convinced that the hyper-infected ciliated cells show in fact an anti-inflammatory gene expression status. Furthermore, the genes shown in Extended Data Fig 6c do not support an attenuation of the interferon response in hyper-infected ciliated cells. To support this assumption a larger number of ISGs should be shown here.	of these top ranking genes can be associated with an immunosuppressive phenotype. We have now expanded the dotplot that delineates marker gene expression by infected and hyperinfected ciliated cells (Extended Data Figure 5g), which shows that antigen presentation machinery and multiple antiviral responses (e.g. interferon, innate, APR) are indeed dampened in hyperinfected ciliated cells compared to low-infected ciliated cells. This figure now also includes more ISGs. Of note, to improve the flow and focus of our story on the most novel and solid findings, we have moved these observations to the supplemental note that accompanies the revised manuscript.
C2.12 2g) How the fraction of acute phase response gene (APR) positive cells was determined? Please describe this in the methods section. Which gene set was used for classification?	R2.12 APR+ ciliated cells are novel and their identification was not based on pre-existing gene sets. They were identified by Leiden clustering analyses in an unbiased and unsupervised manner, which was the approach used for annotating all cell types and cell response states. Similar to the approach used in annotating all of our 202 cell states, clustering revealed a distinct ciliated cell cluster that we investigated using differential gene expression analysis, which indicated that acute phase response and MHC class II genes were the main markers for this cluster.
C2.13 Line 246-248. Ciliated cells exhibit a unique response to high viral amounts. Here a set of genes is mention including ERG1, NFKNIA.... How these genes where selected (most significant once?)? What about the genes involved in the innate anti-viral response? What about chemokines involved in the recruitment	R2.13 As outlined in two comments above, these genes were indeed most significantly associated with viral gene expression, and selected in an unbiased manner as top ranking. Based on literature, we then identified that most of these top ranking genes can be associated with an immunosuppressive phenotype. We have now expanded the dotplot that delineates marker gene expression by infected and hyperinfected ciliated cells (Extended Data Figure 5g), which shows that the antigen presentation machinery and multiple antiviral responses (e.g. interferon, innate, APR) are indeed dampened in

of inflammatory immune cells? What makes you sure about the anti-inflammatory potential of these cells?	hyperinfected ciliated cells compared to low-infected ciliated cells. Of note, to improve the flow and focus of our story on the most novel and solid findings, we have moved these observations to the supplemental note that accompanies the revised manuscript.
C2.14 Line 268-271. Why a focus was set on APR+ and MHCII+ cells? Do these pathways were identified following enrichment analyses or differential gene expression analyses?	R2.14 As described in more detail two comments above, these were not selected a priori based on existing hypotheses, but identified as distinct cell clusters in unbiased clustering analyses.
C2.15 Figure 3b. Is this data from the sustained infection group only? If so, please explain this in the legend	R2.15 This data comes from all activated T cells of the entire dataset. We now explain this in the legend.
C2.16 Line 342-343. From Figure 3b it is not obvious that CD4+ T cells express high amounts of cytotoxicity genes, in particular PRF1. A comparison of the cytotoxicity gene expression should be shown (with significance) between day -1 and the later time points, as well as between the three groups.	R2.16 The seemingly low PRF1 expression in activated CD4+ cells might be a visual deception as these CD4+ cells are contrasted to highly cytotoxic activated CD8+ T cells. As shown in Reviewer Figure R2.16 and Extended Data Figure 3a and 4a, Activated Cytotoxic CD4+ T cells express high amounts of PRF1 compared to other CD4+ subsets. In Reviewer Figure R2.16a, we show additional cytotoxicity associated genes (based on Gene Ontology), which further strengthens our observation that activated CD4+ T cells are more cytotoxic than other CD4+ T cell subsets.

	Reviewer Figure R2.16: Scaled expression of cytotoxicity associated genes across CD4+ T cell types.
C2.17 It is speculated that the activation of cytotoxic CD4+ T cells is potentially a result of MHC class II presentation by infected ciliated cells. This is only a hypothesis. Without providing any supportive experimental data such a speculation should be avoided. I would suggest to remove this speculative part.	R2.17 We agree that this is speculative and have therefore removed this hypothesis from the revised manuscript.
C2.18 Figure 3j. Not clear which data are shown here. From the sustained group? Please add a more detailed description to the legend	R2.18 This figure indeed only shows B cell dynamics in the sustained infection group, and is a subset of the full cell type dynamics plot across all infection groups in Extended Data Figure 7e. We now clarified this in the legends of the revised manuscript.
C2.19 Line 411-413. Activated T cells has been described already in earlier studies in	R2.19 Based on our data, we define an activated T cell as a T cell that is proliferative, expresses both naive, activation, and maturation markers, and that is clonally expanding in an antigen specific manner. The activation subset that we discovered is highly transient, expresses a distinct

nasopharyngeal samples. Please adapt the wording and cite this earlier studies.

set of marker genes, lacks classical activation markers (such as OX40 or CD69), and allows for identification of converging TCR motifs. To the best of our knowledge, this cell state has not been captured and described in detail before in other COVID19 studies. After an extensive literature search for this manuscript and for our patent application that has recently been filed by both ourselves and our attorneys, which focuses on translational application of activated T cells, we have not been able to identify existing literature that characterize the same cellular population in a comprehensive manner. All existing literature that we could find is using targeted assays to measure or isolate activated T cells, for example in time course after vaccination (Reinscheid et al. *Nat Commun* 2022: <https://doi.org/10.1038/s41467-022-32324-x>), or in natural infections where cycling and CD38+ were isolated (Mathew et al. *Science* 2020: <https://doi.org/10.1126/science.abc8511>; COMBAT consortium *Cell* 2022: <https://doi.org/10.1016/j.cell.2022.01.012>).

Beyond these studies, T cell activation has been extensively studied using for example antigen stimulation assays and other in vitro assays, such as the Activation Induced Markers (AIM) assay. Here, T cells are sorted based on CD154, CD69, CD107a, CD137, OX40, CD25, PD-L1, or a combination of these. It is important to note here that none of these markers are upregulated (at either protein or RNA level) on our activated T cell populations, underscoring the uniqueness of our population, which is likely explained by the transient nature of this cell state. We believe that these 'classical' markers mostly demarcate reactivation events of already primed T cells, which is fundamentally different from the activation event that initiates the onset of a T cell response.

We furthermore looked for activated T cells as defined above in a recent single cell antigen stimulation dataset (Soskic et al. *Nat Genet* 2022: <https://doi.org/10.1038/s41588-022-01066-3>). This analysis (see **Reviewer Figures R.2.19a-b** below) showed that even after 5 days of antigen stimulation, only 30% of T cells exhibited the activation state we describe in this manuscript, indicating that our activation state is a distinct subset of antigen-stimulated T cells and has not been specifically described in previous literature.

Reviewer Figure R2.19a: Number of cells from Soskic et al. by time point, stratified by predicted activation state.

Reviewer Figure R2.19b: Fraction of all T cells in predicted activated state in Soskic et al. stratified by time point.

If we inadvertently missed literature that has already comprehensively identified these populations before, we would greatly appreciate the references for these studies.

C2.20 Figure 4b. Please provide a definition of the activated T cells

R2.20 Activated T cells are indeed defined by the marker genes shown in Figure 3b, where - amongst T cells - CD38 is the most specific for the various subtypes of activated T cells that we identify. As described in the main text and supported by further analyses and Dextramer experiments, we further define these cells as temporally restricted, clonally expanding, and antigen specific. Based on the timing of their appearance, we hypothesise that we capture a transient initial phase of activation, also supported by the mixture of naive, activation and early maturation markers, which - to our knowledge - has not been described before using unbiased profiling screening of blood or nasopharyngeal cells.

C2.21 Figure 4f. This Figure refers to the earlier published COVID-19 studies where the case numbers are provided in Extended Table 1d and e. However, the case numbers of different disease severities can not be found anywhere, please add. Furthermore, for the COVID-19 patient data, the information on the sampling time-point (days post infection) is missing. In general, the COVID-19 patient data set is poorly described

R2.21 We have added **Extended Data Table 1m** listing each sample and metadata (study, sample_id, patient_id, severity, days_since_onset_symptoms), and expanded information on severity in **Extended Data Table 1e**.

Reviewer Figure R2.21: Sample counts for each day since onset of symptoms, stratified by public COVID-19 patient dataset source.

(Due to space constraints this reviewer figure is not included in the manuscript, but all participant information is available in the Extended Data Table 1.)

C2.22 In general, a statistical analysis is completely missing. Which cell population significantly increase or decrease over time. Which gene/gene sets are differentially expressed over time? Significance data should be added to each of the Plots.	R2.22 We believe there is a misunderstanding in the statistical approach that we took, and the way that we report this. All our analyses in which we made comparisons (between for example infection groups or time points) were subjected to a high degree of statistical rigour. We used generalized linear mixed modeling to capture significant changes in cell type abundance and distinguish temporal from infection-group-specific changes, while controlling for technical and biological covariates, and for unbalanced group sizes (using random effect terms). Importantly, we determined the significance of all observed changes by calculating the local true sign rate and performing multiple testing hypothesis correction using the Benjamini-Hochberg procedure, which allowed us to report both fold changes and false discovery rates (FDR) of each cell type/state over time and infection group. In many of our figures, these FDRs are reported in dotplots, where the size of each dot corresponds to the significance of the change. Because we believe it's important to perform the Benjamini-Hochberg procedure across all comparisons that we made, we calculated FDRs once and reported those in Extended Data Figure 7e,f,g. In some of our figure panels where we focus on dynamics of specific cell types, we have then cropped this large dotplot into easy-to-digest subsets (for example Figure 3f), or we have extracted the underlying data and shown it as boxplots or bar graphs. When showing the data changes in cell type abundance as bar graphs or boxplots, we have deliberately not included a separate significance test, because this would artificially lower the resulting FDRs and is considered to be poor practice in statistics. Instead, we refer to the cellular landscape-wide analysis in Extended Data Figure 7e-g for significance. We have now clarified this approach in each of the relevant figure legends.
C2.23 Discussion The authors state in the discussion (line 505-508) that multiple response states have been detected that precede the onset of clinical manifestation and that this represents a newly discovered immune	R2.23 This statement is referring to 1) the >10 fold reduction in circulating inflammatory monocytes, 2) the near complete activation of MAIT cells upon exposure, and 3) the immediate nasopharyngeal immune infiltration. Despite the absence of a sustained infection, the first two responses are prominently present in both abortive and transient infections, while the third is specific for transient infections only. Of note, we don't claim here or in our manuscript that these responses are the preventive cause of the abortive or transient infections, but these observations

response that emerges when an individual is exposed to SARS-CoV-2, but manages to prevent the onset of viral spread. This is not correct. In fact, different response states are described very detailed for individuals with sustained infection. Information which response states prevent viral spread (transient and abortive groups) is completely missing.	do show that some parts of the systemic immune system are able to sense viral exposure before it has become an established infection that can sustain itself. In addition, as is described in more detail above, we have now discovered HLA-DQA2, whose baseline expression is associated with preventing the onset of a sustained infection. Because the identification of immune responses in non-sustained infection cases represents one of our key findings, we have now re-focused our revised manuscript to focus on our key findings and by moving some of the smaller findings to a Supplementary Note.
C2.24 Not clear which feature could be used as biomarkers (line 508 discussion). The results section does not provide data on potential biomarkers. For many different cell populations and gene sets an up or down has been described in a descriptive manner. No idea which of those feature could serve as biomarkers. I would suggest to remove this.	C2.24 Thank you for pointing this out, and we agree that the word “biomarker” needs to be clarified. We have now re-worded this sentence to point out the application of the reduction of inflammatory monocytes and the activation of MAIT cells. For instance, one could analyse these subpopulations by FACS to identify transient or abortive infection cases. As part of our validation experiments, we now show that FACS can be used to identify the activation of MAIT cells in exposed healthcare workers. In the revised manuscript we now clarified which features we were referring to.
C2.25 In the discussion it is further mentioned, that a major advantage of this challenge experiment is that also abortive and transient infections can be studied to unravel immune signatures preventing the infection. This would be in fact an important issue, but unfortunately is not really addressed in this manuscript.	R2.25 As discussed in more detail in the responses above, we have now included additional analyses to focus more on the (baseline) differences in abortive and transient infections, which - in addition to the reduced inflammatory monocytes, activated MAIT cells, and transient infiltration of the nasopharynx - now also includes a new biomarker for protection against the onset of a sustained infection.
C2.26 Figure 5.1. Instead the time course of immune infiltration here the induction of the innate immune	R2.26 We agree that this is an important aspect of our study, and it is therefore outlined in panel 5.3 of this figure, alongside with other sustained-infection-specific responses.

response both in immune and epithelial cells would be of interest.	
C2.27 In line 524-525 it is said that during sustained infections that lead to COVID-19, a novel acute phase response was observed in ciliated cells together with an activation of MAIT cells and a decrease of inflammatory monocytes in blood. This is exactly shown in Figure 5.2. but under the headline of “antiviral responses in non-sustained groups”. If this is really data from non-sustained groups, it is not clear to which data/Figures this summary refers to. At least one chapter should be added to the results part describing in detail the immune and epithelial responses in non-sustained groups. It is further mentioned, that these two responses (shown in 5.2) are the only immune responses that are also observed in participants with transient and abortive infections. Again, where in the manuscript this data is shown? How the authors have proved that these are the only responses in the non-sustained groups? How the comparison between the sustained and non-sustained groups has been done?	R2.27 This is shown in main Figure 2b and 2c. Comparisons across all time points, cell states, and infection were shown in Extended Data Figure 7e-f. These results were discussed in the sections ‘Novel MAIT subset is activated in both sustained and abortive infections’ and in the second paragraph of ‘Temporal reduction of myeloid subsets immediately after viral exposure’. The primary message of these paragraphs is highlighting the presence of a response in non-sustained cases. These comparisons have been done using generalised linear mixed models (GLMM) to quantify the changes in cell type abundances over time since inoculation compared to the day prior to inoculation (-1). This allowed us to perform paired longitudinal modelling of donor-specific effects while accounting for technical and biological variation using random effect terms.
C2.28 Figure 5.3 3 Three stages of the local ciliated cell response are mentioned here. Where in the	R2.28 These are described in two sections of the main text: ‘Hyper-infection and anti-inflammatory response of ciliated cells’ and ‘Ciliated cell acute phase response to virus’, totaling in 5 distinct ciliated cell states.

manuscript these three stages and their time course are described?	
C2.29 Figure 5. Community contributions iii) What kind of vaccine targets and therapeutics have been identified? If none of the individuals has developed a severe disease course, would it be possible at all to suggest therapeutics based on the data presented here?	R2.29 We report 29'486 putative SARS-CoV-2 specific TCR motifs (Extended Data Table 1f), of which 326 contain two or more unique TCR clones and 266 are found in two or more COVID-19 patients, and resolve their specific antigen in 1'179 cases. We believe that this repository has a clear potential to be used to identify vaccine targets and TCR based therapeutics. In addition, the biomarkers discovered by us could have applications in diagnostics and as potential targets to augment protection.
C2.30 Line 534-538. It is further mentioned in the discussion that in sustained infection the activation of interferon signaling in blood precedes activation at the site of inoculation. According to the results presented in Figure 2a this seems to be not true. In this Figure the IFN stimulation occurred both in blood and nasopharyngeal cells at day 3 and started even earlier (day 1) in the nasopharyngeal cells.	R2.30 While a few cell types in Fig. 2a show minor or non-significant changes at days 1 and 3 in nasopharyngeal cells, the majority of cells start showing a strong interferon stimulation signature score from day 5 onwards. In PBMCs, nearly all cell types are already completely stimulated at day 3. This quantitative difference is also highlighted in Extended Data Fig. 5a, where up to 100% of the cells are classified as interferon stimulated. In our revised manuscript, we have now validated this observation in an additional 20 challenged individuals in a bulk-RNA-seq time course that was sampled at higher temporal resolution (Reviewer Figure R1.2.a, Extended Data Figure 7h). This analysis shows a clear delay of the local interferon response, confirming our single cell analyses.
C2.31 Line 539. The relatively slow immune response at the site of infection is in contrast to the immediate immune infiltration observed in transient infection. Again, this sentence seems to be incorrect. As Extended Data Figure 6a shows, there is a very fast interferon-response in transient infection.	R2.31 We agree that this is a confusing sentence, 'Immune response' is indeed a multi-interpretable term in this context. We reworded the text to clarify that we were referring to immune cell infiltration.

C2.32 Line 550. It is not only possible but have been already shown in many early studies that patients that require hospitalization exhibit an perturbed interferon response and an exacerbated inflammatory response. This sentence should be corrected accordingly. Furthermore I would suggest to discuss the limitation of being able to study only mild disease course a little bit more.	R2.32 We agree and have now removed the uncertainty claim here and included references that show that hospitalisation leads to perturbed immune responses. We now have also discuss the limitation of our study: “Participants enrolled in this study cleared the infection with mild symptoms, which means that caution should be taken when extrapolating our findings to critically ill COVID-19 patients.
C2.33 The participants have been inoculated with a very low virus dose. It should be discussed how representative the observed results are for an infection occurring under natural conditions with potentially much higher virus concentrations.	R2.33 Please refer to our detailed response to Reviewer 1 (R.1.4) on how the dose used in this study (10 TCID50) compares with a physiologically relevant real-world exposure.
C2.34 Furthermore, a comparison of the observed response dynamics under experimental conditions with the response pattern described in COVID-19 patients with more severe disease courses would be helpful.	R2.34 This comparison is difficult to perform as patient data lacks exact timing information and is not controlled for important covariates that heavily impact the immune response. Nevertheless, we have integrated our challenged individuals with 325 COVID-19 patients, including 128 severe or critical ones (Extended Data Table 1d-e). Again, while we can see that timing of sampling is the main source of variance, which cannot be accounted for in the patient data, we can fit our temporal model onto the patient data. This shows that the molecular timing inferred by us corresponds with time since onset of symptoms (if available), which underscores the usefulness of our resource, but indeed reveals that this integration and temporal modeling works best for patients with mild symptoms, while severe and critical patients appear to have a perturbed and delayed immune response (Extended Data Figure 8f). However, we do note that most severe and critical patients in particular are sampled at very late stages in their disease that don’t overlap with our time points, which makes direct comparisons difficult to perform and interpret.

C2.35 Line 556. Activated conventional T cells have already been described in earlier single cell studies focusing on more severe COVID-19 course. I would suggest to remove this sentence.	R2.35 As discussed in detail in the response on activated T cells above (R2.19), we believe that we are the first single cell study to comprehensively characterize these early responding cell states, presumably developing from naive precursors, as part of the natural immune response to infection.
C2.36 The conclusion should be more cautious and should say, that this is a detailed time-resolved description of sustained infection with mild disease course.	R2.36 We have toned down our concluding statement by mentioning that this is mild disease specific.

Comments referee #3

Comment by the referee	Response by the authors
C3.1 The manuscript by Lindeboom and collaborators outlines a description of the immune response to SARS-CoV-2 through scRNAseq, CITE-seq, TCR, and BCR sequencing. For this, investigators exposed healthy unvaccinated human participants to pre-Alpha-SARS-CoV-2 in a controlled environment with subsequent collection of longitudinal nasopharyngeal and blood samples. The manuscript describes local immune infiltration post-exposure, systemic induction of interferon response, decrease in the frequency of nasopharyngeal myeloid cells post-exposure in patients who developed a sustained infection, systemic activation of MAIT cells, SARS-CoV-2 ssRNA detection in various cell types including CD8+ T cells, restriction of SARS-CoV-2 replication to ciliated and goblet cells, MHC class II expression in infected epithelial cells, clonal expansion and activation of SARS-CoV-2-specific T cells, presence of non-TRDV2/TRGV9-expressing g/d T cells, and clonal expansion of B cells. Additionally, the authors present the Cell2TCR approach for the detection of clonotype clustering. The study offers a unique opportunity to analyze patients prior to exposure, in the days between	R3.1 We thank the reviewer for the kind words and for highlighting that our study is heroic, unique and of public importance. Based on your insightful and constructive feedback, we have made major improvements to our manuscript: • We now include a large validation cohort of 20 challenged participants to increase our sample size from 16 to 36 (Extended Data Figures 5h, 6h, 6k, 7h, 8g, 8h).• We provide additional patient- (Extended Data Fig. 5e,f) and in vitro derived (Fig. 2g, Extended Data Fig. 5b) validation data.• Another key improvement made based on your comments is our expanded analysis of the pre-infection correlates with infection outcome, which revealed a new biomarker (HLA-DQA2) predictive for preventing the onset of a sustained infection (Fig 2h-j, Extended Data Fig. 6f-h).• In addition, we have improved the discussion of existing literature, the flow and conciseness of our manuscript, clarity of our figures, legends and statistics used. Below, you can find our detailed response to your comments, which has significantly improved the quality of our manuscript.

exposure and development of symptoms and even in the first days of symptoms, as well as exposed patients who avoided sustained infection. These are all aspects of public interest and importance. The manuscript covers a wide dataset translating great investment and effort from the group and describes a broad array of more or less well-established immune processes. However, three years after the start of the pandemic, from the immune response description standpoint, the current presentation of the data brings limited uniquely novel advances and there is an opportunity to elevate this study to meet its potential. Additionally, choices in the organization of the manuscript and interpretation of the data need improvement. Still, the group's extensive effort and work to thoroughly provide longitudinal and single-cell level descriptions of host responses to SARS-CoV-2 infection is heroic.

Major concerns

C3.2 One of the unexpected findings that the authors state is the “Widespread systemic interferon response precedes response at site of inoculation” (149) However, it appears that ISG responses are indeed starting in NP cells at day 3 also (Fig. 2a). Measuring IFN levels in the nasopharynx and plasma would be important to support these findings and

R3.2 We agree that this is a key point of our manuscript, and that it would benefit from validation and additional temporal resolution. We therefore obtained temporal expression profiling data from an additional 20 challenged participants (Rosenheim et al. *bioRxiv* 2023 (<https://doi.org/10.1101/2023.06.01.23290819>)). In contrast to our single cell resolved dataset, this validation dataset was profiled using bulk RNA-seq, which enabled our collaborators to obtain higher temporal resolution. With the validation cohort, we are able to validate our observation that the interferon in the blood precedes the interferon response at the site of inoculation (**Reviewer Figure R1.2.a** and **Extended Data Figure 7h**). In addition, the increased temporal resolution adds more detail to this observation and reveals that the circulating interferon response is abruptly activated to peak activation between day 2 and 3. In contrast, the local interferon

claims. Also, why is day 1 missing from PBMC in Fig. 2a? This information is needed to support the authors' claim.

response is still at baseline on day 3 in most participants, and gradually increases to its peak at day 5 - 7.

In addition to validation, we have now show the statistical significance of the interferon stimulation over time in **Figure 2a**, which underscores that the few strong increases in nasopharyngeal interferon activation at day 3 post-inoculation are sporadic changes and not significant that likely originate from very small cell populations or from individual participants. The few cell populations that do already significantly increase at day three only show very small increases in cell numbers. Based on our new data, these are thought to originate from outlier participants.

We can only obtain a single nasopharyngeal swab per participant per time point, which makes these samples very precious and limits us in performing targeted assays to measure individual biomolecules such as interferons. However, single cell RNA-seq is a great way to study the RNA production of interferons, which - unlike targeted methods - gives a cell type resolved insights in cytokine production. **Reviewer Figure R3.2** and **Extended Data Table 1n-o** shows that despite the large activation of signalling, neither nasopharyngeal nor circulating cells dynamically produce type I, II or III interferons in a way that could explain the observed activation events. The lack of interferon production dynamics in either circulation or at the site of inoculation, form the basis for our hypothesis that the interferons that promptly stimulated circulating immune cells are possibly produced in lymph nodes and secondary immune organs.

	Reviewer Figure R3.2: Interferon mRNA production by nasopharyngeal cells (left) and PBMCs (right). Reads from all interferon type 1 genes (top) or from all interferon type 3 genes (bottom) have been aggregated to make this dotplot. Color scale represents the mean (aggregate) log1p expression value within each cell type at each time point and infection group. The size of each dot represents the percentage of cells with detectable expression.
C3.3 Related to the above, for the statement “immediate response to SARS-CoV-2 infection includes informing circulating immune cells before tissue-resident cells through interferon signaling” (162-163) - is it possible that local response is delayed in sustained infection instead?	R3.3 We believe that the reviewer here hypothesises that an immediate local interferon response is protective, and that the observed delayed interferon response could be a reason that these individuals become sustained infections. This is an interesting idea and agrees with studies by us and others that SARS-CoV-2 can suppress interferon signalling. However, using our unique transient and abortive infection groups, we can show that people that ‘fight off’ the virus before the infection becomes sustained, do not mount an interferon response at all. This argues that the widespread interferon stimulation that we observe, is not necessary immediately after exposure in order to avoid a sustained infection.
C3.4 Another unexpected finding is the presence of viral RNA genomes within CD8 T cells in the nasopharynx (Fig. 2e). Figure 1f shows that there is no increase in CD8 T cell number in any of the infected individuals within the nasopharynx at day 7. Does this mean that CD8 T cells that are resident in the NP are being infected? What do their TCR sequences look like – are they the ones identified to be specific to SARS-CoV-2 antigens? How about other transcripts – what type of CD8 T cells are being infected, and any hint of activation/cycling/cell death in the RNA signature? This is an important opportunity to learn	R3.4 We agree that this is a very surprising finding. Of note, SARS-CoV-2 infection of T cells through an ACE2 independent manner has been reported before in vitro and in COVID-19 patients lungs (Wang et al. Sig Transduct Target Ther 2020: https://doi.org/10.1038/s41392-020-00426-x; Shen et al. Sig Transduct Target Ther 2022: https://doi.org/10.1038/s41392-022-00919-x; Brunetti et al. Elife 2023 https://doi.org/10.7554/eLife.84790). References for these findings are now included in the manuscript. In line with this, we have performed additional analyses investigating the expression of viral entry genes, which revealed that infected CD8 T cells do not express viral entry genes (Discussed in detail in R1.14). In addition, we investigated if there are any distinguishing features of infected CD8 T cells. This revealed that CD8 T cells are indeed highly similar to interferon-stimulated tissue resident CD8s, as the reviewer had already noted. The only distinguishing feature that we could detect beyond the viral reads is the expression of CXCL10, which appears to be a recurring marker for all infected subpopulations (Extended Data Figure 6a). The reviewer raises an excellent suggestion to use TCR data to further investigate their origin. We performed an in-depth analysis of the TCR motifs of infected nasal T cells with resolved TCR (n=323

more about the potential impact of SARS-CoV-2 infection on CD8 T cell function and fate.

T cells, of which 84% CD8+ T cells). 13% of the motifs (25/187) from the infected T cells are shared with T cells sampled at day -1 in the nasopharynx (see **Reviewer Figure R3.4.a** below, baseline vs infected motif counts). Matches were found with T resident memory cells for 96% of cases.

Reviewer Figure R3.4.a: Distribution of TCR motifs from baseline (day -1, number of motifs = 386) and infected T cells (number of motifs = 187) of the nasopharynx, indicating whether a TCR motif was found in baseline T cells, infected T cells or both. Only 25 TCR motifs are found in both categories.

When comparing the motifs of infected T cells and activated T cells (n=498) from the nasopharynx, we found six overlapping TCR motifs (see **Reviewer Figure R3.4.b** below). The low number of matches with the putative SARS-CoV-2 specific population (3% of all nasal activated TCR motifs) together with the almost 5 fold increased proportion of motif matches with baseline resident memory T cells indicates that SARS-CoV-2 infection of T cells mostly takes place in pre-existing and

unrelated tissue-resident CD8 T cells, and is not biased by antigen-specificity of the T cells.

Reviewer **Figure R3.4.b**: Distribution of TCR motifs from nasal activated (number of motifs = 445) and infected T cells (number of motifs = 187) of the nasopharynx, indicating whether a TCR motif was found in activated T cells, infected T cells or both. Only 6 TCR motifs are found in both categories.

C3.5 Another major claim of this study is the reduction in circulating inflammatory monocytes in all exposed individuals, regardless of the sustained nature of the infection. However, they don't seem to be migrating into the nasopharynx to fight off infection (Fig. 2b). Where are they going?	R3.5 Indeed, we find a reduction of circulating inflammatory monocytes at day 3 across all infection conditions (Fig 2b) and also a reduction of monocytes in the nasopharynx (d1) in sustained and abortive infections (Fig 1f, also see Extended Data Fig. 5i). Our study sampled both local and systemic immune responses, but clearly it is not practically possible to sample all immune locations of the patients. Sites not sampled include secondary and tertiary lymphoid structures and additional surface mucosal areas such as the throat, which has been reported as an early site of infection (Killingley et al. Nat Med 2022: https://doi.org/10.1038/s41591-022-01780-9). Monocytes are also known leave the circulation in great numbers and are able to enter draining lymph nodes or migrate to tissues and differentiate into other cell types (reviewed in Muller et al. J Exp Med 2001: https://doi.org/10.1084/jem.194.9.f47 and Hampton et al. Front Immunol 2019: https://doi.org/10.3389/fimmu.2019.01168). We also note that in COVID-19 monocytes have been reported to undergo increased cell death via pyroptosis (Junquera et al. Nature 2022: https://doi.org/10.1038/s41586-022-04702-4) As we are unable to further explore these possibilities within the current study, we prefer not to speculate but highlight the monocyte responses in the results section and Supplementary Note.
C3.6 Authors do not explore differences in viral load between patients. In a setting that stands out for the controlled initial viral load, there is a remarkable	R3.6 Our experimental setup has been designed to ensure high statistical resolving power between infection groups and time points. However, our sample sizes become too limiting when studying heterogeneity within infection groups. Likely due to these technical constraints, we were not able to find baseline associates with viral load within the sustained infection group. However,

opportunity to analyze which baseline or early features correlate with lower or higher viral loads.

these analyses inspired by the suggestions from the reviewer did reveal that the amount of certain cell types did significantly associate with viral load. A correlation analysis between viral load determined by qPCR and relative cell type abundances across time, revealed that certain nasopharyngeal innate immune cell types, including monocytes, gamma/delta T cells, activated cDCs, and pDCs, significantly associate with viral load. Importantly, hyperinfected ciliated cells, but not low-infected ciliated cells or any other infected cell subtype, significantly correlate with viral load (**Reviewer Figure R3.6** and **Extended Data Figure 7i**). This strengthens our hypothesis that hyperinfected ciliated cells are major contributors to the total amount of virions that are produced and spread.

Reviewer Figure R3.6: Correlation analysis of relative cell type abundance and viral load as determined by qPCR. Pearson correlation coefficients are shown on the X axis. Minus log₁₀

	transformed p values shown on the Y axis were corrected for multiple testing by the Benjamini-Hochberg procedure. Only infected cell types or cell types with an FDR<0.01 are labelled. Dots from infected cell types were coloured black.
C3.7 To me, the most interesting finding of this study is that about half of the exposed individuals do not get productively infected. What are the factors that determine this? The authors can mine information from their day -1 samples to examine this issue. Also, factors predictive of transient infection phenotype would be important to determine.	R3.7 Please refer to R2.3 above.
C3.8 Fig 2i shows spearman correlation analysis of SARS-CoV-2 reads and gene expression – it does not show upregulation of the mentioned genes per se, as implied in the text (245-248). “suggests that SARS-CoV-2 is capable of inducing” (248-249) In addition to not showing upregulation, the data also does not account for SARS-CoV-2 directly inducing changes in gene expressions.	R3.8 Thank you for your suggestion, we have now reworded this to: ‘This suggests that SARS-CoV-2 is associated with a unique response state in hyper-infected ciliated cells’, and furthermore moved this entire paragraph into a Supplementary Note.
C3.9 While I understand that these human challenge studies were approved by the ethics committee, I wonder whether any of the subjects developed long-term sequelae. This is an important outcome that	R3.9 Although we have only focused on the early time points here within this particular study (<28 days post-challenge), participants of the SARS-CoV-2 human challenge study were followed for 12 months post-inoculation. Importantly at the final 12 month follow-up there were no residual symptoms attributed to the viral challenge, no unresolved adverse events and no abnormalities on investigations including UPSIT scores. We have now added a column to Extended Data Table 1g (summary of participant metadata) named “Adverse events” to help highlight this fact to the

should be considered for future proposed studies. Please comment on this aspect.	reader.
C3.10 Statistical analyses are lacking in many figures.	R3.10 We believe there is a misunderstanding in the statistical approach that we took, and the way that we report this. All our analyses in which we made comparisons (between for example infection groups or time points) were subjected to a high degree of statistical rigour. We used generalised linear mixed modelling to capture significant changes in cell type abundance and distinguish temporal from infection-group-specific changes, while controlling for technical and biological covariates, and for unbalanced group sizes (using random effect terms). Importantly, we determined the significance of all observed changes by calculating the local true sign rate and performing multiple testing hypothesis correction using the Benjamini-Hochberg procedure, which allowed us to report both fold changes and false discovery rates (FDR) of each cell type/state over time and infection group. In many of our figures, these FDRs are reported in dotplots, where the size of each dot corresponds to the significance of the change. Because we believe it's important to perform the Benjamini-Hochberg procedure across all comparisons that we made, we calculated FDRs once and reported those in Extended Data Figure 7e,f,g. In some of our figure panels where we focus on dynamics of specific cell types, we have then cropped this large dotplot into easy-to-digest subsets (for example Figure 3f), or we have extracted the underlying data and shown it as boxplots or bar graphs. When showing the data changes in cell type abundance as bar graphs or boxplots, we have deliberately not included a separate significance test, because this would artificially lower the resulting FDRs and is considered to be a bad practice in statistics. Instead, we refer to the cellular landscape-wide analysis in Extended Data Figure 7e-g for significance. We have now clarified this approach in each of the relevant figure legends. In addition, we agree that panels 2a and 2f would benefit from including an additional statistical test, since the comparisons made in these panels are not referring to cell type abundances. We now show p values or FDRs in these panels.

C3.11 A detailed description of the participants' demographics should be included.

R3.11 A table summarising the participants' demographics, plus selected metadata (i.e. reported symptoms, clinical test results) can be found in the **Extended Data Table 1g (Study_metadata)**. We have expanded this to include additional categories of interest such as “Antibody Titre” and “Adverse events”.

			Total (n=16)	Substained infections (n=6)	Transient Infections (n=3)	Abortive Infections (n=7)	
Demographic	Age (years)	Mean age (s.d.)	21.9 (3.0)	22.3 (2.4)	22.3 (1.9)	21.4 (3.7)	
		Min, Max	18, 29	19, 26	21, 25	18, 29	
	Sex, n (%)	Male	12 (75)	4 (66.7)	2 (66.7)	6 (85.7)	
		Female	4 (25)	2 (33.3)	1 (33.3)	1 (14.3)	
	Ethnicity, n (%)	Hispanic or Latino	2 (12.5)	1 (16.7)	0 (0)	1 (14.3)	
		Non-hispanic or latino	14 (87.5)	5 (83.3)	2 (100)	6 (85.7)	
	Race, n (%)	White	14 (87.5)	5 (83.3)	3 (100)	6 (85.7)	
		Black	0 (0)	0 (0)	0 (0)	0 (0)	
		Mixed	0 (0)	0 (0)	0 (0)	1 (14.3)	
		Other	1 (6.3)	1 (16.7)	0 (0)	0 (0)	
	Reported symptoms	Any symptoms reported on two consecutive day, n (%)		12 (75)	6 (100)	2 (66.7)	4 (57.1)
	Clinical Tests	Anosmia or dysosmia	UPSIT Smell Test	6 (37.5)	5 (83.3)	1 (33.3)	0 (0)
		Temperature ($\geq 38^{\circ}\text{C}$)		1 (6.3)	1 (16.7)	0 (0)	0 (0)
	Antibody titers at 28 days after inoculation	Neutralising antibody titer (median)			899.0 (1300)	Undetectable	Undetectable
Spike-specific IgG titer (ELU/ml-1, median)			1,777 (2340)	Undetectable	Undetectable		
Adverse events	Any serious adverse event		0	0	0	0	

Reviewer Figure R3.11: Overview of selected metadata for the 16 participants enrolled in the single cell RNA-seq cohort of this study.

Minor concerns

C3.12 Lack of clarity: Phrasing choices, clarity of data, and proper reference of previous work mentioned should be improved. (ex., “up to 100% of some cell types at a given time” (154),)). Authors are not consistently clear in how statistical significance is defined throughout the different analysis. (ex. “highly significant” (308), “slight increase” (472)).

R3.12 Thank you for bringing this to our attention. We have increased clarity throughout the text, included p values in text, and modified the wording in these lines of the manuscript.

For a detailed discussion of the statistical rigour with which we analysed our data, please refer to the answer to your concern in R3.10 above.

C3.13 Figure legends should always state patients and time points included, which statistical analysis was used, and what each dot represents.	R3.13 We now clarified these points in both the figure legends and the labels in the figures.
C3.14 The manuscript does not adequately acknowledge the extent of previous work in the field. Authors claim that “our understanding of the cellular disease dynamics remains limited” (29), but there has been extensive work from naturally infected individuals. The authors should also reference previous work in “Studies by us and others” (49), “In contrast to previous studies” (213), “we and others” (284).	R3.14 We acknowledge your point and have changed line 29 to more specially refer to the gaps in our knowledge regarding the cellular dynamics within the early acute phase of natural infections, where our knowledge is still limited largely due to the difficulties collecting clinical samples at these early time points. We have made additional changes to the manuscript’s text, but where applicable we have also added additional references.
C3.15 Markers used for grouped analysis (ex. ISG score in Fig 2a) should be more clearly stated.	R3.15 We apologise for the oversight, we have now included a methods section on how the interferon stimulation signature was defined and quantified. In short, we used the signature defined by Yoshida et al. Nature 2022 (https://doi.org/10.1038/s41586-021-04345-x), the ‘AddModuleScore’ function was used to quantify an aggregate measure of interferon stimulation per cell, and we classified cells as stimulated if this measure was higher than 0.5.
C3.16 The word “strikingly” is overused.	R3.16 We agree and have reduced the use of this word in our revised manuscript.
C3.17 Cell annotation process could be more clearly addressed – from Extended Figure 3, it seems like not all cells annotated as CD4+ express CD4 and that not all infected cells had detectable SARS-CoV-2.	R3.17 Cell type and state annotation of single cell RNA-seq data is performed by clustering similar cells together, and not based on the marker gene expression of an individual cell. Marker gene expression is then assessed at a per cluster basis, and used (together with differential gene expression analysis on each cluster) to annotate the cell state of each cluster. This approach makes cell type annotation more robust and enables the use of lowly-expressed cell type markers such as CD4. This also means that an individual cell in which we don’t detect any CD4 expression can

	be annotated as a CD4+ T cell if the expression dynamics of the thousands of other genes it expresses are highly similar to many other CD4+ cells. This is standard practice in the field of single cell transcriptomics.
C3.18 Please be more specific in the statements. “all immune cell types significantly infiltrate the site of inoculation after exposure to SARS-CoV-2” (132-133). However, abortive infection patients were also exposed to SARS-CoV-2 and did not display this pattern.	R3.18 The quoted sentence does specify that this applies in sustained and transient transfections. To make this more clear, we have now re-structured the sentence: “In sustained and transient infections we observed that all immune cell types significantly infiltrate the site of inoculation after exposure to SARS-CoV-2 (Fig 1f).
C3.19 Line 208 – Are all 2512 cells from the sustained infection group? Was there no detection of SARS-CoV-2 in cells from the other groups?	R3.19 This is explained in the same sentence from the main text: infected cells were almost exclusively found in the nasopharynx of participants with sustained infections (2505 out of 2512 cells with viral RNA)
C3.20 Fig 2g. I found this figure difficult to understand. The text mentions 5 cell groups, yet there are only four in the figure. The mean values and statistical analysis are not clear. The information is difficult to follow overall.	R3.20 The conventional ciliated cell state was left out of this figure as they don’t show dynamic behaviour. To clarify this, we have now reworded this sentence: “...we delineated the ciliated cell compartment into conventional ciliated plus four distinct dynamic cell states.”
C3.21 “when an individual is exposed to SARS-CoV-2, but manages to prevent the onset of viral spread” (507-508) Were these features not also observed in sustained infection patients?	R3.21 Correct, we did not intend to argue that these features were unique to non-sustained infections, nor that they helped prevent the onset of sustained infection in non-sustained cases. We have now clarified this.

C3.22 "Figure X, X, and X" (1034) should be corrected	R3.22 We have corrected this in the manuscript.
C3.23 The authors show good commitment to transparency of reporting.	R3.23 Thank you for acknowledging this.

Reviewer Reports on the First Revision:

Referees' comments:

Referee #1 (Remarks to the Author):

In this revised submission, the Teichmann group has enhanced the support and clarity of their original findings in response to insightful comments from their reviewers. In the opinion of this reviewer, what has materialized is one of the most comprehensive characterizations regarding how healthy individuals respond to a SARS-CoV-2 challenge from the moment of exposure. While the paper primarily maintains a descriptive approach, the extensive characterization of their three cohorts (sustained, transient, and abortive infections) offers unparalleled insight into the interaction between our immune system and the virus. Overall, I appreciate the authors' responsiveness to my critiques, but I do have two additional comments that I believe should be considered before publication:

Comment #1: The nomenclature of the three cohorts as "abortive," "transient," and "sustained" is subject to misinterpretation. Although the authors provide justification for these labels, it might be more appropriate to refer to their "abortive" cohort as "negative controls" or employ a term that clearly indicates the absence of an infection. The author's cohort referred to as "transient infection" could then be labelled as "abortive," while "sustained infection" could be relabeled as "acute infection" or a term that conveys a typical infection pattern. This adjustment not only aligns more accurately with the observed data but also eliminates the use of "persistent," as an infection lasting 7-10 days should not be categorized in this way. This is particularly important when considering long COVID, which some argue represents a genuine persistent infection.

Comment #2: The statement concerning "abortive infection" in CD8+ cells continues to be considered premature by this reviewer, given the data presented. The data found in Extended Data Fig 6d reveals only modest differences in viral reads captured between CD8+ T cells, macrophages, and secretory cells. These variations could be attributed to viral RNA captured on the cell surface during single-cell library construction and may reflect a bias for CD8+ T cells to retain this debris. While the authors' hypothesis may ultimately prove to be correct, the data on which this claim is presently based does not currently provide strong enough evidence to draw this conclusion without additional follow-up studies. It is recommended that the authors include the data but specify that the conclusions drawn should be considered speculative at this stage.

I congratulate the authors for generating this amazing data set and look forward to seeing it published.

Ben tenOever

Referee #2 (Remarks to the Author):

The authors have made huge efforts to address the mentioned critical points. However, I still have some doubts that the presented data support the conclusions and bold statements made in the manuscript.

R2.2. To overcome the problem, that the presented results are based on samples of only 6 participants (C.2.2) a bulk RNAseq validation dataset was added that includes nasal swabs and blood samples of additional 20 challenged individuals. Unfortunately, there is no information provided, how many of those 20 additional patients developed sustained infections. This information is essential and has to be provided. From Figure ED 7h one may estimate, that again only 7 of the 20 patients developed sustained infection (at least nasal samples seem to be available only for 7 patients). In the interferon response paragraph it is mentioned, that bulk data from 20 patients have been added, but this number is obviously not true. Even more important, Extended Data Figure 7h does not provide clear evidence that the interferon response is earlier in blood. In some nasal samples the interferon response seems to be earlier, in others the blood response. Showing the paired data for blood and nose would be helpful and more convincing here. This is even more important since the earlier interferon response in blood is one of the main results of this study mentioned also in the abstract.

R.2.3. A comparison of the gene expression between transient, abortive and sustained infection samples has been done at day -1 prior to infection both in blood and in nasopharyngeal samples. Interestingly, a higher expression of HLA-DQA2, a non-polymorphic non-canonical subunit of MHC class II, was observed in both the circulating and nasopharyngeal professional-antigen-presenting cells of participants that become abortively or transiently infected, compared to sustained infection cases that become COVID-19 patients. This result is important in so far, as HLA-DQA2 was in earlier studies was already been found to be higher expressed in mild compared to severe COVID-19 cases. Unfortunately, the authors conclude from this result that HLA-DQA2 expression is a biomarker for protection against productive SARS-CoV-2 infections. This is an absolute over-statement. Apart from the fact, that this result is based on very small case numbers, the individual expression levels of the different patients in the single cell data groups are not shown. As shown by the bulk RNA seq data (Ext data Fig 6h), there is obviously a strong variability of the expression of this gene from patient to patient and also within each. The overlap between the groups is immense and does not qualify the expression of this gene as a biomarker. There is in fact a predictive value, but not on the individual level.

Validation with the bulk RNA Seq data is not convincing. Extended data figure 6h shows the summarized data points from all timepoints. It is not explained

why here the transient group shows the lowest expression and why here all time points are included. The strong overlap between the sustained and abortive infection data points is not discussed. Finally, since here all time points (not only the day -1 data) have been included, I would assume that the data before infection alone are not convincing. These data has to be shown.

R2.4. Regarding the comparison between the sustained and transient groups regarding the early innate anti-viral response. The authors mentioned, that they now have added detailed differential gene expression analyses to their manuscript (Fig. 2h, Extended Data Fig. 6e), which will enable readers to investigate individual markers and responses over time and between infection groups. In fact, these two Figures show the differential gene expression pre-infection between sustained and abortive infection. With these two Figures no information provided on responses over time and nothing regarding the early anti-viral response.

Reviewer Figure 2.4 providing information on PRR expression over time is highly interesting and should be included in the manuscript.

R2.7. Other studies using also microfluidics based single cell sequencing were able to identify neutrophils and describe their functional role in COVID-19. One could assume that the used protocol for sample preparation and cryopreservation was not sufficient to keep the neutrophils. It is still mentioned in the paragraph "Cellular trends over time and infection groups" that the data sets contain all expected cell types; this is obviously not true. A more critical discussion on this limitation seem to be appropriate. In fact, Wigerblad et al. do not say that neutrophils cannot be found in scRNA Seq data. They explain that modifications to the data analysis pipeline, rather than to the existing scRNA-seq chemistries, can significantly increase the detection of human neutrophils in scRNA-seq. Having this in mind, it should be checked whether the suggested modification of the data analysis could rescue the neutrophile population.

R2.30 I would not agree that the validation data support the earlier IFN activation in blood (see R2.2.)

Referee #3 (Remarks to the Author):

The authors have done a tremendous job in addressing my original concerns. The use of a validation cohort and the analysis of differences between infection groups at baseline increases confidence in this study. The manuscript now reads with a much better flow with a clearer explanation of the data. There are a few points that require authors' attention.

- The methods section would benefit from a clearer definition of how the group actively assessed patients' health in the year following the challenge. For example, what were the criteria for "symptoms attributed to the viral challenge"?
- In several places in the manuscript, the authors refer to the "nose" to describe the nasopharynx. Please correct this. For example in line 1407: Were samples used for TR-qPCR from nasal swabs? Or nasopharyngeal swabs? The term "nose" does not clarify.
- 1054-1055: These statements are speculative. The study does prove whether it is a highly efficient relay or a dampening of local IFN signaling by the virus.
- 1059-1060: This sentence extrapolates the actual finding presented by the authors. The presented data show that individuals who did not develop sustained infections had higher expression of HLA-DQA2 at baseline. There does not seem to be evidence to establish a straight correlation between HLA-DQA2 expression and immune cell mobilization. According to Fig 1f, transient but not abortive infection groups had an early immune cell infiltration at NP.
- 1370-1397: Is this meant to be in the "PBMC isolation from peripheral blood" section?

Author Rebuttals to First Revision:

Comments referee #1

Comment by the referee	Response by the authors
C1.1 In this revised submission, the Teichmann group has enhanced the support and clarity of their original findings in response to insightful comments from their reviewers. In the opinion of this reviewer, what has materialized is one of the most comprehensive characterizations regarding how healthy individuals respond to a SARS-CoV-2 challenge from the moment of exposure. While the paper primarily maintains a descriptive approach, the extensive characterization of their three cohorts (sustained, transient, and abortive infections) offers unparalleled insight into the interaction between our immune system and the virus. Overall, I appreciate the authors' responsiveness to my critiques, but I do have two additional comments that I believe should be considered before publication:	R1.1 We thank Professor tenOever for his kind words and praise, and fully agree that based on the reviewer's comments we were able to significantly enhance the quality of our story and robustness of our findings.
C1.2 The nomenclature of the three cohorts as "abortive," "transient," and "sustained" is subject to misinterpretation. Although the authors provide justification for these labels, it might be more appropriate to refer to their "abortive" cohort as	R1.2 To prevent misinterpretation, we now clarify the exact meaning of each group and explain our reasoning behind the chosen nomenclature in an additional section in the methods that we refer to in the main text (Methods, subheading Infection group nomenclature).

"negative controls" or employ a term that clearly indicates the absence of an infection. The author's cohort referred to as "transient infection" could then be labelled as "abortive," while "sustained infection" could be relabeled as "acute infection" or a term that conveys a typical infection pattern. This adjustment not only aligns more accurately with the observed data but also eliminates the use of "persistent," as an infection lasting 7-10 days should not be categorized in this way. This is particularly important when considering long COVID, which some argue represents a genuine persistent infection.	We agree that the nomenclature of the infection groups needs to be carefully thought through to prevent misinterpretation. However, we strongly fear that the proposed changes would add additional confusion and make the group names multi-interpretable. We believe that the use of "acute" is not suitable for the sustained group, because this term is often used to refer to timing of an infection event and could therefore easily be confused with early versus late samples analyzed in our study. We would also prefer to be cautious in changing our transient to abortive group. As defined in the Oxford Reference Dictionary, abortive infections refer to events where a virus was not able to replicate. The fact that we detect virus using qPCR in transient but not abortive infections suggests that the virus has replicated at least to some extent in the transient infections to borderline measurable levels. We therefore think that abortive infection is not the appropriate name for the transient group. Lastly, referring to the abortive group as negative controls, would imply that these participants were not exposed to live virus at all, thus serving as true negative controls, while these participants were also challenged and are in fact a very unique perturbation group that is nearly impossible to capture outside a challenge model. Additionally, we have sought advice from various experts in the COVID-19 Human Challenge Study Consortium and we all agreed on the above nomenclature.
C1.3 The statement concerning "abortive infection" in CD8+ cells continues to be considered premature by this reviewer, given the data presented. The data	R1.3 We agree with the reviewer that our data cannot distinguish between viral reads captured at the surface of cells versus those present within the cell. We have therefore changed our concluding statement to the following:

found in Extended Data Fig 6d reveals only modest differences in viral reads captured between CD8+ T cells, macrophages, and secretory cells. These variations could be attributed to viral RNA captured on the cell surface during single-cell library construction and may reflect a bias for CD8+ T cells to retain this debris. While the authors' hypothesis may ultimately prove to be correct, the data on which this claim is presently based does not currently provide strong enough evidence to draw this conclusion without additional follow-up studies. It is recommended that the authors include the data but specify that the conclusions drawn should be considered speculative at this stage.	“Our results therefore show that, while epithelial cells can support viral replication, mucosal CD8+ T cells are either infected non-productively or might capture viral fragments from surrounding cells.”
C1.4 I congratulate the authors for generating this amazing data set and look forward to seeing it published. Ben tenOever	R1.4 Many thanks for the kind words and insightful comments.

Comments referee #2

Comment by the referee	Response by the authors
C2.1 The authors have made huge efforts to address the mentioned critical points. However, I still have some doubts that the presented data support the conclusions and bold statements made in the manuscript.	R2.1 Many thanks for acknowledging our huge efforts to address your points so far. We strongly feel that our data support the conclusions and have outlined further evidence below.
C2.2 R2.2. To overcome the problem, that the presented results are based on samples of only 6 participants (C.2.2) a bulk RNAseq validation dataset was added that includes nasal swabs and blood samples of additional 20 challenged individuals. Unfortunately, there is no information provided, how many of those 20 additional patients developed sustained infections. This information is essential and has to be provided. From Figure ED 7h one may estimate, that again only 7 of the 20 patients developed sustained infection (at least nasal samples seem to be available only for 7 patients). In the interferon response paragraph it is mentioned, that bulk data from 20 patients have been added, but this number is obviously not true. Even more important, Extended Data Figure 7h does not provide clear evidence that the interferon response is earlier in blood. In some nasal samples the interferon	R2.2 We apologise for the oversight and have now included detailed information about the composition of our validation cohort. This information is included in Extended Data Table 1q. Importantly, this table shows that our validation cohort included an additional 12 sustained infection cases, which triples the total amount of sustained infections reported in this study. We disagree that Extended Data Figure 7h is not a clear validation of our observation that interferon signaling in blood precedes the interferon response at the site of infection. In fact, all of the cyan lines of the blood samples exhibit a strong upregulation of the interferon signaling pathway before day 5, while the clear majority of the nasal samples in magenta do not exhibit activation before this time point. In the re-revised manuscript, we now qualify and describe this observation using a paired ranking-based statistical test which reveals that interferon signaling is significantly earlier upregulated in blood samples compared to nasal samples at a p value of 0.008669, and with a median difference in upregulation of interferon signaling of 2 days (Subheading: Interferon response in blood before nose).

response seems to be earlier, in others the blood response. Showing the paired data for blood and nose would be helpful and more convincing here. This is even more important since the earlier interferon response in blood is one of the main results of this study mentioned also in the abstract.

While the provided analysis in the revised manuscript was already paired between time points of each participant, we agree that pairing the blood and nasal samples within this analysis at each time point can give additional confidence to the finding. In **Reviewer Figure R2.2** and in **Extended Data Figure 7i** we show a comparative analysis of the time points at which paired blood and nasal gene expression were measured, which indeed confirms that interferon signaling in blood samples is significantly upregulated at early time points compared to nose, and that these differences become indistinguishable at later time points.

	Reviewer Figure R2.2: Boxplots showing bulk RNA-seq measurements of type I interferon signaling in blood and nasal (mid-terminate) swabs over time as shown in Extended Data Fig. 7h, but only focussing on samples with a paired blood and nasopharyngeal measurement to perform paired analyses. P values of a paired Mann-Whitney U test comparing nasal and blood samples at each time point are shown at the top. Preinfection baseline nasal and blood samples were collected at the day before and the same day as the inoculation, respectively.
C2.3 R.2.3. A comparison of the gene expression between transient, abortive and sustained infection samples has been done at day -1 prior to infection both in blood and in nasopharyngeal samples. Interestingly, a higher expression of HLA-DQA2, a non-polymorphic non-canonical subunit of MHC class II, was observed in both the circulating and nasopharyngeal professional-antigen-presenting cells of participants that become abortively or transiently infected, compared to sustained infection cases that become COVID-19 patients. This result is important in so far, as HLA-DQA2 was in earlier studies was already been found to be higher expressed in mild compared to severe COVID-19 cases. Unfortunately, the authors conclude from this result that HLA-DQA2 expression is a biomarker for protection against productive SARS-CoV-2 infections. This is an absolute over-statement. Apart from the fact, that this result is based on very small case numbers, the individual expression levels of the different patients in the single cell data groups are not shown. As shown by	R2.3 Thank you for your comment. While we had already stated in the last version that HLA-DQA2 is a “possible protective biomarker”, we have further toned down the description of this observation by removing the term biomarker in this context. While we agree it is better to be conservative in our wording, we would like to note that we tested the predictive power of the expression of this gene in a cross validation analysis which showed good performance as a biomarker in various professional antigen-presenting cell types (Extended Data Figure 6f-g, ‘good’ defined as AUC>0.75 (Raghavan et al., Adv Nutr, 2016)). In line with this, the predictive value of this gene is only expected to be present when looking in a cell-type-resolved manner. This will partially explain the increased heterogeneity in the bulk RNA-samples shown, where professional antigen-presenting cells will only be responsible for a small fraction of the signal. We agree that showing the HLA-DQA2 expression of each participant is important to show. We now included Reviewer Figure R2.3 as Extended Data Figure 6q to show this. In agreement with our cross-validation results, this figure indeed shows that HLA-DQA2 expression is low or absent in all but one sustained infection case, while being highly expressed in more than half of the non-sustained cases, depending on the cell type.

the bulk RNA seq data (Ext data Fig 6h), there is obviously a strong variability of the expression of this gene from patient to patient and also within each. The overlap between the groups is immense and does not qualify the expression of this gene as a biomarker. There is in fact a predictive value, but not on the individual level.

Validation with the bulk RNA Seq data is not convincing. Extended data figure 6h shows the summarized data points from all timepoints. It is not explained why here the transient group shows the lowest expression and why here all time points are included. The strong overlap between the sustained and abortive infection data points is not discussed. Finally, since here all time points (not only the day -1 data) have been included, I would assume that the data before infection alone are not convincing. These data has to be shown.

Reviewer Figure R2.3: Boxplot showing the preinfection expression of HLA-DQA2 in nasopharyngeal (left) and circulating (right) professional antigen presenting cell types, across participants and the infection groups.

C2.4 Regarding the comparison between the sustained and transient groups regarding the early innate anti-viral response. The authors mentioned, that they now have added detailed differential gene expression analyses to their manuscript (Fig. 2h, Extended Data Fig. 6e), which will enable readers to

R2.4 We have shared **Extended Data Tables 1n** and **1o** in the last revision and referenced them in the text. They comprise results from differential gene analysis for PBMCs and nasopharyngeal cells, respectively. Each column is a particular cell type and infection group, and shows the difference in expression between the baseline sample of that cell type and a given sampling time point, with p value. Thus, readers can investigate whether genes differ significantly between time

investigate individual markers and responses over time and between infection groups. In fact, these two Figures show the differential gene expression pre-infection between sustained and abortive infection. With these two Figures no information provided on responses over time and nothing regarding the early anti-viral response. Reviewer Figure 2.4 providing information on PRR expression over time is highly interesting and should be included in the manuscript.	points for a given cell type and infection group combination, and indeed find information on responses over time and the anti-viral response. While we highly appreciate your enthusiasm about the expression of the PRRs that we highlighted in our previous rebuttal, limited figure space makes the inclusion of specific gene sets difficult, including many interesting ones such as PRR. However, we welcome readers to browse the Extended Data Tables 1n-o and explore genes of interest through our interactive web browser (covid19cellatlas.org). To ensure that the PRR expression dynamics figure is published alongside this manuscript and visible to the readers, we will opt-in for ‘Transparent peer review’ to publish our rebuttal including the reviewer figures.
C2.5 R2.7. Other studies using also microfluidics based single cell sequencing were able to identify neutrophils and describe their functional role in COVID-19. One could assume that the used protocol for sample preparation and cryopreservation was not sufficient to keep the neutrophils. It is still mentioned in the paragraph “Cellular trends over time and infection groups” that the data sets contain all expected cell types; this is obviously not true. A more critical discussion on this limitation seem to be appropriate. In fact, Wigerblad et al. do not say that neutrophils cannot be found in scRNA Seq data. They explain that modifications to the data analysis pipeline, rather than to the existing scRNA-seq chemistries, can significantly increase the detection	R2.5 The main text now explicitly states that we detect ‘almost all expected cell types’. We agree that identifying neutrophils in datasets like the one presented in this study warrants extra care and effort. During the cell type annotation progress, we extensively tested different filtering criteria and annotation strategies, including the suggestions by Wigerblad et al (J Immunol 2022 Aug 15;209(4):772-782. doi: 10.4049/jimmunol.2200154). However, these efforts did not yield any cell type clusters that we could confidently annotate as neutrophils. It is noteworthy that Wigerblad et al analyzed red blood cell depleted samples, while we studied peripheral blood mononuclear cells (PBMCs) where neutrophils are removed during sample collection and hence not expected to be in our PBMC data. This is in line with our previous work where we did not detect any neutrophils in PBMC samples (Yoshida et al 2022, doi.org/10.1038/s41586-021-04345-x).

of human neutrophils in scRNA-seq. Having this in mind, it should be checked whether the suggested modification of the data analysis could rescue the neutrophile population.	Furthermore, as noted by the reviewer the preparation and cryopreservation of the nasopharyngeal samples required in the sample pipeline of this study would also have a significant effect on our ability to capture granulocyte populations including neutrophils due to their fragility. In a comparison, looking at the difference in abundance of tissue resident cell type captured from fresh vs cryopreserved nasopharyngeal swabs, on which our dissociation protocol is based (Ziegler et al. 2021, https://doi.org/10.1016/j.cell.2021.07.023), they noted a underrepresentation of granulocytes recovered reflected in our dataset. To highlight the possible limitation of not identifying neutrophils using our assays, we state the following in our manuscript: “We also note that neutrophils, which play an important role in COVID-19, are frequently underrepresented in microfluidics based scRNA-seq. This is likely further attenuated by cryopreservation of samples used within this study.”
C2.6 R2.30 I would not agree that the validation data support the earlier IFN activation in blood (see R2.2.)	R2.6 We have addressed this point in R2.2.

Comments referee #3

Comment by the referee	Response by the authors
C3.1 The authors have done a tremendous job in addressing my original concerns. The use of a validation cohort and the analysis of differences between infection groups at baseline increases confidence in this study. The manuscript now reads with a much better flow with a clearer explanation of the data. There are a few points that require authors' attention.	R3.1 Thank you very much for the positive feedback.
C3.2 The methods section would benefit from a clearer definition of how the group actively assessed patients' health in the year following the challenge. For example, what were the criteria for "symptoms attributed to the viral challenge"?	We agree with the reviewer that this is important information to include, particularly with the ongoing global effort to better understand long-COVID-19. We have therefore added an additional paragraph in the method section: "No participants enrolled in the study were observed to present with any long-COVID symptoms at this final time point (1 year), which included an interview by a study clinician to assess for symptoms and a complete physical examination. The UPSIT scores for all participants had returned to baseline and no other symptoms were reported, with physiological observations and physical examination of vital signs were all seen to be normal (including temperature, heart rate, blood pressure, respiratory rate, saturation of peripheral oxygen level [SpO2], spirometry, electrocardiogram)."

C3.3 In several places in the manuscript, the authors refer to the “nose” to describe the nasopharynx. Please correct this. For example in line 1407: Were samples used for TR-qPCR from nasal swabs? Or nasopharyngeal swabs? The term “nose” does not clarify.	Thank you for drawing this to our attention. All the references to the “nasal/nose” in the text of the manuscript have now been double checked to ensure they are correct and altered to be more specific where necessary. For example when discussing the site of the samples used for the RT-PCR results“ ..twice daily samples (swabs) were taken from both nose (mid-turbinate) and throat (pharyngeal)....”
C3.4 1054-1055: These statements are speculative. The study does prove whether it is a highly efficient relay or a dampening of local IFN signaling by the virus.	R3.4 We agree that this is speculative and have therefore toned down this statement.
C3.5 1059-1060: This sentence extrapolates the actual finding presented by the authors. The presented data show that individuals who did not develop sustained infections had higher expression of HLA-DQA2 at baseline. There does not seem to be evidence to establish a straight correlation between HLA-DQA2 expression and immune cell mobilization. According to Fig 1f, transient but not abortive infection groups had an early immune cell infiltration at NP.	R3.5 We agree that the responses described here were speculative and inferred. We have therefore rephrased this discussion point to make it more factual from ‘Our data suggest that individuals with high HLA-DQA2 expression can rapidly mobilize cellular responses to achieve clearance.’ to

	'Our data suggest that individuals with high HLA-DQA2 expression are better at preventing the onset of a sustained viral infection'.
C3.6 1370-1397: Is this meant to be in the “PBMC isolation from peripheral blood” section?	R3.6 This paragraph was indeed wrongly placed. We have now corrected this under the 'Study participants and design' section of the methods. Many thanks for the careful proof-reading.

Reviewer Reports on the Second Revision:

Referee #1 (Remarks to the Author):

Dear Editor,

After carefully reviewing the revised submission and the accompanying critiques and responses, I believe that the authors have adequately addressed the concerns raised by myself and the other reviewers. One of the more significant concerns that remains pertains to the pipeline used for single-cell RNA sequencing (scRNA-Seq) and its ability to capture neutrophils before they undergo cell death. The authors' modification of the text to indicate that they captured "most of the cells" is a reasonable acknowledgment of this limitation. Overall, I find that this paper presents a comprehensive dataset that effectively consolidates various findings related to COVID-19 into a cohesive narrative. This work contributes significantly to our understanding of the disease and provides a valuable resource for the scientific community.